

# Assessing the glacier projection uncertainties in the Patagonian Andes (40–56° S) from a catchment perspective

Rodrigo Aguayo[1], Fabien Maussion[2,3], Lilian Schuster[2], Marius Schaefer[4], Alexis Caro[5], Patrick Schmitt[2], Jonathan Mackay[6,7], Lizz Ultee[8], Jorge Leon-Muñoz[9,10], and Mauricio Aguayo[1]

[1]Centro EULA, Facultad de Ciencias Ambientales, Universidad de Concepción, Concepción, Chile.
[2]Department of Atmospheric and Cryospheric Sciences (ACINN), Universität Innsbruck, Innsbruck, Austria
[3]School of Geographical Sciences, University of Bristol, Bristol, UK
[4]Instituto de Ciencias Físicas y Matemáticas, Universidad Austral de Chile, Valdivia, Chile
[5]Univ. Grenoble Alpes, CNRS, IRD, INRAE, Grenoble-INP, Institut des Géosciences de l'Environnement, Grenoble, France
[6]British Geological Survey, Keyworth, Nottingham, United Kingdom
[7]School of Geography, Earth and Environmental Sciences, University of Birmingham, Edgbaston, Birmingham, UK
[8]Department of Geology, Middlebury College, Middlebury, US
[9]Departamento de Química Ambiental, Universidad Católica de la Santísima Concepción, Concepción, Chile
[10]Centro Interdisciplinario para la Investigación Acuícola (INCAR), Concepción, Chile

*Correspondence to*: Rodrigo Aguayo (rodaguayo@udec.cl)

**Abstract.** Glaciers are retreating globally and are projected to continue to lose mass in the coming decades, directly affecting
downstream ecosystems through changes in glacier runoff. Estimating the future evolution of glacier runoff involves several sources of uncertainty in the modelling chain, which to date have not been comprehensively assessed on a regional scale. In this study, we used the Open Global Glacier Model (OGGM) to estimate the glacier evolution of each glacier (area > 1 km$^2$) in the Patagonian Andes (40–56° S), which together represent 82% of the glacier area of the Andes. We used different glacier inventories (n = 2), ice thickness datasets (n = 2), historical climate datasets (n = 4), general circulation models (GCMs; n = 25   10), emission scenarios (SSPs; n = 4), and bias correction methods (BCMs; n = 3) to generate 1,920 possible scenarios over the period 1980–2099. For each scenario and catchment, glacier runoff and melt on glacier time series were characterized by ten glacio-hydrological signatures (i.e., metrics). We used the permutation feature importance of random forest regression models to assess the relative importance of each source on the signatures of each catchment. Considering all scenarios, 61% ± 14% of the catchment area (30% ± 13% of glacier area) has already peaked in terms of glacier melt (year 2020), and 43% ± 30   8% of the catchment area (18% ± 7% of glacier area) will lose more than 80% of its volume this century. Considering the melt on glacier signatures, the future sources of uncertainty (GCMs, SSPs and BCMs) were the main source in only 18% ± 21% of the total catchment area. In contrast, the reference climate was the most important source in 78% ± 21% of the catchment area, highlighting the importance of the choices we make in the calibration procedure. The results provide a basis for prioritizing future efforts (e.g., improve reference climate characterization) to reduce glacio-hydrological modelling gaps in poorly 35   instrumented regions, such as the Patagonian Andes.



## 1 Introduction

Glaciers are retreating worldwide (Hugonnet et al., 2021) and are projected to continue to lose mass (Marzeion et al., 2020). Recent projections by Rounce et al. (2023) indicate that glaciers will lose 26 ± 6% (1.5 °C scenario) to 41 ± 11% (4 °C scenario) of their mass by 2100 (median ± 95% confidence interval), contributing between 90 ± 26 and 154 ± 44 mm to sea level rise,

respectively. The rapid glacier shrinkage has led to cascading effects on downstream systems (Milner et al., 2017; Huss et al., 2017), affecting the availability and quality of water resources (IPCC, 2022), and causing changes in the ecological (Cauvy-Fraunié and Dangles, 2019) and socioeconomic (Rasul and Molden, 2019) aspects of downstream environments. As glaciers retreat increases due to climate change, hazards such as glacial lake outburst floods (GLOFs) are undergoing shift in their occurrence rate (Veh et al., 2022), posing an increasing risk to nearby communities (Taylor et al., 2023).

One of the most important impacts of glaciers on downstream systems is the contribution of meltwater to streamflow (Huss and Hock, 2018), which is essential for irrigation, industry, domestic use, hydropower and ecosystems (Immerzeel et al., 2020; Viviroli et al., 2020). However, as glaciers continue to shrink, the reliability and quantity of this water reserve becomes increasingly uncertain, potentially increasing drought stress (Kaser et al., 2010; Van Tiel et al., 2021, 2023; Pritchard, 2019). Ultee et al. (2022) showed globally that accounting for glacier runoff reduces simulated drought frequency and severity, even

in basins with low glacier cover (< 2%). The buffering effect is higher in moderately glaciated arid regions, such as the Central Andes, and is projected to increase through the 21st century. In this region, glaciers have provided an important drought mitigation capacity during the current Mega Drought (Ayala et al., 2020; McCarthy et al., 2022), which is unprecedented in recent centuries according to dendrochronological studies (Garreaud et al., 2017; Morales et al., 2020).

Recent global estimates suggest that Andean glaciers are likely to be one of the largest per unit area contributors to sea level

rise, with a contribution of 0.057 ± 0.006 mm SLE yr$^{-1}$ (-20.7 ± 2.1 Gt yr$^{-1}$) representing 7.7% of the global mass loss between 2000 and 2019 (mean ± 95% confidence interval) (Hugonnet et al., 2021). Glaciers in the Patagonian Andes account for 96% of the total ice loss in the Southern Andes (Braun et al., 2019), which has accelerated in recent decades (Davies and Glasser, 2012; Dussaillant et al., 2019). Due to the high precipitation levels in the Patagonian Andes (Garreaud et al., 2013), the contribution of glaciers to regional water supply is generally low, with glacier runoff serving as a flow buffer during dry

periods rather than a major source of streamflow (Ruiz et al., 2022). Nevertheless, the accelerated mass loss over recent decades has significantly increased streamflow in many Patagonian rivers (Masiokas et al., 2019; Vries et al., 2023), some of which have only begun to show significant trends in the last decade (e.g., Santa Cruz; Pasquini et al., 2021).

Despite efforts to improve the understanding of glacier processes in the Patagonian Andes, there are still important limitations due to the lack of ground-based validation data (Masiokas et al., 2020; Aguayo et al., in review). Lowland meteorological

stations far from glaciers have had to be used, but their scarcity and lack of continuity remain significant issues in this region



(Table 1). To address these limitations, many modelling studies have used dynamic and/or statistical downscaling methods based on climate reanalyses (Table 1). However, the different approaches and data sources have diverged towards systematic overestimations according to numerical simulations of regional moisture fluxes (Sauter, 2020). Despite the severe lack of data on melt patterns and snow accumulation in the upper plateaus of the Patagonian Icefields (Bravo et al., 2019a, b), most regional
modelling efforts have focused on this region (Table 1). In this area, glacier modelling has generally relied on energy balance approaches based on downscaled reanalysis data. Only two studies have modelled the regional hydrological contribution of the Patagonian glaciers (Mernild et al., 2017; Caro et al., 2023). Although recent modelling efforts have benefited from the increased availability of geodetic mass balances to calibrate and validate surface mass balance models (Table 1), important sources of uncertainty in the future evolution of Patagonian glaciers remain.

There are several sources of uncertainty in the modelling chain of glacier projections. At the global scale, results from the Glacier Model Intercomparison Project Phase 2 (GlacierMIP2) showed that the emission scenario is the largest source of uncertainty by the end of the century, but the uncertainty from the glacier models, which use different data sources and calibration setups, is the largest source until 2050 (Marzeion et al., 2020). Locally, several studies have shown that individual choices during model initialization and calibration, such as the historical climate (Compagno et al., 2021; Watanabe et al.,
2019), the glacier inventory (Li et al., 2022), the ice thickness (Gabbi et al., 2012), and the downscaling strategy (Schuster et al., 2023), have an impact on glacier evolution. However, few studies have examined the influence of multiple components of the modelling chain on projected glacio-hydrological changes. Huss et al (2014) found that winter snow accumulation and the glacier retreat model have the greatest influence on the glacier runoff projections in the Findelengletscher basin (Switzerland), while the downscaling strategy, calibration data quality and the surface mass model are of secondary importance. Mackay et
al. (2019) used hydrological signatures, which are quantitative metrics that describe the dynamic properties of hydrological time series (McMillan, 2021), to measure changes in the hydrology of the Virkisá basin (southern Iceland). They found that the main source of uncertainty were the climate model chain components (global circulation models and emission scenarios), but for certain hydrological signatures, the most important source was the representation of glacio-hydrological processes. Overall, adding additional data to the calibration of glacio-hydrological processes has shown to be more important than
increasing the complexity of the model (Van Tiel et al., 2020).

In this study, we investigated the importance of six sources of uncertainty in the glacier modelling chain for ten glacio-hydrological signatures (i.e., metrics) that characterize the hydrological regime of each catchment. For this, we used the Open Global Glacier Model (OGGM) to project the evolution of each glacier (area > 1 km$^2$) in the Patagonian Andes (40–56° S) over the period 1980–2099. The experimental setup used different glacier inventories (n = 2), ice thickness datasets (n = 2),
historical climates (n = 4), global circulation models (n = 10), emission scenarios (n = 4) and bias correction methods (n = 3), resulting in 1920 potential evolution scenarios. Finally, we used the permutation feature importance of random forest





regression models to assess the importance of each source of uncertainty on the different glacio-hydrological signatures of each catchment.

**Table 1. Regional surface mass balance models applied in the Patagonian Andes (40–56º S). In parenthesis the initial spatial resolution of the gridded climate. AWS: Automatic weather station. PP: Precipitation. T2M: Air temperature at 2m. GMB: Geodetic mass balance. The area acronyms are defined in Section 2.**

| Area | Period | Reference climate | Downscaling | Target resolution | Timestep | SMB model | Calibration/validation of mass balance | Reference |
|---|---|---|---|---|---|---|---|---|
| NPI | 1975–2099 | Output from WRF run (5 km) based on NCEP-NCAR (2.5º) | T2M: Constant lapse rate of 6.5 ºC km$^{-1}$. PP: Gradient of 0.05% m$^{-1}$. AWSs were used for validation. | 450 m | Daily | Simplified energy balance | GMB: Willis et al. (2012) and Rignot et al. (2003) | Schaefer et al. (2013) |
| GCN | 2000–2005 | PP: NCEP-NCAR (2.5º). T2M: AWSs | T2M: Constant lapse rate of 5.8 ºC km$^{-1}$. PP: Gradient of 0.15% m$^{-1}$. Orographic precipitation model as an alternative. | 90 m | Daily | Degree-day model | Ablation stakes for validation | Weidemann et al. (2013) |
| NPI + SPI | 1979–2012 | Output from RACMO run based on ERA-Interim (~80 km) | No downscaling. AWSs were used for model evaluation | 5.5 km | 6 hours | Energy balance (RACMO2.3) | Ice cores for validation | Lenaerts et al. (2014) |
| SPI | 1975–2011 | Follows Schaefer et al. (2013) | Follows Schaefer et al. (2013) | 180 m | Daily | Simplified energy balance | Parameters from Schaefer et al. (2013). Ablation stakes and ice cores for validation | Schaefer et al. (2015) |
| Andes | 1979–2014 | NASA MERRA (~0.5º) | Downscaling based on MicroMet (Liston and Elder, 2006) | 1 km | 3 hours | Energy balance (SnowModel) | SMB observations of seven glaciers (only one in the Patagonian Andes) | Mernild et al. (2017) |
| NPI + SPI | 1976–2050 | RegCM4.6 output (10 km) based on MPI-ESM-MR model | No downscaling. AWSs were used for model evaluation | 10 km | Daily | Energy balance | Validation based on multiple GMBs (NPI and SPI) | Bravo et al. (2021) |
| NPI + SPI | 1980–2015 | RegCM4.6 output (10 km) based on ERA-Interim model | Follows Schaefer et al. (2013). CR2MET was used for validation | 450 m | 3 hours | Simplified energy balance | Calibration based on SMB estimates from Minowa et al. (2021) | Carrasco-Escaff et al. (2023) |
| CDI | 2000–2022 | ERA5 (0.25º) and AWSs | Several methods depending on the variable | 200 m | 3 hours | Four different models | Multiple strategies using ablation stakes, geodetic mass balance and mass budgeting | Temme et al. (2023) |
| Andes | 2000–2019 | Bias-corrected version of TerraClimate (4 km) | Lapse rates depend on the glaciological zones | f (glacier area) | Monthly | Degree-day model (OGGM) | GMB from Hugonnet et al. (2021) and volume from Farinotti et al. (2019) | Caro et al. (2023) |



## 2 Study area

Our study area comprises the Patagonian Andes (40–56º S; Fig. 1), which is characterized by an almost pristine environment, where freshwater ecosystems interact with the adjacent coastal system. In this region, glaciers are a crucial natural feature of

the terrestrial geography and their seasonal melting is essential for the long-term sustainability of the local ecosystems and coastal human populations (Iriarte et al., 2014). Glaciers in the Patagonian Andes cover an extensive area of 25,886 km$^2$, which represents 82% of the total glacier area of the Andes (RGI Consortium, 2017). This region includes the Northern and Southern Patagonian Icefields (NPI and SPI), which form the largest freshwater reservoir in the Southern Hemisphere outside of Antarctica, with a total area of 17,195 km$^2$ in 2011 (Davies and Glasser, 2012), and an estimated ice volume of 4,756 ± 923

km$^3$ (Millan et al., 2019).

We selected 847 catchments and aggregated them into nine hydrological zones (Fig. 1). Each catchment has at least one glacier and presents a glacier area greater than 0.1% of the catchment area. The hydrological zones were selected based on the spatial patterns of precipitation and temperature that have shown a high explanatory power of recent glacier changes (Caro et al., 2021). Along the latitudinal gradient covered by the nine zones, the mean annual 0 °C isotherm decreases with elevation from

about 3,000 m a.s.l. in the northern area to less than 1,000 m a.s.l. in the southern area (Condom et al., 2007; Carrasco et al., 2008). The northern area (~ 41–46º S; Fig. 1) is characterized by two zones (PPY and PCA) that aggregate large catchments with a low glacier area. The Northern Patagonian Icefield (NPI; ~ 46–48º S) was divided into two zones according to its main aspect (NPI-E and NPI-W). The eastern side coincides with the location of the Baker River Basin, one of the catchments with the largest glacier area in the study area and the focus of regional (Dussaillant et al., 2019) and global (Huss and Hock, 2018)

glacio-hydrological studies. The Southern Patagonian Icefield (SPI; ~ 48–52º S) was divided latitudinally according to the main catchments on the eastern side (Pascua in SPI-N, Santa Cruz in SPI-C and Grey in SPI-S). Finally, the southern area was divided into the Gran Campo Nevado (GCN; ~ 52–54º S) and the Cordillera Darwin Icefield (CDI; < ~ 54º S), which are characterized by many small catchments. In contrast to the rest of the area, both zones receive uniform precipitation throughout the year, with no clear seasonality.






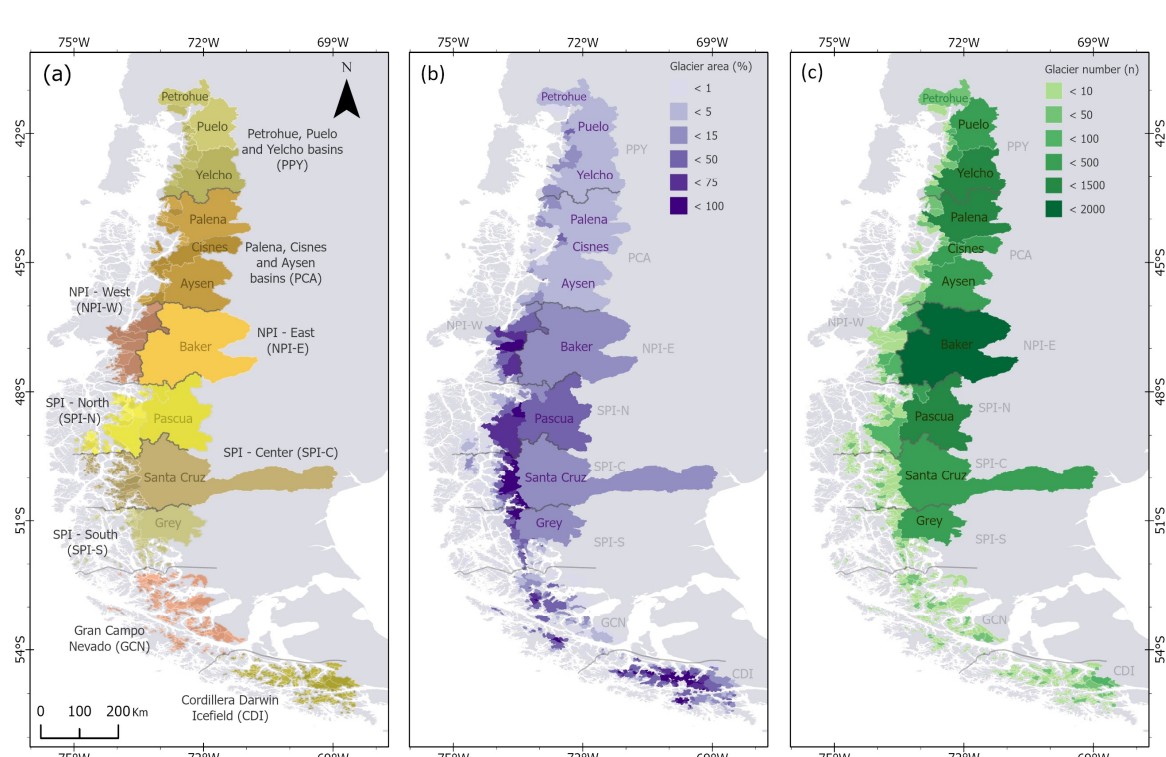

**Figure 1. Study area. a) Hydrological zones (n = 9) for the 847 catchments. The names in grey correspond to the names of the main catchments (area > 5,000 km²) in the study area. b) Glacier area for each catchment. c) Number of glaciers in RGI6 per catchment.**

## 3 Methods

### 3.1 The Open Global Glacier Model (OGGM)


We used the Open Global Glacier Model v1.5.4 (OGGM, Maussion et al., 2019) to model the evolution of all the glaciers in the study area. The OGGM is an open-source model that couples a surface mass balance model with a model of glacier dynamics. The model has been used in global studies (Farinotti et al., 2019; Rounce et al., 2023; Marzeion et al., 2020; Malles et al., 2023) and regional hydrological studies (Zhao et al., 2023; Khadka et al., 2020; Mannan Afzal et al., 2023; Shafeeque

and Luo, 2021; Caro et al., 2023). The surface mass balance model is based on an extended version of the temperature-index model used by Marzeion et al. (2012). In this approach, the monthly mass balance ($B_i$) at elevation z is calculated as:

$$B_i(z) = P_f \cdot P_i^s(z) - \mu^* \cdot \max(T_i(z) - T_{melt}, 0), \tag{1}$$



where $P_f$ is a precipitation factor used to account for measurement biases in mountainous topography, to further downscale precipitation to the glacier resolution, and to account for missing processes (e.g., debris cover, firn densification, avalanches) not explicitly included in the mass balance. $P_i^s$ and $T_i$ are the monthly solid precipitation and air temperature, $\mu^*$ is the
temperature sensitivity of the glacier, and $T_{melt}$ is the monthly mean air temperature above which ice melt is assumed to occur. The air temperature at each reference grid elevation is adjusted to the glacier surface using constant lapse rates of -6.5 ºC km[-1], a value commonly used in the study area (Table 1). The solid precipitation fraction is calculated using an upper and a lower temperature threshold. When the temperature is within the range defined by the thresholds, the solid precipitation varies linearly between 100% and 0% at the lower and upper limit, respectively. The contributions of positive degree-months and
solid precipitation are combined to calculate the monthly mass balance, which is used to update glacier geometry annually.

Glacier geometry (i.e., outlines) is derived from global or local inventories that are projected onto a local grid for each glacier, with a spatial resolution that depends on the area of each glacier (between 10 and 200 m). In this study, the elevation data is obtained from NASADEM (NASA JPL, 2020). Based on this dataset, each glacier was divided into binned elevation bands following the algorithm proposed by Werder et al. (2020). The ice dynamics flowline model of OGGM uses a shallow ice
approximation (SIA) with a depth-integrated flowline model to explicitly compute the flux of ice along the glacier.

In this study, we set the precipitation factor ($P_f$) to 1.0 to assess the influence of different reference climates on the evolution of each glacier (Fig. 2), assuming that the estimated precipitation from the different products corresponds to the "true" values. Given the low availability of regional ground-based data, we used the default thresholds for melting ($T_{melt}$ = -1 ºC since melting can still occur on some days when the monthly mean temperature is below 0 ºC) and accumulation (0 ºC and 2 ºC). The frontal
ablation of marine-terminating and lake-terminating glaciers was not simulated explicitly. However, Malles et al. (2023) recently showed that the mass-balance model (through different temperature sensitivities) implicitly accounts for the effect of frontal ablation when calibrated against the Hugonnet et al. (2021) observations, resulting in relatively small changes in the projections. This is an acknowledged shortcoming of our study and should be further investigated in future studies.

The calibration of each glacier consisted of a newly developed iterative process that involves three parameters: the temperature
sensitivity ($\mu^*$; Eq. 1), the composite deformation-sliding parameter (A; encapsulating basal sliding and ice deformation, following Glen's flow law; Eq. 3 in Maussion et al. 2019), and finally, the spin-up temperature ($T_{spinup}$; used to find a historical glacier state). The calibration procedure, shown in Fig. 2, unfolds through the following steps:

1.  Define a new value of $\mu^*$ (Eq. 1). The first value is obtained by matching the modelled specific mass balance with the geodetic mass balance of Hugonnet et al. (2021). This is calculated using the period 2000 to 2020, the reference climate



and the static surface geometry, which refers to the outline obtained from the Randolph Glacier Inventory (RGI; see next section).

2.  Compute an apparent mass balance using the static surface geometry and the reference climate in the geodetic mass balance period (2000–2020). In particular, this step searches for a mass balance residual to add to the mass balance profile so that the average specific mass balance is zero. This is needed for the underlying equilibrium assumption during the

inversion, as described in Maussion et al. (2019).

3.  Use the derived apparent mass balance for an inversion for the underlying glacier bed. Throughout this inversion, parameter A is defined such that the resulting inversion glacier volume matches the estimates for each hydrological zone defined in Fig. 1. The inversion method follows Maussion et al. (2019) when the sliding parameter is set to 0.

4.  The next step is to find a glacier state in the past (first try 1980) from which a dynamic glacier run to the RGI date (approx.

year 2000) results in the given RGI area. To define different glacier states in the past, the temperature spin-up $T_{spinup}$ was added to the reference climate during a 20-year constant mass balance run (first guess of $T_{spinup}$ is -1 °C). The constant mass balance is defined as the mean mass balance given by the reference climate between 1980 and the RGI date. How consecutive guesses of $T_{spinup}$ are found is described in Appendix A. If the resulting glacier is too large even when we start from an ice-free initial glacier state in the past, or the resulting glacier is too small and the algorithm grows the glacier

outside the domain, a shorter spin-up period is tried two times (starting at year 1985 or 1990). If the spin-up period is shortened, a fixed geometry volume is calculated by going backwards to 1980, using the calculated mass change on the constant surface geometry. This is done to have a continuous volume time-series for all glaciers. We only move on if this step has successfully found a proper past glacier state to match the RGI area within 1 km$^2$ or 1% of the total area, whatever is smaller.

5.  Initiate a dynamic simulation from 1980 to 2020, using the reference climate inputs and starting from the glacier state inferred in the previous step.

6.  Finally, the geodetic mass balance resulting from the dynamic simulation is calculated and compared it with the observed values from Hugonnet et al. (2021). If the difference between these values is within the defined uncertainty (±250 kg m$^{-2}$ yr$^{-1}$), the calibration/initialization workflow is terminated and the resulting glacier in 2020 and the parameters μ* and A

are used as inputs for the projection runs. If not, a new μ* is defined (Appendix A) and the whole process starts again from step Nº1).





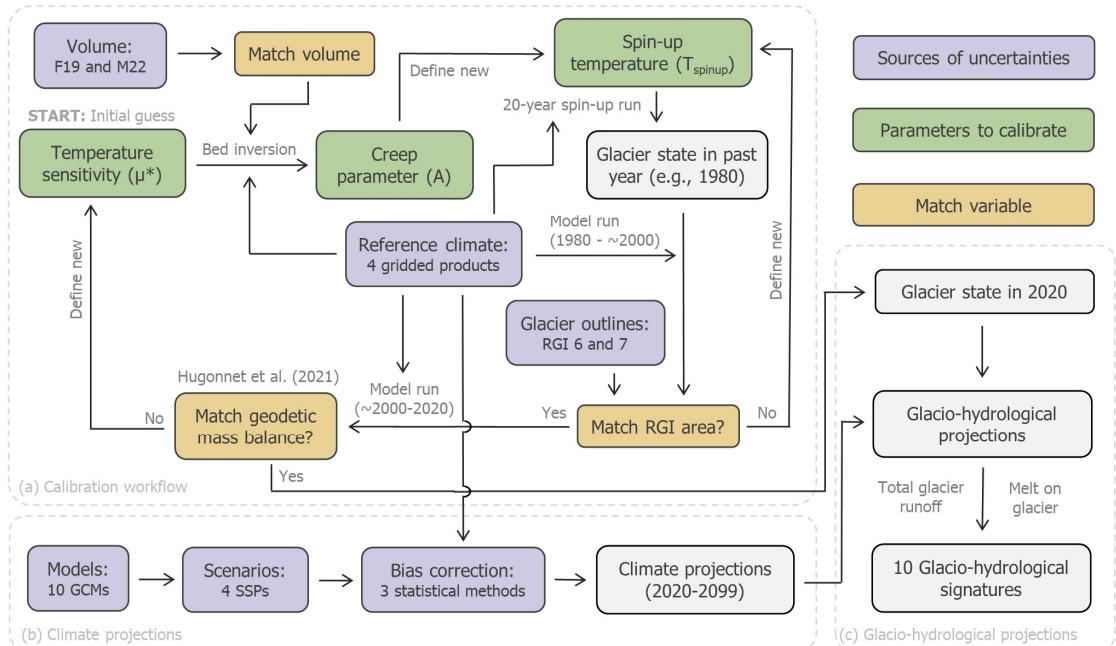

**Figure 2. Methodological framework. a) Open Global Glacier Model (OGGM) dynamic calibration workflow. b) Climate projections. c) Glacio-hydrological projections. GCM: General Circulation Models. SSP: Shared Socioeconomic Pathways. F19 and M22: Thickness estimated from Farinotti et al. (2019) and Millan et al. (2022). RGI: Randolph Glacier Inventory.**

### 3.2 Sources of uncertainty

#### 3.2.1 Geometry and volume

The geometry, represented by the glacier outlines, was obtained from RGI6 (Randolph Glacier Inventory - Version 6) (RGI Consortium, 2017) and RGI7 (RGI Consortium, 2023). In the latest version, RGI7 integrates the national inventories of Chile (Barcaza et al., 2017) and Argentina (Zalazar et al., 2020). Previous assessments of the complete RGI region (20–56º S) have shown that both datasets show similar latitudinal patterns in terms of area (-3% of total area according to national inventories; Zalazar et al., 2020). Nevertheless, the national inventories showed a higher number of glaciers ($\Delta n = 8,493$) and area ($\Delta = 651$ km$^2$) for the smallest glaciers (< 0.5 km$^2$), and a reduction of 7% in the area covered by the largest glaciers in the icefields compared to RGI6.

Individual volumes for each glacier were derived from the thickness estimated from Farinotti et al. (2019) and Millan et al. (2022) (hereafter F19 and M22, respectively). F19 is a consensus estimate from five models that use principles of ice flow dynamics to infer ice thickness from surface properties. In contrast, M22 uses glacier flow mapping to reconcile the spatial



distribution of ice masses with glacier dynamics, morphology, and ice divides. Hock et al. (2023) reported that the total ice volume of M22 showed a positive difference of 13% compared to F19 for the corresponding RGI region of the southern Andes.

Considering that both alternatives do not have a complete coverage of all glaciers in RGI6 (100% and 98.2% of the area for F19 and M22, respectively) and RGI7 (99.1% and 96.4% of the area for F19 and M22, respectively), we used a volume-area scaling (VAS, Hock et al., 2023) to complete the coverage and compare the datasets at a regional and catchment scales. In this approach, we calculated the VAS parameters for each hydrological zone (defined in Fig. 1) and dataset separately.

### 3.2.2 Reference historical climate (1980–2015)

We used monthly precipitation and air temperature time series from ERA5 (0.25°; Hersbach et al., 2020) and three gauge-corrected alternatives that use ERA5 in the bias correction process (CR2MET v2.5, MSWEP v2.8/MSWX and PMET v1.0). CR2MET v2.5 (0.05°; Boisier, 2023) is the current national reference for hydrometeorological studies in Chile, and is based on a statistical downscaling technique that uses ERA5, meteorological records, satellite land surface temperature and topographic descriptors. MSWEP v2.8 (0.1°; Beck et al., 2019) is a global precipitation product that merges gauges, satellites

and reanalysis data, and has outperformed other state-of-the-art precipitation products over Chile (Zambrano-Bigiarini, 2018). Precipitation from MSWEP v2.8 was complemented with air temperature from MSWX (0.1°; Beck et al., 2022), a bias-corrected meteorological product compatible with MSWEP. Finally, PMET v1.0 (0.05°; Aguayo et al., in review) was developed for Western Patagonia using statistical bias correction procedures, spatial regression models (random forest), and hydrological methods (Budyko framework) to correct the underestimation of precipitation reported in areas with pronounced

elevation gradients and significant snowfall. As part of the validation process, PMET outperformed ERA5, CR2MET and MSWEP in terms of hydrological modelling performance (Aguayo et al., in review).

### 3.2.3 Climate projections (2020–2099)

Climate projections of monthly precipitation and air temperature were obtained from 10 General Circulation Models (GCMs, Table S1) of the Coupled Model Intercomparison Project 6 (CMIP6; Eyring et al., 2016). Previous hydrological studies have

suggested that 10 GCMs can ensure that the median of all possible combinations produces similar uncertainty components as the entire ensemble (Wang et al., 2020). Following the recommendations of Hausfather et al. (2022), all selected GCMs show a transient climate response (TCR; temperature change at the time of $CO_2$ doubling) that lies in the "likely" range of 1.4 - 2.2 °C (66% likelihood, Table S1). Given that the warming constraints assessed in the Sixth Assessment Report (AR6) are correlated with the TCR (Tokarska et al., 2020), this provides a good approximation of the assessed warming. Note that all

GCMs have an equilibrium climate sensitivity (ECS) that falls in the "very likely" range, but only 80% of them fall in the 'likely' range of 2.5–4.0 °C (Table S1). Considering that future scenarios are the main source of uncertainty at the end of the century in the southern Andes (Marzeion et al., 2020), we used four different Shared Socioeconomic Pathways (SSPs; O'Neill





et al., 2016): SSP1-2.6, SSP2-4.5, SSP3-7.0 and SSP5-8.5. Each GCM was initially resampled to 1.0º using a bilinear filter, and only the standard model realization was considered (r1i1p1f1 in all cases).

### 3.2.4 Bias correction method

Three statistical bias correction methods were evaluated to assess their impact on the glacier projections. The objective of bias correction is to minimize the systematic error of the climate projections obtained from general circulation models (Section 3.2.3) using the reference climate used in the calibration process (Section 3.2.2). The selected methods were: Mean and Variance Scaling (MVA; Chen et al., 2011), Quantile Delta Mapping (QDM; Cannon et al., 2015) and Multivariate Bias Correction with N-dimensional probability density function transformation (MBCn; Cannon, 2018). The MVA approach was commonly used in GlacierMIP2, as it guarantees that the bias-corrected time series has the same mean and standard deviation (i.e., variance) as the reference time series in the reference period. QDM is a hybrid method that combines quantile-based delta change and bias correction methods. Thus, it not only preserves the quantile changes predicted by climate projections, but also corrects the biases of the modelled series with respect to those of the reference time series. Finally, MBCn is a multivariate bias correction that has the advantage of transforming all aspects of the reference multivariate distribution to the multivariate distribution simulated by the climate models. The bias correction parameters of all methods were calculated on a monthly basis to account for the seasonality of GCM biases. Following the protocol of the Inter-Sectoral Impact Model Intercomparison Project (ISIMIP3b; Lange, 2021), the reference period was 1980–2015 for all correction methods. Climate outputs based on the QDM and MBCn approaches were obtained using the *xclim* package v0.4 (Logan et al., 2022).

### 3.3 Analysis of sources of uncertainty

Taking all glaciers into account, each source of uncertainty was analysed to quantify the difference between the alternatives. For area and volume, we calculated the relative and absolute differences for each catchment and hydrological zone defined in Fig. 1. In addition, we compared the acquisition dates of the glacier geometries for both inventories. In both cases, the area and volume were aggregated according to the location of the glacier terminus. To assess the influence of the reference climate on the glacier mass balance, we calculated the solid precipitation and positive degree-day sum in addition to precipitation and temperature. To isolate the effect of the spatial resolution, temperature from ERA5 and MSWEP/MSWX was downscaled to 0.05º using the same lapse rate used by OGGM (-6.5 ºC km$^{-1}$). Similarly, solid precipitation and positive degree-day sum were calculated using the default parameters from OGGM. Specifically, we calculated and compared annual means for each variable, catchment, and product for the reference period (1980–2015) using only the glacierized grid cells.

The climate projections were another source of uncertainty. To assess the impact of the raw climate projections, we calculated the relative change between the reference period (1980–2015) and the future period (2070–2099) for each GCM and SSP. In addition, we calculated the model agreement of precipitation following Iturbide et al. (2021), who defined a high model




agreement when more than 80% of the GCMs agree on the sign of the change. Finally, to assess the individual impact of each climate uncertainty source, we calculated the standard deviation across different reference climates, GCMs, SSPs, and bias

correction methods. Specifically, we calculated the standard deviation based on the long-term annual mean of each variable, catchment, and alternative. Analogous to the reference climate, we calculated the annual mean for the future period (2070-2099) using only the glacierized grid cells.

### 3.4 Glacio-hydrological runs

We used the OGGM model to estimate the evolution over the period 1980–2099 of all glaciers with an area > 1 km$^2$ in the

Patagonian Andes (40–56° S). This corresponds to 2,034 and 1,837 glaciers that accumulate 99.0% and 98.5% of the total volume estimated by Millan et al. (2022) for RGI6 and RGI7, respectively. For each glacier, we evaluated 1920 potential scenarios generated by all combinations of glacier outlines (n = 2), volume datasets (n = 2), reference climates (n = 4), GCMs (n = 10), future scenarios (n = 4), and bias correction methods (n = 3) (Fig. 2). To estimate the glacier mass that is unsustainable under current conditions, we additionally run 16 scenarios based on historical conditions (2 · 2 · 4). For this, we run a simulation

for 80 years with a pseudo-random climate based on the climate (30 years) around the year 2000. For each scenario, we extracted the annual glacier area, volume, and specific mass balance of each modelled glacier. To assess the hydrological contribution, we additionally extracted glacier runoff which corresponds to all water originating from the initially glacierized area (i.e., here year 1980; Huss and Hock, 2018). In this approach, OGGM calculates the glacier runoff from the sum of on- and off-glacier melt and on- and off-glacier liquid precipitation. To avoid overestimating the influence of the climate

uncertainty (e.g., precipitation reductions in the climate projections), we also extracted the melt on the glacier from the glacier runoff (Fig. 2c). The time series were aggregated at the catchment and hydrological zone scales according to the location of the glacier terminus.

For each catchment and potential SSP-based scenario (n = 1920), glacier runoff and melt on glacier were characterized by 10 glacio-hydrological signatures (i.e., metrics) to describe the dynamic properties of the hydrological time series (Table 2). The

set of signatures was selected to cover the different categories proposed by Richter et al. (1996): magnitude, timing, frequency, duration, and rate of change. Poff et al. (1997) used these categories to characterize the hydrological regime and proposed that these components fully describe the streamflow characteristics that are important to the aquatic ecosystem. The set of selected signatures is presented in Table 2.



**Table 2. Glacio-hydrological signatures used to characterize glacier runoff and melt on glacier time series of each catchment. The regime characteristics corresponds to the initial categories proposed by Richter et al. (1996). The reference and future period correspond to 1980–2015 and 2070–2099, respectively.**

| Signature or metric | Regime characteristics | Description | Units |
|---|---|---|---|
| Reference magnitude | Magnitude | Annual mean value (runoff and melt) calculated from the reference period. The value was normalized by the catchment area. | mm yr$^{-1}$ |
| Peak water year | Timing | Following Huss and Hock (2018), the peak water year was calculated using an 11-year moving average. | date (year) |
| Peak water magnitude | Magnitude Timing | Maximum annual value in the peak water year. The value was normalized by the catchment area. | mm yr$^{-1}$ |
| Peak water duration | Duration Timing | Number of years in which the annual value is greater than 90% of the peak water magnitude | years |
| Inter-annual variability | Frequency | Standard deviation of the detrended and normalized time series. For the detrending, we used the same 11-year moving average. | mm yr$^{-1}$ |
| Seasonal contribution | Duration Magnitude | Percentage of annual runoff that occurs during the summer season (DJF). Value calculated from the reference period | % |
| Seasonal variability | Frequency | Standard deviation of the percentage of the annual runoff that occurs during the summer season (DJF). | % |
| Seasonal shift | Timing Rate of change | Absolute change in summer contribution between the reference and future periods | % |
| Long-term trend | Rate of change Timing | Indicator of the long-term decline after reaching the peak water. The indicator is defined as the slope between the peak water year and 30 years later. | % dec$^{-1}$ |
| Long-term change | Rate of change Magnitude | Relative change between the reference magnitude and the mean annual value obtained from the future period | % |

## 3.5 Uncertainty analysis

We build random forest (RF) regression models based on the six sources of uncertainty to predict the glacio-hydrological signatures (n = 10) of each catchment with at least one glacier (area > 1 km$^2$) in both inventories (n = 329). RF regression models generate predictions using an adaptation of Leo Breiman's random forest algorithm, a supervised machine learning method (Breiman, 2001; Svetnik et al., 2003). We used the permutation feature importance (PFI) to assess the influence of each source (in this case, categorical predictors). The PFI is defined as the decrease in a model score when a single feature or predictor value is randomly shuffled (Breiman, 2001). By breaking the association between the feature and the target (i.e., glacio-hydrological signatures), this procedure leads to a decrease in the model score, revealing the extent to which the model relies on the feature. This method has been successfully used as a sensitivity analysis tool in several studies (e.g., Ahn, 2020; Bennett et al., 2022; Schmidt et al., 2020). For each catchment and signature, the training set was selected to be 90% of the full dataset of scenarios, and the remaining 10% was used to measure the permutation importance. The importance of each





feature was represented as the percentage of the average change in the Root Mean Square Error (RMSE) over 30 experiments of shuffling one feature. For all RF models, we used 500 regression trees as an ensemble, with each tree having a minimum

leaf size of five. For each split, two variables were randomly selected as candidates. The complete procedure was performed using Scikit-learn v1.3.0 (Pedregosa et al., 2011).

## 4 Results

### 4.1 Analysis of sources of uncertainty

#### 4.1.1 Historical conditions (1980–2015)

The historical conditions involved in the calibration process considered the geometry obtained from the glacier inventories (RGI6 and RGI7), the volume obtained from ice thickness datasets (F19 or M22), and the reference climate dataset (PMET, CR2MET, ERA5 and MSWEP). The incorporation of national inventories in RGI7 resulted in important differences compared to RGI6 (Fig. 3). The total number of glaciers increased from 10,544 in RGI6 to 21,285 in RGI7. Relative to this, RGI6 showed a higher number of glaciers with an area greater than $1.0 \ km^2$, but RGI7 has considerably more smaller glaciers ($< 1.0 \ km^2$).

The total glacier area decreased by 4.0% in RGI7 ($\Delta = 1,024 \ km^2$), with important regional differences (Fig. 3a,b). The northern area between the Puelo and Aysen catchments (PPY and PCA) showed positive differences between 4% and 15% (Fig. 3b). In contrast, the area located south of the SPI (GCN and CDI) showed the maximum negative differences with values as low as -31% (Fig. 3b). There were also differences in the date of acquisition of the two inventories, which made a direct comparison difficult (Fig. 3d). While 84% of the glacier area in RGI6 have an acquisition date in the year 2000, only 10% of the glacier

area in RGI7 have an acquisition date in the same year (91% between 2000–2003, including both years).



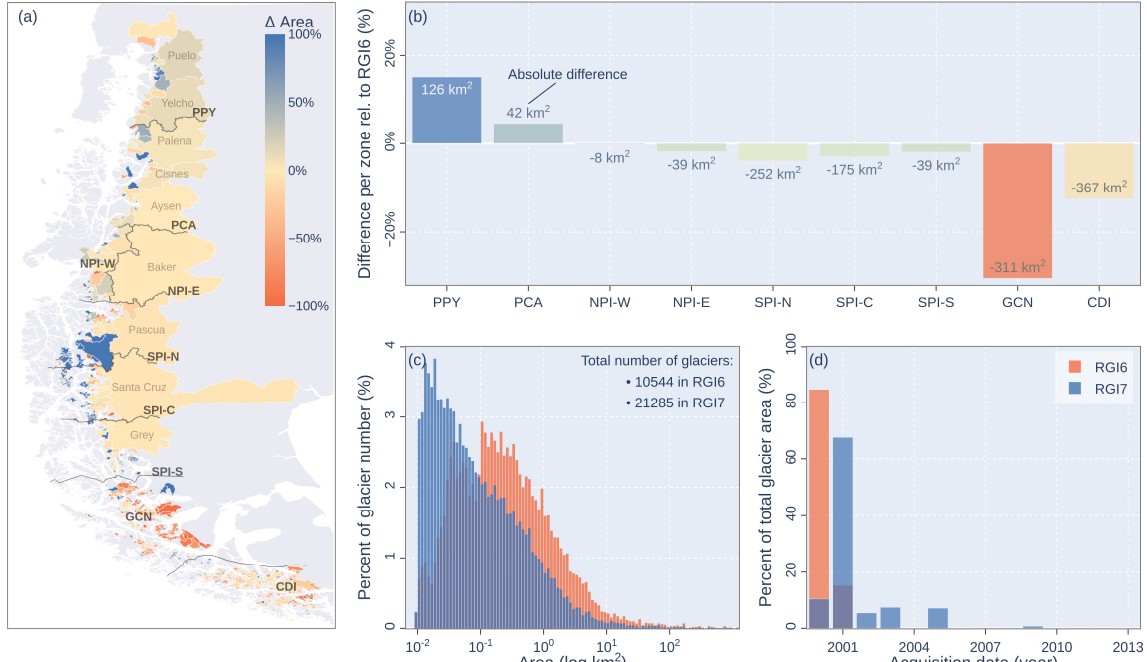

**Figure 3. Comparison between Randolph Glacier Inventory (RGI) versions 6 and 7. Difference in area a) per catchment and b) per hydrological zone considering RGI6 as reference. The area assigned to each catchment and zone was calculated according to the location of the glacier terminus. The names in grey in (a) correspond to the names of the main catchments, while the solid black line corresponds to the division between the hydrological zones defined in Fig. 1. The text in b) indicates the absolute difference in area (RGI7 – RGI6). c) Distribution of glacier area. d) Percent of glacier area per year of acquisition.**

Ice volume was another source of uncertainty analysed in this study (Fig. 4). According to the F19 dataset, the hydrological zones comprising the SPI have an ice volume of 3,526 km$^3$, representing 69% of the study area. Conversely, the PPY, PCA, GCN and CDI zones accounted for only 8.8% of the total ice volume. The 27% of the total catchment area had a normalized thickness (ice volume divided by catchment area) of less than 1.0 m (Fig. 4a). The M22 dataset showed more ice volume than the F19 dataset in 82% of the total catchment area (overall volume difference of 11%; Fig. 4b), mainly in the Patagonian Icefield (Fig. 4c). In this area, the NPI and SPI zones showed positive differences of 135 km$^3$ and 469 km$^3$ (relative to F19), respectively. Only the PCA and CDI zones showed the opposite change, where the M22 dataset shows a lower total ice volume (Fig. 4c).



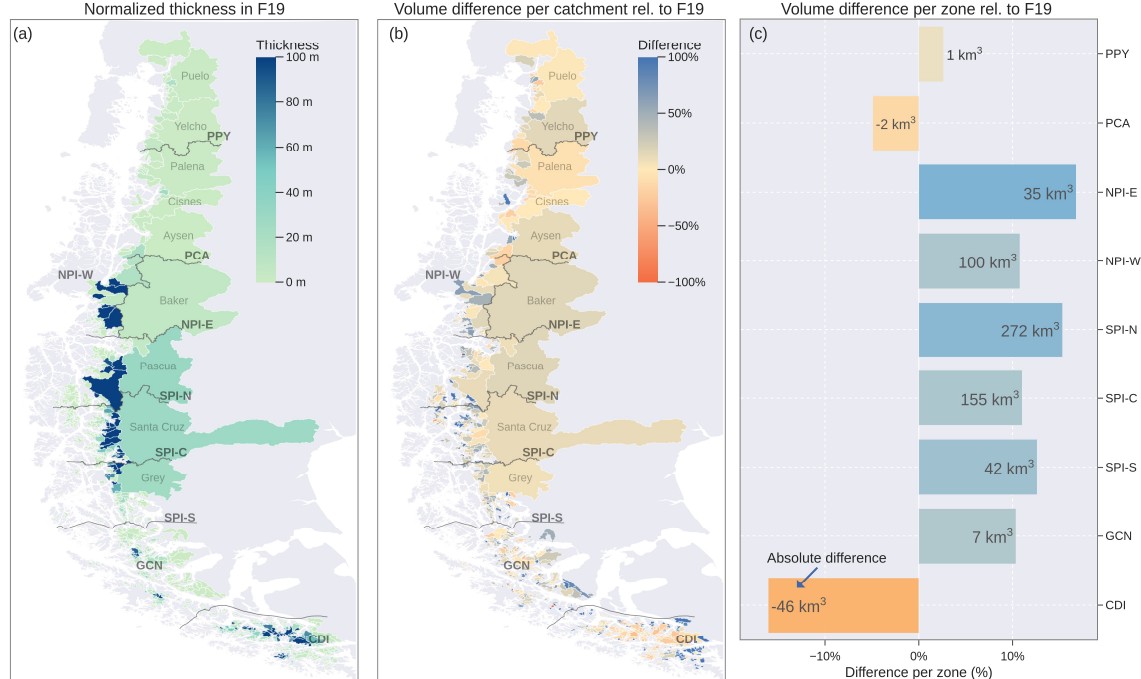

**Figure 4. Volume comparison between Millan et al. (2022) (M22) and Farinotti et al. (2019) (F19) based on RGI6. a) Thickness normalized by the catchment area from F19. Percentage difference between M22 and F19 per b) catchment and c) hydrological zone considering F19 as reference. The grey names in (a) and (b) correspond to the names of the main catchments, while the solid black line corresponds to the division between the hydrological zones defined in Fig. 1. The text in c) indicates the absolute difference in volume (M22 - F19). The volume assigned to each catchment and zone was calculated according to the location of the glacier terminus.**

The historical climate of the glaciers of the southern Andes showed an important climate diversity according to the PMET dataset, with an annual mean precipitation varying between 1,000 and 8,000 mm yr$^{-1}$ (Fig. 5a; 1980–2015). The spatial pattern of precipitation showed a clear difference between the western (> 4,000 mm yr$^{-1}$) and the eastern (< 2,000 mm yr$^{-1}$) side of the Andes (Fig. 5a). As a result of this orographic effect, mean precipitation was greater than 4,000 mm yr$^{-1}$ in 51% of the glacier area, which represents only 22% of the catchment area. 95% of the total glacier area (99% of the catchment area) showed an annual mean temperature above 0 ºC (Fig. 5b). The four climate products used to model the historical evolution of the glaciers (PMET, CR2MET, ERA5 and MSWEP/MSWX) showed important differences in precipitation and temperature (Fig. 5c-e). In relation to PMET, CR2MET and MSWEP showed a difference of nearly -50% in solid precipitation. ERA5 showed a similar pattern in solid precipitation with mean differences of -20%, but similar values in the main catchments (Fig.




5c). The mean temperature differences between the products were less than 1 ℃, which resulted in differences of less than 20% for the positive degree-day sum (Fig. 5d,e).

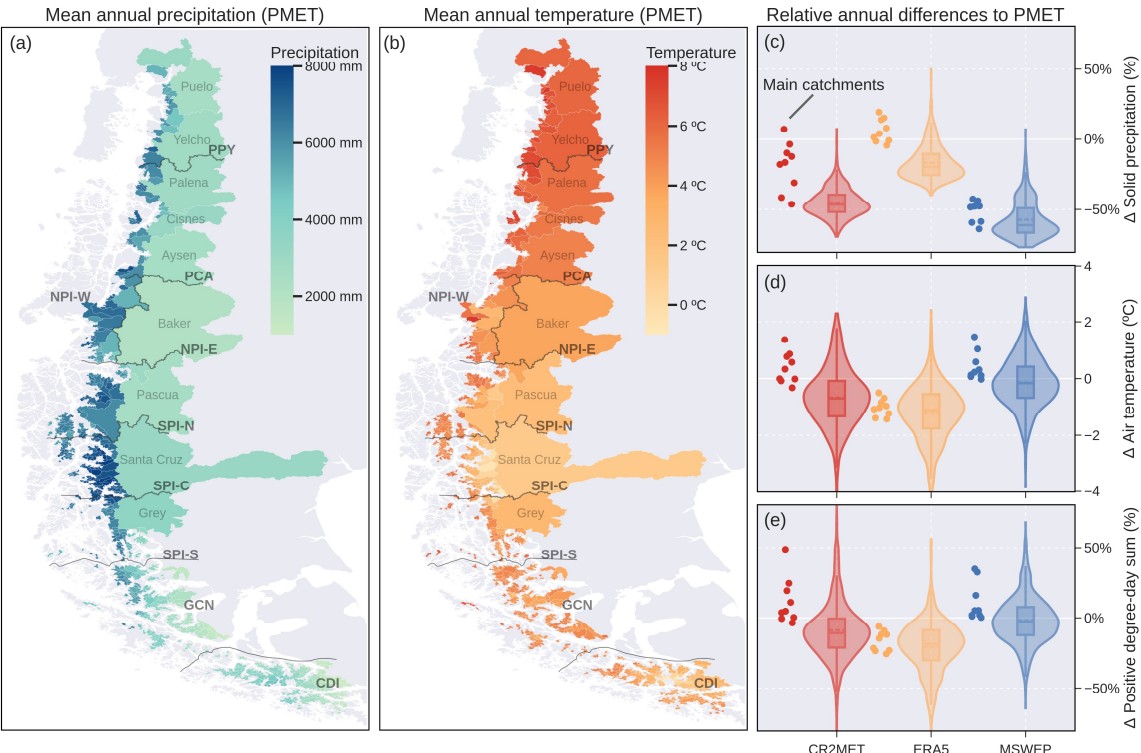

**Figure 5. Mean annual catchment precipitation (a) and temperature (b) according to the PMET dataset (1980-2015). Catchment means were calculated using only the glacierized grid cells. The grey names in (a) and (b) correspond to the names of the main catchments, while the solid black line corresponds to the division between the hydrological zones defined in Fig. 1. Relative catchment differences between PMET, CR2MET, ERA5 and MSWEP/MSWX for solid precipitation (c), temperature (d), and positive degree-day sum (e). The dots in (c-e) represent the values for the main catchments, which account for 68% of the total area. The dotted line**
**corresponds to the mean value.**

**4.1.2 Precipitation and temperature climate projections (2020–2099)**

The climate projections for the end of the century (2070–2099) showed clear latitudinal patterns (Fig. 6a, b). Overall, the northern area was characterized by a warmer and drier future climate, while the southern area showed a slight increase in precipitation accompanied by a slight increase in temperature. The GCMs showed a high model agreement in all zones (> 80%

of the models agree on the sign of the change), except in the SPI zones (Fig. 6a), areas characterized by a high ice volume (Fig. 4a). The climate projections for the catchments varied according to the SSP scenario.



Under the SSP1-2.6 scenario, the 67% of the catchment area (55% of the total glacier area) is projected to experience a decline in precipitation (Fig. 6c). This percentage increases to 91% under the SSP5-8.5 scenario. For temperature, the mean catchment warming varies from 0.96 ± 0.05 ºC in SSP1-2.6 to 2.46 ± 0.23 ºC in SSP5-8.5 (Fig. 6d). Overall, the main catchments showed

a higher warming than the average of all catchments (Fig. 6d).

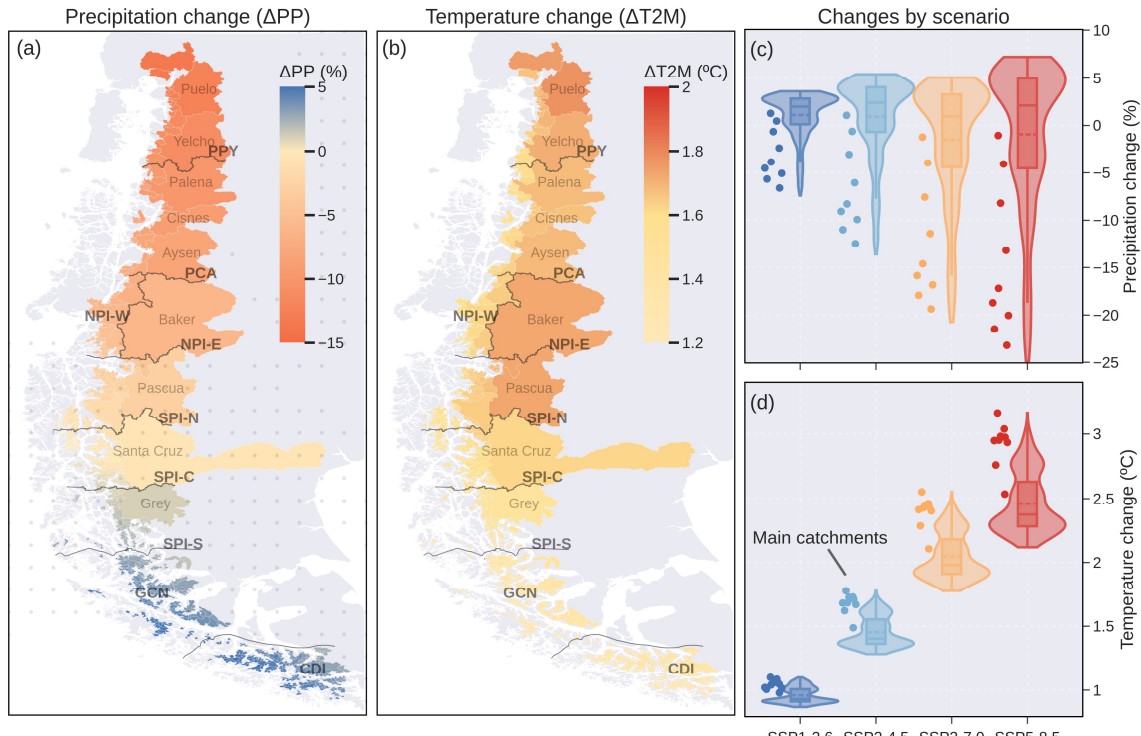

**Figure 6. Multi-model catchment means (n = 10) of precipitation (a) and temperature change (b) considering the SSP 2-4.5 scenario (1980–2015 *vs.* 2070–2099). Catchment means were calculated using only the glacierized grid cells. The dots in (a) indicate low model agreement where less than 80% of the models agree on the sign of the change. The grey names in (a) and (b) correspond to the names**
**of the main catchments, while the solid black line indicates the division between the hydrological zones defined in Fig. 1. Catchment differences by scenario for precipitation (c) and temperature (d). The dots in (c) and (d) represent the values for the main catchments, which account the 68% of the total area. The dotted line corresponds to the mean value.**

**4.1.3 Combined uncertainty of future climate**

Future climate uncertainty considered four reference climate products, ten General Circulation Models (GCMs), four different
Shared Socioeconomic Pathways (SSPs), and three bias correction methods (MVA, QDM, and MBCn), resulting in 480 possible combinations (Fig. 7). The standard deviation of the mean annual precipitation in the long term (2070–2099) was



greater than 1,000 mm in 40% of the catchment area, which represents 68% of the glacier area (Fig. 7a). Similarly, the standard deviation of the temperature was greater than 1.0 ℃ in 90% of the catchment area (Fig. 7b; 88% of glacier area). The precipitation showed a greater variability (expressed as coefficient of variation) in the catchments located on the western side

of the Southern Andes (Fig. 7a). On the other hand, the greater variability of temperature was concentrated in the SPI-C and CDI zones (Fig. 7b). For all variables, the reference climate was the most important source of uncertainty (Fig. 7c-e). The difference between SSPs and GCMs was more pronounced for temperature and positive degree-day sum (Fig. 7d,e) than for solid precipitation (Fig. 7c). The different bias correction methods (BCM) converged to similar values with no important differences between them.

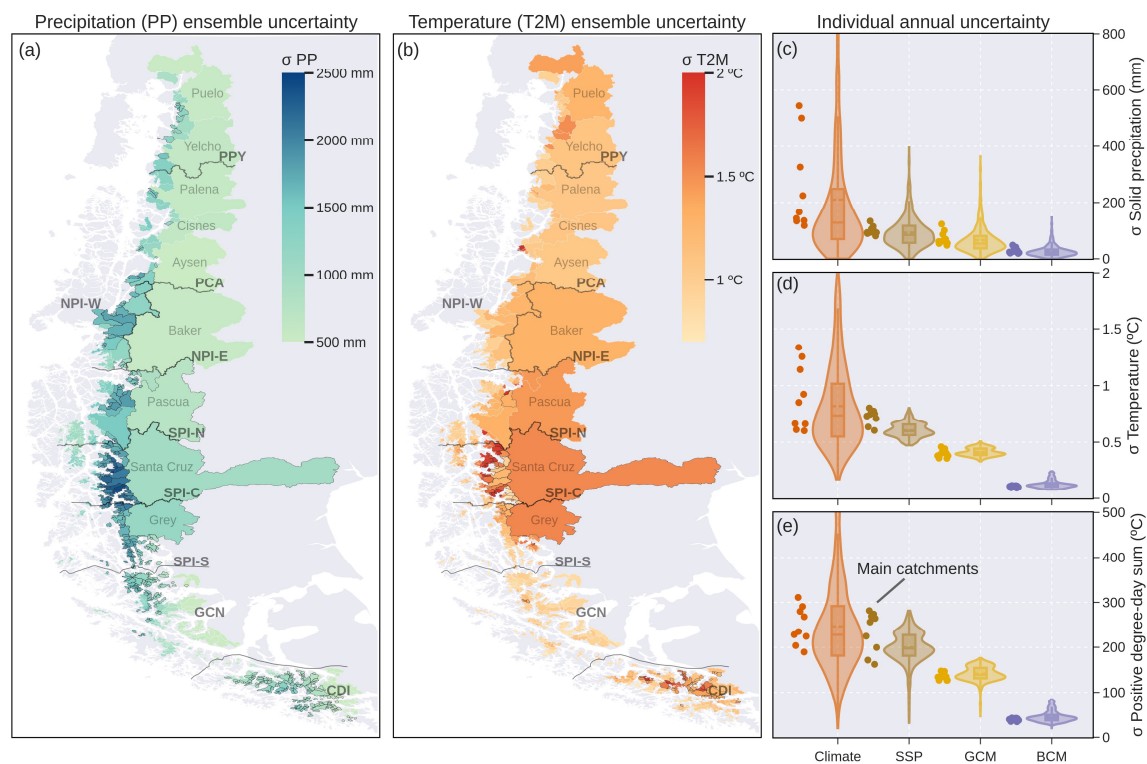


**Figure 7. Combined uncertainty of future climate.** Standard deviation (σ) of mean annual precipitation (a) and temperature (b) obtained from the complete ensemble (n = 480). The catchments with a coefficient of variation greater than 35 % have black outlines. Catchment means were calculated using only the glacierized grid cells. The grey names in (a) and (b) correspond to the names of the main catchments, while the solid black line indicates the division between the hydrological zones defined in Fig. 1. Standard deviation

of solid precipitation (c), temperature (d) and positive degree-day sum (e) across different climates, emission scenarios (SSPs), general circulation models (GCMs) and bias correction methods (BCM). In all cases, the standard deviation was calculated from the annual mean of the 2070–2099 period. The dotted line corresponds to the mean value.



**4.2 Glacio-hydrological projections**

Glacier projections from OGGM indicate a prolongation of the generalized mass loss of recent decades (Fig. 8). Considering the mean derived from the full set of SSP scenarios (n = 1920), 43% of the total catchment area (18% of glacier area) will lose more than 80% of their current (year 2015) volume by the end of the century (Fig. 8a). The results suggest that ice loss will vary according to different sources of uncertainty. Considering only historical sources (n = 16), 26% ± 9% of the total glacier ice is committed to melt in the long term (year 2099 in Fig. 8b). Aggregating the time series by emission scenario (n = 480 per

SSP), the volume loss varied from 48 ± 9% in SSP1-2.6 to 69 ± 10% in SSP5-8.5, with clear regional differences (Fig. 8c-f). In the northern region (PPY and PCA), the percentage loss will be higher than 70% regardless of the scenario (Fig. 8c). In NPI, SPI and the southern area (GCN and CDI), the percentage loss under the high emissions scenario (SSP 5-8.5) will be 65% ± 8%, 68% ± 12% and 72% ± 7%, respectively (Fig. 8d-f). At the hydrological zone scale (Fig. S1), the confidence intervals for volume and area in the reference period are consistent with the differences found between the glacier inventories

(Fig. 3) and ice thickness datasets (Fig. 4). The mean specific mass balance for the period 2077–2099 diverges strongly depending on the emission scenarios (Fig. S1). For example, the mean specific mass balance in NPI-E (Baker Basin) ranged from -483 ± 192 kg m$^{-2}$ in SSP1-2.6 to -1,705 ± 243 kg m$^{-2}$ in SSP 5-8.5.



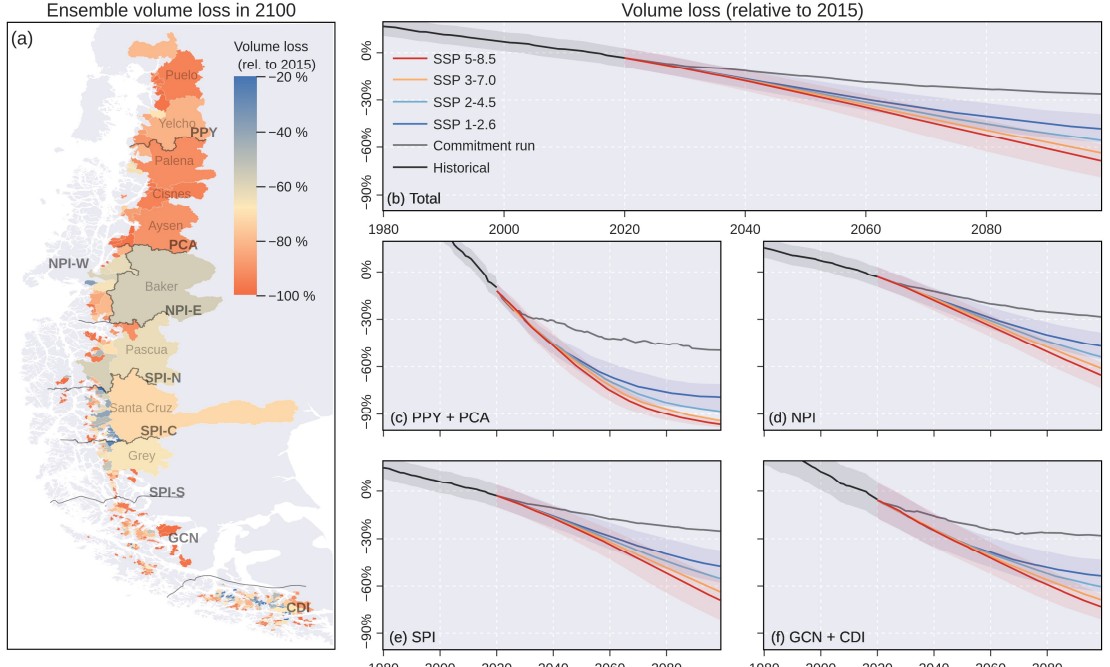

**Figure 8. Glacier volume loss relative to 2015. a) Mean volume loss in 2100 derived from the full ensemble (n = 1920). Volume loss for the sum of all catchments (b), the northern area including PPY and PCA (c), the Northern Patagonian Icefield (d), the Southern Patagonian Icefield (e) and the southern area including GCN and CDI (f). The solid line represents the mean for each scenario, while the uncertainty bands are calculated using one standard deviation (shown only for historical, SSP 1-2.6 and SSP 5-8.5 for visualization purposes). Volume, area and specific mass balance by hydrological zone are shown in Fig. S1. The commitment run considers a pseudo-random climate based on the period 1985–2015.**

The mass loss drives changes in the hydrological contribution of glaciers in the Patagonian Andes (Fig. 9). Considering the mean derived from the full set of SSP scenarios (n = 1920), 61% of the catchment area, which contains the 30% of the glacier area, has already peaked in terms of glacier melt (year 2020; Fig. 9a). The total melt on glacier in the reference period (1980–2015) was 2,055 + 199 $m^3 s^{-1}$ (Fig. 9b). For this total, the northern area (PPY and PYA), NPI, SPI and the southern area (CGN and CDI) contributed with 4.7%, 20.5%, 66.0% and 8.8% (Fig. 9c-f), respectively. The evolution of the melt on glacier varied slightly among emissions scenarios (n = 480 per SSP), and the projections and their uncertainties tended to converge towards the end of the century (Fig. 9b). For example, the mean melt on glacier in 2070–2099 varies from 1,555 ± 124 $m^3 s^{-1}$ in SSP 1-2.6 to 1,784 ± 165 $m^3 s^{-1}$ in SSP 5-8.5 for the whole region. While most hydrological zones are projected to experience a steady decrease in melt on glacier, the SPI zone shows slightly diverging trajectories in its mid-century meltwater contribution depending on the emission scenario (Fig. 9c-f). Relative to total runoff (Fig. S2), the uncertainty (i.e., standard deviation) in the melt on glacier evolution was lower due to a lower influence of the reference climate (Fig. 5), the main contributor of



climate uncertainty (Fig. 7). To the south of SPI, the slight increase in precipitation projections (Fig. 6) buffers the decrease in melt on glacier, maintaining the contribution of total glacier runoff. In all hydrological zones, the ratio between melt on glacier and the total glacier runoff is close to 60% in the reference period, percentage that decreases to 40% towards the end of the

century (Fig. S2).

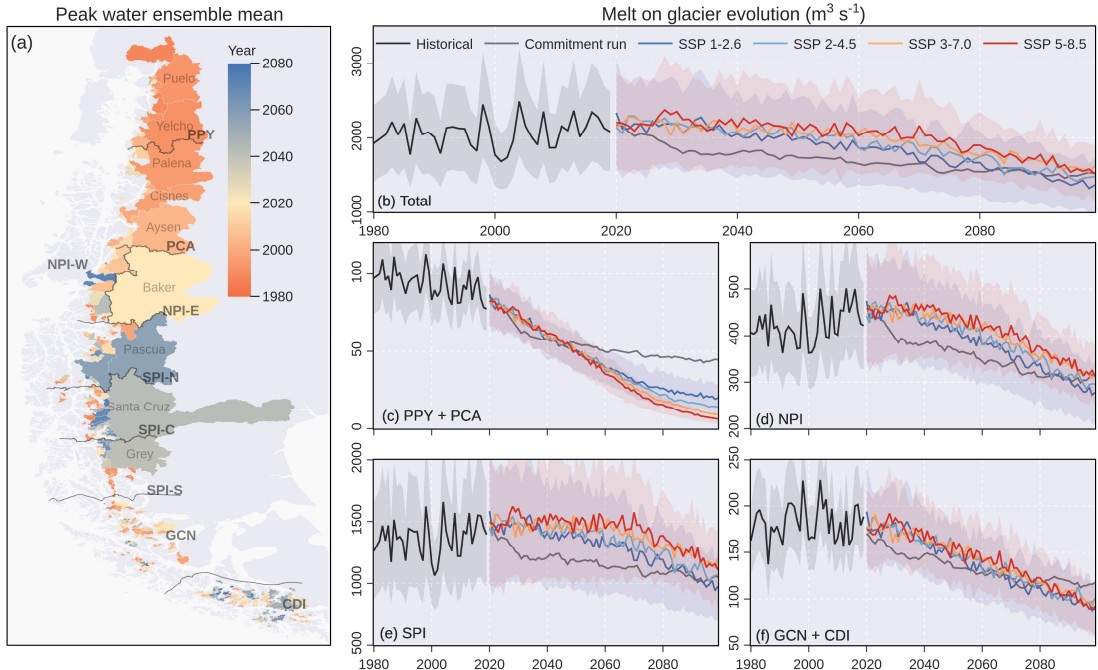

**Figure 9. Melt on glacier projections for the Patagonian Andes. a) Peak water year obtained from the complete ensemble (n = 1920). Melt on glacier evolution for the sum of all catchments (b), the northern area including PPY and PCA (c), the Northern Patagonian Icefield (d), the Southern Patagonian Icefield (e) and the southern area including GCN and CDI (f). The solid line represents the**

**mean for each scenario, while the uncertainty bands are calculated using one standard deviation (shown only for historical, SSP 1-2.6 and SSP 5-8.5 for visualization purposes). Total glacier runoff, melt on glacier, and the ratio between the two, disaggregated by hydrological zone, are shown in Fig. S2. The commitment run considers a pseudo-random climate based on the period 1985–2015. For visualization purposes, the commitment run was smoothed using a 10-year moving average.**

**4.3 Influence on glacio-hydrological signature**

The glacio-hydrological signature was represented by ten metrics that characterize the hydrological regime of each catchment (Table 2). Regardless of the variable (melt on glacier or total glacier runoff), the permutation feature importance (PFI) of RF models showed that the differences between the historical reference climates contributed most to the total uncertainty (Fig. 10). This was especially clear for the reference magnitude, peak water magnitude, inter-annual variability and seasonal




contribution, metrics where the reference climate accumulated more than 60% of the total RMSE loss after the permutations.
The accumulated RMSE loss of the historical sources of uncertainty (geometry, volume, and climate) was greater than that of
the future sources (GCM, SSP, and BCM) in six signatures, including the peak water metrics. In the long-term (trend and
change signatures), the historical sources accumulated a RMSE loss similar to that of the future sources. In these cases, the
selection of the reference climate or the GCM was as important as the emission scenario (SSP). Overall, the selection of the

glacier inventory was more important than the ice thickness dataset for most metrics. The importance of the bias correction
method (BCM) was significant (median > 10%) only for the seasonal variability and shift. Consistently, the RMSE loss of
future sources was 0% for all metrics calculated from the reference period (Table 2).

No clear spatial patterns were detected in the main source of uncertainty (i.e., the variable that accumulated most RMSE loss
in each catchment; Fig. S3). Considering the complete set of signatures calculated from the melt on glacier, the future sources

of uncertainty were the main source in only 18% ± 21% of the total catchment area, which contains 21% ± 23% of the glacier
area. In contrast, the reference climate was the main source of uncertainty in 78% ± 21% of the catchment area (74% ± 23%
of the glacier area). The lower importance of the climate was limited to a few metrics and hydrological zones. For example,
the GCMs were the main source of uncertainty in the Patagonian Icefields for the long-term trend, and in the northern area
(PPY and PCA) for the long-term change. In comparison to the total runoff metrics, the importance of the reference climate

decreases to 56% ± 33% of the total catchment area, corresponding to 57% ± 31% of the glacier area (not shown).





**Figure 10. Importance of each source of uncertainty for the glacio-hydrological signatures obtained from the total glacier runoff (dark colours) and the melt on glacier (light colours). The importance of each source on the glacio-hydrological signatures (Table 2) is represented as the percentage of the average change in the Root Mean Square Error (RMSE), with high values indicating a greater**

**relative importance for the signature. The dotted line corresponds to the mean value. Note that each panel has a different range.**



## 5 Discussion

### 5.1 Uncertainty in the modelling chain

For the first time, this study incorporated six different sources of uncertainty (both historical and future) into the glacio-hydrological modelling chain, resulting in a total of 1920 future evolution scenarios. The historical scenarios (n = 16)
considered the glacier inventory, the ice thickness dataset and the reference climate, while the future scenarios (n = 120) included different GCMs, SSPs, and bias correction methods (BCM). The exploratory analysis of both glacier inventories was consistent with Zalazar et al. (2020). While the northern area (PPY and PCA) showed an increase in glacier area in RGI7 relative to RGI6, the opposite was found in the southern zones (GCN and CDI) (Fig. 3). The decrease in glacier area in major ice masses may be due in part to differences in the acquisition dates (Fig. 3d), as only 10% of the glacier area in RGI7 has an
acquisition date in the year 2000. The relative differences in glacier area were smaller than the reported differences in glacier volume (Fig. 3 and 4). While the 69% of the total catchment area showed a relative difference of less than ± 10% between the glacier inventories, only 27% of the catchment area showed differences below this threshold when considering both volume (i.e., ice thickness) estimates. The use of observational data can help to select a better dataset for the study area, potentially reducing the overall uncertainty. However, direct or indirect observations of ice thickness, such as those based on ground-
penetrating radar or airborne surveys, are spatially and temporally scarce in the Patagonian Andes (e.g., Millan et al., 2019; Gacitúa et al., 2021). In addition, large-scale datasets require a large number of observational datasets such as ice velocity fields, surface topography, surface slope and area, which are derived from different acquisition dates (sometimes years or even decades), making an adequate assessment difficult (Hock et al., 2023).

The reference climate corresponds to one source of uncertainty that can be assessed using ground-based data. However, the
current scarcity, poor quality control protocols, and lack of continuity and reliability of meteorological stations is a very important limitation to properly understand the atmospheric processes at high elevations that have driven glacio-hydrological changes in the Patagonian Andes (Masiokas et al., 2020; Condom et al., 2020). Recent studies have attempted to narrow the ranges of uncertainty using, for example, regional estimates of moisture flux. For example, Sauter (2020) reported that the icefield-wide precipitation averages (period 2010–2016) are likely to be within in the range of 5,380 and 6,090 mm yr$^{-1}$ in the
NPI and 5,060 and 5,990 mm yr$^{-1}$ in the SPI. Based on the catchment hydrological balance, Aguayo et al. (in review) derived gridded precipitation factors ($P_f$ = 1.2 ± 0.6) to correct for the pronounced undercatch of mountainous areas. Using these factors, the PMET v1.0 showed long-term mean values of 6,050 mm yr$^{-1}$ and 5,900 mm yr$^{-1}$ for the NPI and SPI, respectively (Fig. 5a). Similar values of precipitation factors have been used to avoid underestimating mass accumulation (e.g., Temme et al. (2023) used $P_f$ = 1.2 in CDI).

The primary objective of glacio-hydrological modelling studies has been to assess the future impacts of climate change (Van Tiel et al., 2020), and therefore GCM and SSP have commonly been part of the uncertainty analysis. In the Patagonian Andes,



the climate projections (Fig. 6) showed clear spatial patterns that follow the observed trends of the last decades (Aguayo et al., 2021; Pabón-Caicedo et al., 2020; Boisier et al., 2018; Weidemann et al., 2018). Compared to the global scale, the region is projected to experience a lower degree of warming in the long term (~3 ºC in SSP5-8.5), compared to the global continental

warming of 5.9 ± 0.8 ºC under a high emissions scenario (1981–2010 *vs.* 2081–2100; Iturbide et al., 2021).The selected GCMs showed a high agreement in most hydrological zones, except in the SPI where less than 80% of the GCMs agreed on the sign of the change (Fig. 6). The GCM uncertainty was indirectly constrained by their selection. The ten selected GCMs showed a transient climate response (TCR) in the likely range of 1.4–2.2 °C (Table S1), which is a good approximation of the assessed warming (Hausfather et al., 2022). Future studies could potentially narrow the uncertainty based on the skill of GCMs to

reproduce historical trends (Collazo et al., 2022), and key drivers of change in the Southern Hemisphere atmospheric circulation, such as ozone depletion (Revell et al., 2022; Ivanciu et al., 2021).

There are several sources of uncertainty that were not considered in this study, such as downscaling strategies (e.g., temperature lapse rates), the geodetic mass balance, the use of frontal ablation parameterization schemes, the surface mass balance model (e.g., degree-day *vs.* energy balance), and the ice-flow model itself. These are acknowledged shortcomings of our study and

should be further investigated in future studies. For example, Schuster et al. (2023) using the OGGM model showed that the use of spatially and seasonally variable lapse rates has the most systematic influence on glacier projections with smaller glacier volumes by the end of the century compared to the constant option. Similarly, Bravo et al. (2019a) reported spatial patterns in lapse rates for the SPI that can change the ablation rates by up to 60%, depending on the extrapolation method applied. The geodetic mass balance used in the dynamic calibration process was another source of uncertainty. In comparison to Hugonnet

et al. (2021) who obtained a specific mass balance of -720 ± 70 kg m$^{-2}$ yr$^{-1}$ for the complete RGI region (2000–2019), Braun et al. (2019) and Dussaillant et al. (2019) recently estimated values of -640 ± 20 kg m$^{-2}$ yr$^{-1}$ (2000–2015) and -720 ± 220 kg m$^{-2}$ yr$^{-1}$ (2000–2018), respectively. The complex dynamics of marine-terminating and lake-terminating glaciers is also a source of uncertainty in projections of regional glacier loss. In the Patagonian icefields, Minowa et al. (2021) estimated that frontal ablation was −24.1 ± 1.7 Gt a$^{-1}$ (2000–2019), representing 34 ± 6% of total ablation. The study of calving glaciers adds a layer

of complexity, as additional processes require potential parameterizations and adjustments, which are also subject to uncertainty (Van Tiel et al., 2020). Using the OGGM model, Malles et al. (2023) found that the global mean sea level rise contribution at the end of this century is reduced by ~9% when marine frontal processes were considered in Northern Hemisphere glaciers. Surface mass balance models can also play an important role in glacier evolution, but the lack of calibration and validation data hinders the assessment of the added value of more model complexity (e.g., Temme et al., 2023;

Schuster et al., 2023; Huss and Hock, 2015).



**5.2 Changing glacier hydrology**

Glacier hydrology in the Patagonian Andes is expected to continue changing as a result of the sustained mass loss (Fig. 8 and 9). Considering the mean derived from the full set of SSP scenarios (n = 1920), 43% of the catchment area, which contains 18% of the glacier area, will lose more than 80% of its current volume (year 2015) this century (Fig. 8a). Aggregating the time series by emission scenario (n = 480 per SSP), the 21$^{st}$ century volume loss varied from 48 ± 9% in SSP1-2.6 to 69 ± 10% in SSP5-8.5 for the complete study domain (Fig. 8b). These mass loss projections are similar to the recent projections from Rounce et al. (2023) (Fig. S4). When comparing the mean results disaggregated by hydrological zone and SSP scenario, the RMSE of the projected mass loss was only 4.4% between the two studies. Despite differences in the sources of uncertainty considered in the two studies, the uncertainty (i.e., standard deviation) associated with mass loss was similar, suggesting that GCMs and SSPs are the main source of variability in future mass loss.

Compared to future mass loss, glacier melt showed a greater overall uncertainty, and the projections aggregated by SSP scenario tended to converge towards the end of the century (Fig. 9 and S2). Despite the dependence of the specific mass balance on the emission scenarios (Fig. S1), the melt on the glacier does not show a clear dependence, as it also depends on the glacier area, which decreases during the century (Fig. S1). This is shown, for example, by the regional melt on glacier, which only varies from 1,555 ± 124 m$^3$ s$^{-1}$ in SSP1-2.6 to 1,784 ± 165 m$^3$ s$^{-1}$ in SSP5-8.5 in the long-term (2070–2099; Fig. 9b). Only a few hydrological zones showed clear differences in their mid-century meltwater contribution between SSP scenarios, such as NPI-W and SPI-N. This is partly explained by the fact that 61% ± 14% of the catchment area, which contains 30% ± 13% of the glacier area, has already peaked in terms of glacier melt (year 2020; Fig. 9a). Compared to total runoff (Fig. S2), the uncertainty (i.e., standard deviation) in melt on glacier was reduced because the reference climate, which is the primary source of climate uncertainty (Fig. 7), had less influence, as the strongly past climate-dependent liquid precipitation is only included in the total runoff components. The slight increase in precipitation projections south of the SPI (Fig. 6) buffers the decrease in melt on glacier, maintaining the contribution of glacier runoff throughout the 21$^{st}$ century (Fig. S2).

Glacio-hydrological studies are scarce in the Patagonian Andes, but the Baker River Basin (NPI-E in Fig. 1; 7.2% of glacier area) has been a point of comparison due to its recent trends (Dussaillant et al., 2012) and historical GLOFs (Vandekerkhove et al., 2020). Using ASTER stereo images and stream gauges, Dussaillant et al. (2019) found a slight increase in the decadal glacier contribution to streamflow from 2% to 3% over the period 2001–2017. In a similar period (2000–2019), Caro et al., (2023) modelled a volume change of -10.7%, and a decadal increase in the mean annual glacier melt of 10% (66% of glacierized area modelled). In a longer time frame, Huss and Hock (2018) found that regardless of the emissions scenario, the peak water of the Baker River has already occurred or will occur in the coming years (2015 ± 18 and 2020 ± 16 for RCP 2.6 and RCP 8.5, respectively). In our study, we found that the peak water year based on total glacier runoff may have been reached



under all scenarios (2021 ± 15), which coincides with a slight increase in glacier runoff of 3% per decade over the reference period 1980–2015.

**5.3 Drivers of glacio-hydrological signature variability**

The ten metrics characterizing the hydrological regime of each catchment showed an important variability as a result of the
1920 scenarios generated from the six sources of uncertainty. The uncertainty analysis showed that the reference climate was the most important source in 78% ± 21% of the total catchment area (74% ± 23% of the glacier area; Fig. S3), and accumulated more than 60% of the total RMSE loss in the metrics relative to reference magnitude, peak water magnitude, inter-annual variability, and seasonal contribution (Fig. 10). Similar approaches have assessed the influence of the reference climate, but their focus has been limited to the mass loss evolution. For 18 glaciers in High Mountain Asia, Watanabe et al. (2019) showed
that the differences between observed past climate datasets (n = 6) introduced uncertainties of about 15% into projected changes in glacier mass. In Scandinavia and Iceland, Compagno et al. (2021) showed that the choice of the reference climate leads to differences of only 7% in the remaining ice volume by 2100. The small sensitivity was attributed to the model calibration scheme they used, which effectively reduces differences between climate forcing products by adapting precipitation and temperature corrections until they match glaciological observations. Our study, in turn, chooses not to correct the historical
climate dataset, and therefore the historical climate uncertainty is incorporated into the model calibration and then into the projections.

In addition to other historical sources of uncertainty, the individual effects of different inventories and volumes on glacier mass projections have also been assessed. Using two different glacier inventories for China, Li et al. (2022) showed volume differences of $30.4 \pm 2.5$ km$^3$ by the end of the 21$^{st}$ century, differences that are higher than between adjacent emission
scenarios, highlighting the importance of regional inventories. Using a set of ice-thickness measurements, Gabbi et al. (2012) analysed the sensitivity of glacier runoff projections to under- or overestimation of the total ice volume. The analysis revealed that reliable estimates of ice volume are essential, and that incorrect estimates could even lead to deviations in the sign of the projected runoff trend. In our study, the larger relative differences in area played a more important role than volume for most glacio-hydrological metrics (Fig. 10). However, the relative importance was masked by the choice of the climate in the
historical period.

In the long term, the relative importance of the emission scenario was not as important as in previous studies that did not consider the reference climate in the modelling chain uncertainty (Fig. 10). In the Southern Andes, the Glacier Model Intercomparison Project Phase 2 (GlacierMIP2) showed that the uncertainty in the emission scenario is the largest source for the specific mass balance rate (Marzeion et al., 2020). Similarly, Mackay et al. (2019) found that the emission scenarios were
also the dominant source for projections of mean monthly streamflows during the melt season, contributing up to 65% of the





total projection uncertainty. In our study, the accumulated RMSE loss of the historical sources (geometry, volume, and climate) was greater than that of the future sources (GCM, SSP, and BCM) in more than half of the signatures. Furthermore, the future sources of uncertainty were the main source in only 18% ± 21% of the total catchment area. The relatively greater influence of the future sources was limited to the long-term metrics (trend and change), where the selection of the emissions scenario or the GCM was as important as the reference climate (Fig. 10). This underscores that future glacio-hydrological projections are strongly shaped by modellers' choices, which should be guided by a systematic review of local datasets to adequately justify decisions in the modelling chain.

## 6 Conclusions

In this study, we investigated the importance of six sources of uncertainty in the modelling chain for ten glacio-hydrological signatures covering the necessary categories to characterize the glacio-hydrological regime of each catchment (magnitude, timing, frequency, duration, and rate of change). For this purpose, we used the Open Global Glacier Model (OGGM) to project the potential change in the hydrological contribution of each glacier (area > 1km$^2$; 2,034 glaciers in RGI6) in the Patagonian Andes (40–56° S) under 1920 potential scenarios. Based on these projections, we used the permutation importance of random forest regression models to calculate the relative importance of each source of uncertainty. Our main findings are as follows:

- The relative differences between the different alternatives varied across the six sources of uncertainty. The relative differences in area due to different glacier inventories were lower than the reported differences in volume. While 69% of the total catchment area showed relative differences between glacier inventories of less than ± 10%, only 27% of the catchment area showed differences below this threshold when different volume estimates were considered. Among all contributors to future climate uncertainty (2070–2099), the reference climate was the most important source, followed by the SSP, the GCMs, and finally the bias correction method used.

- The hydrological patterns of glaciers in the Patagonian Andes are anticipated to undergo further changes due to the ongoing mass loss and the projected climate trends. By 2020, 61% ± 14% of the total catchment area (30% ± 13% of the glacier area) has already peaked in terms of glacier melt. In addition, 43% ± 8% of the catchment area (18% ± 7% of the glacier area) is projected to lose more than 80% of its glacier volume over the course of the century. Aggregating the time series by emission scenario, the projected volume loss varies from 48 ± 9% in SSP1-2.6 to 69 ± 10% in SSP5-8.5. Compared to future mass loss, glacier melt and runoff showed a greater overall uncertainty, and the projections aggregated by SSP scenario tended to converge towards an overall decrease throughout the 21st century, with particular exceptions (e.g., SPI).

- For more than half of the glacio-hydrological signatures obtained from the glacier melt evolution, the uncertainty from historical sources exceeded that from future sources, underscoring the critical role of modeler decisions during the calibration





process. In the Patagonian Andes (40–56º S), the primary source of uncertainty considering all metrics was the reference historical climate in 78% ± 21% of the catchment area (74% ± 23% of the glacier area). For long-term metrics (trend and change over the period 2070–2099), factors not typically considered in regional studies, such as the selection of GCMs and reference climates, were as important as emission scenarios.

Our results shed light on the evolution of glacier runoff in the Patagonian Andes and provide new insights into local calibration
choices. To our knowledge, the present study is the first large-scale assessment of the impact of multiple sources of uncertainty (both historical and future) from a perspective beyond future glacier mass loss. In order to advance with climate change adaptation plans for the long-term sustainability of local ecosystems, future studies should address sources of uncertainty not considered in this study (e.g., parameterization of frontal ablation, climate downscaling and surface mass balance and ice-flow models), and the relative contribution of non-glacial water sources (Drenkhan et al., 2022; He et al., 2021). The latter will
improve our understanding of the potential relative contribution of glaciers (e.g., Kaser et al., 2010), and the fluxes between the different water stores. The inclusion of surface and subsurface water stores, such as example snowpack, lakes and groundwater, can play a crucial role in the seasonal and interannual water release during dry seasons (e.g., Pokhrel et al., 2021), and thereby buffering glacier shrinkage consequences (Somers et al., 2019; Mackay et al., 2020). Finally, we hope that these new antecedents help to prioritize future efforts (e.g., reference climate) to reduce glacio-hydrological modelling gaps in the
Patagonian Andes.

**Appendix A: dynamic calibration minimization algorithm**

At the beginning, a first guess of the control variable ($T_{spinup}$ or $\mu^*$) is used and evaluated. If the mismatch between model and observation happens to be close enough, the algorithm stops. Otherwise, the second guess depends on the calculated first guess mismatch. For example, if the first resulting area is smaller (larger) than the searched one, the second temperature bias will be
colder (warmer). This is because a colder (warmer) temperature leads to a larger (smaller) initial glacier state. The same idea is used for matching the geodetic mass balance. If the second guess is still unsuccessful, the previous value pairs (control variable, mismatch) are used for all subsequent guesses to determine the next guess. This is done by fitting a stepwise linear function to these pairs and then setting the mismatch to 0 to obtain the next guess (this method is similar to the one described in Appendix A of Zekollari et al. (2019)).

**Code availability**

The complete code repository can be found at: https://github.com/rodaguayo/glacier_uncertainties



**Data availability**

The glacier outlines from RGI6 and RGI7 were downloaded from https://nsidc.org/data/nsidc-0770/versions/6 (last access: September 28, 2023) and https://nsidc.org/data/nsidc-0770/versions/7 (last access: September 28, 2023), respectively.

NASADEM was downloaded from: https://lpdaac.usgs.gov/products/nasadem_hgtv001 (last access: September 28, 2023). Ice thicknesses datasets from Millan et al. (2022) and Farinotti et al. (2019) were downloaded from https://doi.org/10.6096/1007 (last access: September 28, 2023) and https://doi.org/10.3929/ethz-b-000315707 (last access: September 28, 2023), respectively. PMET v1.0 was downloaded from https://doi.org/10.5281/zenodo.7992761 (last access: September 28, 2023). CR2MET v2.5 was downloaded from https://doi.org/10.5281/zenodo.7529682 (last access: September 28, 2023). ERA5 was

downloaded from https://cds.climate.copernicus.eu/#!/home (last access: September 28, 2023). MSWEP v2.8 was downloaded from https://www.gloh2o.org/mswep/ (last access: September 28, 2023). CMIP6 data was downloaded from the Google cloud storage provided by the Pangeo initiative (https://storage.googleapis.com/cmip6/pangeo-cmip6.json, last access: September 28, 2023). The complete set of results provided in this study is available at: https://doi.org/10.5281/zenodo.8384883

**Author contributions**

This study was conceived and designed by RA, FM and LS. RA collected the data and performed the modelling and core data analysis. PS implemented and developed the dynamic calibration algorithm. MS, AC, JL, MA, JM, and LU contributed to the analysis and discussion of the results and drafting of the manuscript. All authors revised the manuscript, provided feedback, and contributed to the preparation of the figures and tables. All authors approved the final version of the manuscript.

**Acknowledgements**

RA would like to acknowledge the support of the Open Global Glacier Model (OGGM) community for their valuable assistance, collaboration, and commitment to open science. JM publishes with the permission of the executive director, British Geological Survey (UKRI).

**Financial support**

RA was supported by the National Agency for Research and Development (ANID) PFCHA/DOCTORADO
NACIONAL/2019 – 21190544. LS's contribution was funded by her DOC Fellowship of the Austrian Academy of Sciences at the Department of Atmospheric and Cryospheric Sciences, University of Innsbruck (No. 25928). LS's, PS's and FM's contributions were funded from the European Union's Horizon 2020 research and innovation programme under grant agreement No. 101003687. This text reflects only the author's view, and that the Agency is not responsible for any use that





may be made of the information it contains. JM's contribution was funded by the Natural Environment Research Council

(NERC) MCNC Grant TerraFirma NE/W004895/1.

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
