# Peer review of "Unravelling the sources of uncertainty in glacier runoff projections in the Patagonian Andes ( $40{\text -}56^{\circ}$ S)"

_EGUsphere, 2023_

## Referee Comment (RC2)

Review of "**Assessing the glacier projection uncertainties in the Patagonian Andes (40–56°S) from a catchment perspective**" by *Aguayo et al*.

In this manuscript the authors quantify the effects of historical and future glacial and climatic conditions on simulated glacier melt in the Patagonian Andes using the Open Global Glacier Model (OGGM) and Random Forest Regression approach. They focus on six categories, or sources of uncertainty, namely glacier outline inventories, ice thickness, historical climate, GCMs, bias correction methods, and emission scenarios. They examine the importance of each source on ten runoff metrics (e.g., peak water year and magnitude, interannual variability, seasonal contribution and variability, etc.) and conclude that the choice of reference climate is the predominant source of uncertainty.

The authors have undertaken a fairly comprehensive assessment, exploring 1920 cases across the six categories, to identify how the choice of input and forcing data affects the outcome in a glacial-hydrological modeling workflow. This is overall a well-written paper, and the figures support the narrative well. The authors have demonstrated the methods clearly and the results are presented in a logical way.

The areas which require considerable improvement are the Discussion and Results sections which lack coherency and do not tie the various pieces together. There are differences across the hydrological zones/catchments and the ten glacier runoff metrics which require more nuanced discussion. Further, the authors need to explicitly highlight the implication/s of this work and explain if/how the findings here can be used for other glaciated regions of the world (i.e., does this study domain encompass a comprehensive set of climatic, glacial, and hydrological conditions to make general deductions).

I recommend moderate revisions before the manuscript is ready for acceptance. These revisions mostly require either clarification or further elaboration in the text. Comments below are provided section-wise and are not in order of importance.

**Introduction**

Ln 61-62: Can you clarify the statements related to "*significantly increased streamflow*" and "*significant trends*". I presume the latter refers to statistical significance and the former to a large magnitude increase? Also linking with the previous statement, is it possible to extract information from these past work on how much (%) the streamflow has increased because of the accelerated glacial mass loss?

Ln 67-68: Is this referring to overestimation of precipitation? Also, what does "*diverged towards*" mean?

Ln 70 onward: I am somewhat unsure about the purpose of this table (and generally this paragraph). The table summarizes past mass balance assessments; however, the current study is not considering various MB schemes or calibration processes as a source of uncertainty. This table seems superfluous and if the authors want to keep it, please consider moving to the supplementary material. Instead, please expand on the two Patagonian glacier hydrological contribution studies (*Mernild 2017* and *Caro 2023*). That will link better with the preceding paragraph and the overall theme of the paper.

Ln 89: "… *adding additional data to the calibration*…" What type of additional data is this referring to?

Ln 91: Please explicitly mention the six sources of uncertainty here, bringing information in Ln 94 – 96 earlier, and then mention the tools used (OGGM, random forest).

Ln 91 onward: The preceding paragraph talks about different sources of uncertainty in the modeling chain in current literature (beyond what the authors have explored in this work). Various statements in this study imply that it focuses on comprehensive sources of uncertainties across the modeling chain. In reality, it only considers two sources i.e., model input data (glacier outlines, thickness) and atmospheric forcing (historical data, GCMs, climate scenarios, and bias-correction methods). I would not consider this encompassing "*uncertainty sources in the modeling chain*" (e.g., Section 5.1 heading). The full chain will include considerations such as model workflow, parameter space, MB or ice flow schemes, etc. (this is duly mentioned as a study limitation in Ln 517).

I recommend removing the term "glacier modeling chain" here and in other instances, including the abstract, and explicitly mention that various configurations of model input and forcing data are used to identify the dominant sources of uncertainty. This is an extensive assessment with these two broad categories as it is, so there is no need to overstate the study's scope.

**Study area**

Ln 103: What is meant by pristine environment here? Is this referring to glaciers or the water resources?

Ln 111: These nine hydrological zones are referred to extensively throughout the paper. Please fully name all of them here.

Ln 111: I understand how the nine hydrological zones were identified and demarcated, but can you please elaborate how the 847 catchments were selected. Was having a 0.1% glacier area of the total catchment area the only criteria? Why was this threshold selected?

Ln 113: Can you please explain "*high explanatory power of recent glacier change*" or reword this statement.

Ln 123: "*are characterized by many small catchments*". Replace characterize by 'which hosts many small catchments' or something similar.

**Methods**

Ln 142: Please mention the upper and lower thresholds here in brackets.

Ln 146: OGGM is a flowline model so what is meant by "local grid" of 10 – 200m? Is it referring to the spacing between cross-sections across the centerline?

Ln 151: If the precipitation factor is set a single value of 1, then it is no longer accounting for biases due to topography, missing processes etc. (Ln 137). Also, what does it mean to "… *assess influence of different reference climates …*" Is this referring to the temperature sensitivity parameter as the precipitation factor is set to 1 for all cases.

Ln 163: Replace "*new value*" with initial value or first guess.

Ln 172: Is the simulated glacier volume set to match the input volume accumulated over each of the nine hydrological zones and not the individual 847 sub-catchments?

Ln 187: Remove "it" after compared.

Ln 191: This might be a typographical error, what is N°1?

201: What does "*latitudinal patterns in terms of area*" mean?

Ln 209: It helps to keep the language simple. Can this be rephrased as M22 having 13% larger volume than F19… or something along those lines.

Ln 210: "Both alternatives" - consider replacing with the 'two volume data sources' or something similar.

Ln 213: By dataset do you mean the 4 gridded products?

Ln 213: VAS was done for the 9 hydrological zones; can you explain here why not for the 847 individual catchments?

Ln 228: Can you please elaborate on why/how these specific GCMs were selected? Was this a subset from a larger initial pool and the 10 GCMs were selected based on their TCR and ECS?

Ln 234 – 236: This is rather confusing: all GCMs have ECS falling in "very likely" range but only 80% in "likely"? Is the very likely range broader than the likely range? The "likely' range in *Hausfather22* was narrower (1.4 - 2.2 C).

Ln 257: It is somewhat unclear where, when, and why catchments and hydrological zones are considered separately, e.g., volume-area scaling was done at the hydrological zone scale, glacio-hydrological signatures were assessed at catchment scale (Ln 288), etc.

Ln 259: Not sure I understand what is meant by area/volume aggregation based on terminus location.

Ln 261: This lapse rate was used because it is default in OGGM and also used within this domain in the past studies (Ln 141). Table 1 however also shows that literature used different lapse rates depending on the region, e.g., 6.5 for NPI and 5.8 for GCN. Is there a north-to-south gradient in the mean annual lapse rates? Does that affect the downscaling of ERA5 data? Also, is this downscaling step the same as what is mentioned in Ln 141? What about total precipitation, how was that handled when going from a quarter degree to 0.05˚ in ERA5?

263: Please mention these default parameters either here or in supplementary material. I suppose Ln 153-154 mentions a couple of these, but please enlist all the parameters for reproducibility. Model "defaults" can change over time as new information becomes available.

Ln 264, 273, 361, etc.: "*Glacierized grid cells*" – to confirm, these grid cell-based computations are only done for input & forcing data because OGGM is a flowline model and the output information (area, volume, melt) will be for a specific glacier?

Ln 275: To clarify, when uncertainty metrics are computed all glaciers are considered (Ln 256), but for hydrological assessments only this subset is considered? Section 4.1 is for all the glaciers in RGI and Section 4.2 is for these ~2000 glaciers?

Ln 278 – 279: Up till this point, I do not think historical (outline, volume, ref. climate) vs future (scenarios, GCMs, BCM) sources are explicitly mentioned and separated in the text. So, the "*2 . 2 . 4*" reference is not clear.

Ln 283-284: I understand on- and off-glacier liquid precipitation, and "on-glacier" melt. But what is off-glacier melt? Is it referring to snow on non-glaciated areas? Also, the term on-glacier melt is unusual, I believe this is referring to direct melt from glacier?

Ln 284: It seems this line and onward is now referring to the full 1920 scenarios and not the 16 historical ones? How is climate uncertainty influence being overestimated? What is meant by 'climate' here, and what is precipitation reduction?

I suppose "glacier runoff" here is the sum of glacial melt and liquid precipitation on glacier? Please consider rephrasing the 'melt on glacier' with an equivalent term commonly used in literature.

Ln 286: Is this referring to the melt and precipitation time series? Also, I do not understand the meaning of 'according to glacier terminus' here.

Ln 288: Now this assessment is at catchment scale and not hydrological zone scale, again it gets fuzzy where hydrological zones vs catchments are considered. Also, aren't SSP-based scenarios 4, and n=1920 the total number of scenarios?

Ln 321-322: Can you please make the statements explicit here, e.g., RGI7 has 4% and 15% greater area than RGI6. The "*showed positive/negative difference*" is perhaps not the best way to state this.

Ln 325: There is only a 1–year difference in the acquisition date for most glaciers from RGI6 to RGI7 (Fig. 3c) – did that make such a large difference in terms of area reduction?

Ln 335: Just for clarification, both M22 and F19 have the same glacier outlines taken from RGIv6?

Ln 336: Fig. 4b is the M22 minus F19, correct? Most of the area is in yellow shades visually so M22 shows less volume. Perhaps it is the colorbar that needs to be changed.

Ln 351: Replace "in" with "over 51% of the glaciated area".

Ln 352: Throughout the paper, the reference to glacier/glaciated area and catchment area together is very confusing, for example this line mentions 51% of glacier area, 22% of the catchment area for the precipitation reference and 95% of the glacier area, 99% of the catchment area for temperature reference. Please consider presenting this information in some other way (perhaps focus on one of these only).

Ln 356: Similar patterns in terms of spatial patterns?

Ln 356: The spatial resolution of the native data is quite different (0.25 for ERA5, 0.05 for PMET/CR2MET), does that come into play at all?

Ln 367: Also, are there other characteristics that create these latitudinal differences in (a) precipitation change sign and (b) model agreement/disagreement?

Ln 370: Why/how is ice volume relevant to climate projections? This is a rather obtuse statement, talking about GCM model disagreement and then ice volume estimates.

Ln 374: What do you mean by the "main catchments".

Ln 405: What do you mean by the prolongation of the mass loss? Is this referring to Fig. 8b?

Ln 406: Again, please reconsider the discussion related to catchment and glacier area. It is surprising that 18% of glacier area is equivalent to 43% of the catchment area. Also, how is catchment losing volume (it is only the glaciers that will be losing the volume).

Ln 417: Is this the maximum range across all the basins? It seems some basins have less or more spread across the scenarios (e.g., SPI-N). Again, can you comment on the difference between the hydrological zones here or in discussion.

Ln 427: 61% of the total catchment area contains 30% of the total glacier area i.e., the remaining 70% of the glaciated area in the ~40% of the catchment area? Here it is best to talk about glacier area (because it is talking about glacial melt), catchment area reference is somewhat misleading.

Ln 428: Is this melt now talking about all the glaciated area in the full study domain?

Ln 430: Please consider rephrasing "the evolution of the melt on glacier…". Do you mean changes in melt rates?

Ln 435: This is referring to Fig. 9e for SPI, correct?

Ln 436: Which panel of Fig. 5 is this referring to? Please rephrase "*melt on glacier evolution*".

Ln 455: Remove comma after 'contribution'.

Ln 468: What is meant by lower importance of climate here? Reference climate?

Ln 470: Can you explain why there are differences between seasonal contribution, variability and shift in Fig. 10? What are the mechanisms causing these differences in sources of uncertainty.

**Discussion**

Overall, the discussion section needs some attention from the authors. This is not a study that can be easily replicated for other domains in terms of time commitment and computing resources needed. That makes it important to highlight the big picture findings, i.e., statements that provide insights on how to interpret the results for other regions (Are the findings applicable globally? If yes, then how was the conclusion reached? If not, then what are the differences or other cases that might need to be considered?).

Also, for this specific domain, the authors should discuss the differences across the various hydrological zones e.g., how does domain characteristics in terms of climate, topography, glacier size, etc. affect the six sources of uncertainty and the ten runoff metrics?

Ln 484-485: This statement implies that the difference in acquisition year played a significant role in glacier area, while in Fig. 3d there is only a 1-year difference between the two RGI versions. Are you saying there was a large area loss between 2000 and 2001?

Ln 486 – 488: The sentence from "*While the 69% of the total catchment ... estimates*" is unclear, please clarify and rephrase it. Also in Ln 488, what observational data are you referring to (glacier outlines or climate)?

Ln 491 – 493: Not sure what this sentence is getting at.

Ln 537: What is meant by "*continue changing*"?

Ln 539: Just to clarify, 43% of the catchment will lose 80% of their glacial volume? Also, there is latitudinal dependence (North-South divide). Can you talk more about that?

Ln 545: How was this conclusion reached? Is this talking about *Rounce et al. (2023)* or the current study?

Ln 565-566: Statement regarding peak water already reached – can this be supported with observational data for the region (at least that can help understand the historical metrics between 1980 – 2023, if the data is available)?

**Conclusions**

Much of this reads like results or discussion section. For example, the second bullet can be moved to results.

Ln 629: This study did not provide insights into "*local calibration choices*". As the authors rightly point out in the following sentence, future studies need to look into MB schemes, parameterizations, etc. This study is fairly comprehensive as it is, so the authors should try to highlight the significance of the work done and how it can be interpreted for other domains.

**Figures**

Fig. 1 and throughout: Can you please make the delineation between the 9 hydrological zones more prominent (e.g., in thick black line). The zone labels in 1b and c are not visible.

Also, it is better not to use sequential colormap for discrete categories. For example, it is hard to see the difference between <500 and <1500 in 1c. Readers should be able to extract this information quickly.

Fig. 2: This flowchart is hard to follow, can you provide step numbers along the way (preferably in same sequence as the text from Ln 162).

Fig. 3: Please consider using a more distinct diverging colormap for 3a, with white in the middle. It seems most of the regions are around yellow, so there is no (visible) difference between RGI 6 and 7. Also the bar colors in panel b are barely visible. Perhaps remove the grey background and make all the colors in darker shades.

Fig. 4: For panel b, please see same comment as before. For panel a, consider a discrete colorbar with set ranges (0-20, 20-40, etc.). It's very hard to see the differences between the domains.

Fig. 5, 6, 7: Same as 4a. I do not see the dotted line for the mean value mentioned in the caption.

Fig. 6: The dots in panel a are not visible.

---

## Author Comment (AC1)

In the manuscript "Assessing the glacier projection uncertainties in the Patagonian Andes (4-56 S) from a catchment perspective", a large set of glacier model simulations obtained with the OGGM model is used to assess the uncertainty in glacier runoff and glacier melt projections. The study looks at the effect of different glacier outlines, different glacier volume estimates, various historical climate datasets, different GCM, different emission scenarios and different bias correction methods. Each of these different datasets are discussed and it is shown how they vary, mainly focusing on their spatial patterns. Using a random forest regression method, the relative importance of each of these "model choice" uncertainties is examined. The study concludes that the reference climate is the most important source of uncertainty for a range of glacio-hydrological signatures, even for signatures that represent a signal beyond the reference period.

Overall, I think that the study presents an impressive amount of model run comparisons at a large regional scale and shows clear insights into spatial differences of glacier volume and runoff changes in the Patagonian Andes. Moreover, the effect of each source of uncertainty on the glacio-hydrological signatures nicely illustrates their variable importance for different aspects of the change in glacier runoff. However, I feel that some parts of the manuscript could be improved, such as the use of "catchments", and the discussion section, which should describe more the implications and possible hypotheses of the findings, rather than only a summary of the findings and comparison with other studies. Please find below a list of more detailed comments, also explaining these two examples.

R: Thank you for your thoughtful and constructive feedback on our manuscript. We appreciate your positive assessment of the extensive model run comparisons and the insights into spatial differences in glacier volume and runoff changes in the Patagonian Andes. We will consider your suggestions for improvement, particularly regarding the use of "% of catchment area" and the discussion section. We recognise the importance of providing a more in-depth discussion of the implications and possible hypotheses of our results, rather than simply summarising and comparing them with other studies.

We will make the necessary revisions to improve the clarity and depth of the discussion section to ensure a more thorough exploration of the implications of our findings. In addition, we will prioritise the use of 'glacier area (at inventory date)' rather than '% of catchment area' in the manuscript to ensure appropriateness and clarity for potential future readers. Please see our detailed responses to these general comments below.

**Abstract**

The introduction of "catchments" starts here in the abstract. Without stating what "catchment" represents in this study, it is quite hard to follow. The reader doesn't know if the study is about a few catchments, or many, and how they are delineated/defined. Accordingly, all the statements with "xxx% of the catchment area" are difficult to interpret. On a more general note, it did not became clear why catchments are used in the study? Most of the results are rather described on the basis of the hydrological zones. In the study area description, it is noted "847 catchments were selected" without any further information. On the basis of what were these catchments defined? Since the study is not using any downstream information or non-glacierized catchment info ("only using glacierized grid

cells"), I think the "catchment" part should not be part of this study. Results could then be just presented as xx% of glaciers show this and that, while also keeping the aggregation level of the hydrological zones.

R: We acknowledge the concerns raised about the introduction of "catchments" and the resulting confusion for the reader. We recognise the importance of clarity in our presentation, and therefore we agree that the use of "% of catchment area" in the comparison between the different sources of uncertainty may add unnecessary complexity to the study. Consequently, we have decided to remove these values from the manuscript.

In the revised version, we plan to provide a simpler presentation by i) focusing only on "glacier areas (at inventory date)" in our analysis and ii) adding glacier area-weighted averages to summarise the values. These will be incorporated into Figures 5-7 and 10 and will contribute to a clearer and more concise presentation of our results. Finally, we plan to keep the aggregation level of the hydrological zones to explore the spatial differences over a wide area that includes different climates and geographical features.

**Introduction**

L44 undergoing "a" shift – shouldn't shift not be an increase, as otherwise there is no increase in risk? Or does it relate to larger volumes that are stored before bursting out?

R: The specific sentence on GLOFs will be removed to improve the continuity with the following paragraph.

L65 "have had to be used" – have been used? Also check the "but" which is more an additional problem than a contrast?

R: The specific sentence will be removed

I would suggest to move table 1 to the SI. The abbreviations in the first column are not yet clear and in general the aim/message of the table did not become clear in the introduction

R: Thanks for the comment. Table 1 will be moved to the Supplementary Material. The abbreviations in the first column will be defined in the table caption.

L93 "to project the evolution of the glacier area" – the model is not only used for projecting the area?

R: The specific sentence says "to project the evolution of each glacier (area > 1 km$^2$)". The glacier model outputs used are not only the glacier area, but also the volume and hydrological outputs (glacier runoff and melt from formerly glacierised areas, see Section 3.4).

**Study area**

L104-105 "crucial", "essential" in the same sentence, maybe one is enough?

R: The sentence will be summarised: "the seasonal melting of glaciers is essential for the long-term sustainability of the local ecosystems and coastal human populations".

**Methods**

Step 1 of the calibration procedure – I wondered if it should be discussed what the effect is of only calibrating the melt factor/temperature sensitivity using geodetic mass balances. If the precipitation is off (e.g. too little) then the temperature sensitivity might be smaller, to compensate?

R: It is true that the calibrated temperature sensitivity parameter depends strongly on the assumed precipitation, and thus on the chosen precipitation factor (Pf; see e.g. Schuster et al. (2023). However, for most glaciers we have only one robust observation, i.e., the 20-year geodetic mass-balance average. Therefore, we decided to set the Pf to 1, as some regional climate datasets used in this study (e.g. PMET and CR2MET) already include a bias correction process to correct for potential precipitation underestimation, and therefore we expect the range of "true" precipitation to be covered by the different products. This will be added to the discussion.

Step 2 – the description is a little bit unclear, is the gradient adjusted? Or what is meant with "a mass balance residual to add"?

R: Sorry for the confusion, the gradient of the mass balance is not adjusted. In fact, the whole mass balance profile is shifted so that the resulting apparent mass balance is zero, to satisfy the equilibrium assumption. We will make this clearer in the manuscript.

Step 3 – maybe the equation could be given here? I think it is similarly known as equation 1, but helps the reader understanding what is needed for the inversion and at what scales it is applied

R: Thanks for the suggestion. We think that the equations for the inversion are not that relevant for the paper and are already very well documented in Maussion et al. 2019. However, we will add the equation for the ice velocity to illustrate the control of the ice creep parameter A (which is calibrated in this step).

Step 4 – what is a "constant mass balance run"?

R: A constant mass balance run means that we define a mass balance over a period of time (e.g. the average mass balance of all the years in the period), this is our constant mass balance. Then we use this constant mass balance and let the glacier evolve for a few years. We will clarify this in the manuscript.

Figure 2 – I think it would be helpful to add the number of the steps in the figure. And what is the arrow from reference climate to bed inversion? It may refer to one step before that?

R: Thanks for the suggestion, we will add roman numerals in Fig. 2 to refer to the calibration steps in Section 3.1

L259 "according to the location of the glacier terminus" – I am not sure to understand this addition? Maybe some more explanation of glaciers that are crossing catchment/hydrological zone borders could be added here

R: Thanks for the comment. This will be clarified according to your suggestion: "For area and volume, we calculated the relative and absolute differences for each catchment and hydrological zone defined in Fig. 1.

**To calculate these differences, we aggregated glacier area and volume for a given catchment by selecting all glaciers with their terminus location within that catchment. It is assumed that, if the inventory outlines are correct, all the water flowing out of the glacier will flow via its terminus.".**

L291 here the text could benefit from some explanation/ careful discussion how streamflow metrics can be used to apply on aggregated glacier runoff data (i.e. glacier runoff is not the same as downstream streamflow and so their effect on the aquatic ecosystem is not 1:1 comparable)

**R: Thanks for the comment. We will add a sentence to highlight the limitations of our analysis: "However, our analysis of glacier runoff should not be considered as downstream streamflow because our simulations only considered the initially glacierised area and did not include the interaction with other hydrological fluxes (e.g., evaporation and infiltration)"**

Table 2 misses information about which period was used to calculate the signatures, apart from the ones that explicitly state "ref and future period"

**R: We will add a "Period" column for each signature.**

Header of 3.5 – maybe choose the titles of section 3.3 and 3.5 in such a way that it is more clear what their different content is

**R: The original header (Uncertainty analysis) will be renamed to "Hydrological importance of sources of uncertainty"**

L299 – where comes 329 from?

**R: Out of 847 catchments, only 329 catchments have one or more glaciers in both inventories (RGI6 and RGI7). This will be clarified in the text.**

In the calculation of RMSE, what is the baseline? i.e. how is RMSE calculated?

**R: Thank you for your question. The permutation feature importance measures the change in model performance (in this case, the Root Mean Square Error; RMSE) after the values of a single model feature have been permuted (also known as shuffled), with more important features resulting in greater decreases in performance when permuted. The baseline corresponds to the model performance before the permutation of the model features (in this case, the six sources of uncertainty). This will be clarified in the manuscript.**

**Results**

Figure 2: possibly remove the blue/gray background so the results are better visible

**R: We think that this comment refers to Fig 3, as there are no results in Fig 2. We will add black outlines and a zero line to the bar plot in Fig 3b to prevent the bars being confused with the background.**

L335 – why was the glacier thickness divided by the catchment area, while all other variables are focused on the glaciers/glacier grid cells?

**R: In the case of ice volume, which is derived from ice thickness and glacier outline, we decided to normalise the value by catchment area to facilitate the comparison between catchments - hydrological zones.**

L374 – maybe I missed it, but what are "the main catchments"?

**R: The main catchments correspond to the catchments that have an area greater than 5,000 km$^2$ (Fig. 1), which account for the 68% of the total catchment area. All maps have the following sentence to indicate the main catchments: "The names in grey correspond to the names of the main catchments"**

L408 "historical sources" – climate?

**R: Yes. This will be clarified in the text ("Considering the prolongation of historical climate conditions, 26% ± 9% of the total glacier ice is committed to melt in the long term")**

L435-L440 Does isolating the melt from glacier runoff result in more or less uncertainty? There is a hint that melt only has an effect of temperature, but it also states that precipitation can compensate for the change in melt? The 60% suggests that 40% of the glacier runoff is generated from off-glacier melt, or liquid precipitation on+off glacier?

**R: Compared to total runoff (Fig. S2), the uncertainty (standard deviation of the annual glacier melt) was reduced because the reference climate, which is the main source of climate uncertainty, had less influence, as the strongly past climate-dependent liquid precipitation is only included in the glacier runoff components (on- and off-glacier melt and on- and off-glacier liquid precipitation). This is mentioned in the Discussion section (L553-L556). Therefore, the 60% suggests that 40% of the glacier runoff is generated by off-glacier melt and on- and off-glacier liquid precipitation.**

**Discussion**

For the discussion in general, I found it sometimes hard to follow what was described in the different parts. The first part discusses the uncertainties, but the other two parts as well. Maybe by less repetition of the results, and more focusing on the implications of the results would help in restructuring the discussion. I think the comparison with other studies is mostly well described, but would also benefit from an additional thought on what are the implications of that it agrees well or not.

**R: Both reviewers have raised concerns about the discussion section. To address this, we have proposed a new structure for this section. This revised structure aims to minimise repetition of results and to consolidate previous sections on uncertainty. In addition, we plan to incorporate the suggestions of both reviewers by focusing more on the implications of the results, rather than simply comparing them with other studies. This approach will improve the clarity and coherence of the discussion and allow readers to better understand the significance of our findings.**

The proposed new structure is as follows: Section 5.1 ("Changing glacier hydrology") will present regional projections of changing glacier hydrology and compare them with results from previous studies. Section 5.2 ("Hydrological importance of data uncertainty") will discuss the hydrological importance of data uncertainty, summarising comparisons and highlighting the importance of sources of uncertainty, emphasising spatial differences and the importance of domain characteristics, as suggested by Reviewer 2. In addition, the importance of model calibration is addressed, as suggested by Reviewer 1. Finally, Section 5.3 ("Limitations and global implications") discusses limitations such as unconsidered sources and potential global implications suggested by both reviewers.

I was wondering if there should be maybe a discussion on the way such glacier models are calibrated? If all model runs are calibrated equally well (RGI areas and glacier volumes are used for calibrating), then how come the results are so different, especially regarding reference climates? These parameters propogate in all other simulations, right? This must then come from processes that are not captured when looking at annual and long-term metrics only when calibrating model parameters? Or maybe there are other processes that are relevant to discuss for improving glacier modelling?

R: We recognise the need to discuss how glacier models are calibrated and the potential factors that contribute to differences in results despite the use of the same calibration approach. This will be addressed as part of the new section 5.2 entitled "Hydrological importance of data uncertainty" (see previous response). In this section, we will explore the implications of model calibration. In section 5.3 ("Limitations and global implications"), we will discuss the need to use additional data to calibrate other relevant processes that may be critical for improving the accuracy of glacier modelling. The proposed new discussion aims to provide insights into the complexities of glacier modelling and strategies for improving its reliability.

L484 "acquisition dates" – this sentence could benefit from more explanation about what the difference is in both RGI outlines with respect to acquisition dates.

R. Thanks for the comment. The sentence will be modified with further explanation: "The decrease in glacier area in the major ice masses from both inventories may be due to improved outlines from the local inventories and differences in the acquisition year of glacier outlines (Fig. 3d). Furthermore, these differences show the effect of climate and water warming on the complex dynamics of these glaciers (Minowa et al., 2021; cited in the main text)".

L487 "this threshold" – do you mean a 10% threshold for volume?

R: Yes, the sentence compares the relative differences between different sources of glacier inventory and ice thickness estimates.

L494 – Isn't this paragraph suggestion a contradiction? It would help starting the paragraph like that.

R: The main topic of the specific paragraph is the uncertainty associated with the reference climate (in specific precipitation). The content of this paragraph will be summarised and included in the subsection: "Hydrological importance of data uncertainty".

L510 – a space needs to be inserted before "The selected…."

**R: Thanks for the catch. A space will be inserted there.**

L512 "the ten selected….of the assessed warming – not sure how this sentence fits in?

**R: The specific sentence will be removed from the discussion to avoid repetition of the methodology/results.**

L526 Why are the "older" estimates named after the newer estimate and referred to as "recently"?

**R: Thanks for the question. The "recently" was misplaced there and will be removed from the sentence.**

L539 "of its current volume" – although it is clear that the study only discusses glaciers, the confusing use of "catchments" means that here there "glacier" needs to be added

**R: Thanks for the comment. The reference to "catchments" will be removed from the sentence to clarify the source of the volume loss.**

L594 – possibly add that Mackay et al. deals with glacio-hydrological modelling in Iceland

**R: Thanks for the comment. We will add this to clarify the scope of the antecedent.**

**Conclusions**

L610 – "differences", "different", "varied" in one sentence – consider rephrasing.

**R: The specific sentence will be rephrased: "The six sources of data uncertainty showed differences of varying magnitude".**

L613 isn't it a separate point, the one on reference climate being most important for uncertainty?

**R: We decided to aggregate the results of Section 4.1 ("Analysis of sources of uncertainty") as one point of the conclusions. L613 compares the relative differences between glacier inventories and volume data sources (Section 4.1.1), while L614 refers to future sources of climate uncertainty (climate, SSP, GCM and bias correction method; Section 4.1.2 and Section 4.1.3).**

L622 "tended to converge towards an overall decrease" – decrease of what?

**R: Decrease of glacier runoff and melt. This will be clarified in the conclusions: "glacier runoff and melt projections aggregated by SSP scenario tended to converge towards an overall decrease throughout the 21st century"**

L625-628 double?

**R: Yes, the specific sentence indicated that regardless of the spatial scale of interest (catchment or glacier area), the main source of uncertainty was the reference climate. This sentence will be clarified as the "catchment area" will be removed from the Results and Discussion sections**.

L629 – what is meant with "local"?

R: The specific word will be removed from the sentence.

L634 "relative contribution of non-glacial water sources" – shouldn't future studies not focus on the dynamics of these sources, rather than the relative contribution? And what should future studies do with these non-glacial water sources? The follow up sentence does not directly fit here, i.e. there is some gap between knowing a relative contribution and understanding other catchment stores. Maybe a general sentence on extending the scope from glaciers to downstream hydrology fits better?

R: Thanks for the suggestion. We agree that a general sentence about broadening the scope is more appropriate than focusing only on non-glacial water sources. In the revised version, we will modify this as suggested: "future studies should address sources of uncertainty not considered in this study (…) and extend the scope from glaciers to downstream hydrology. Downstream hydrology can play a critical role in the seasonal and interannual water release during dry seasons (Drenkhan et al., 2022), attenuating the consequences of glacier shrinkage (e.g., Somers et al., 2019)"

---

## Author Comment (AC2)

In this manuscript the authors quantify the effects of historical and future glacial and climatic conditions on simulated glacier melt in the Patagonian Andes using the Open Global Glacier Model (OGGM) and Random Forest Regression approach. They focus on six categories, or sources of uncertainty, namely glacier outline inventories, ice thickness, historical climate, GCMs, bias correction methods, and emission scenarios. They examine the importance of each source on ten runoff metrics (e.g., peak water year and magnitude, interannual variability, seasonal contribution and variability, etc.) and conclude that the choice of reference climate is the predominant source of uncertainty.

The authors have undertaken a fairly comprehensive assessment, exploring 1920 cases across the six categories, to identify how the choice of input and forcing data affects the outcome in a glacialhydrological modeling workflow. This is overall a well-written paper, and the figures support the narrative well. The authors have demonstrated the methods clearly and the results are presented in a logical way.

The areas which require considerable improvement are the Discussion and Results sections which lack coherency and do not tie the various pieces together. There are differences across the hydrological zones/catchments and the ten glacier runoff metrics which require more nuanced discussion. Further, the authors need to explicitly highlight the implication/s of this work and explain if/how the findings here can be used for other glaciated regions of the world (i.e., does this study domain encompass a comprehensive set of climatic, glacial, and hydrological conditions to make general deductions).

I recommend moderate revisions before the manuscript is ready for acceptance. These revisions mostly require either clarification or further elaboration in the text. Comments below are provided section-wise and are not in order of importance.

R: Thank you for your thorough review of our manuscript. We appreciate your positive evaluation and constructive feedback. We take note of your suggestions for improvement, in particular to improve the coherence of the Discussion and Results sections. We are committed to making the necessary revisions to address these concerns and provide a clearer and more comprehensive understanding of our findings. In addition, we will explicitly highlight the implications of our work and discuss how the results can be applied to other glaciated regions worldwide. Please see our detailed response on how we plan to restructure the Discussion section in the related comment below.

**Introduction**

Ln 61-62: Can you clarify the statements related to "*significantly increased streamflow*" and "*significant trends*". I presume the latter refers to statistical significance and the former to a large magnitude increase? Also linking with the previous statement, is it possible to extract information from these past work on how much (%) the streamflow has increased because of the accelerated glacial mass loss?

R: Thanks for the opportunity to clarify these points. As you mentioned, the "significant" refers to the existence of a statistical significance. This will be clarified in the text (only Pasquini et al. 2021 used a

statistical test): "recent studies have reported increased flows in rivers with important glacier cover (Masiokas et al., 2019; Vries et al., 2023), some of which have only begun to show significant trends (p < 0.001) in the last decade (e.g., Santa Cruz; Pasquini et al., 2021)"

To our knowledge, there have been no comprehensive studies to date that would allow us to determine the percentage of streamflow that has increased as a result of accelerated glacier mass loss. We are addressing these important questions in an ongoing study that we expect to be available as a preprint this autumn (March - April).

Ln 67-68: Is this referring to overestimation of precipitation? Also, what does "*diverged towards*" mean?

R: Yes, this will be clarified in the text: "the different approaches and data sources have overestimated the precipitation according to numerical simulations of regional moisture fluxes."

Ln 70 onward: I am somewhat unsure about the purpose of this table (and generally this paragraph). The table summarizes past mass balance assessments; however, the current study is not considering various MB schemes or calibration processes as a source of uncertainty. This table seems superfluous and if the authors want to keep it, please consider moving to the supplementary material. Instead, please expand on the two Patagonian glacier hydrological contribution studies (*Mernild 2017* and *Caro 2023*). That will link better with the preceding paragraph and the overall theme of the paper.

R: Thanks for the suggestion. Table 1 will be moved to the Supplementary Material, and we will add a few sentences about the two previous hydrological studies as suggested.

Ln 89: "… *adding additional data to the calibration*…" What type of additional data is this referring to?

R: In parenthesis, we will add the most frequent complementary data used by glacio-hydrological models (snow cover area and glacier mass change; Van Tiel et al., 2020)

Ln 91: Please explicitly mention the six sources of uncertainty here, bringing information in Ln 94 – 96 earlier, and then mention the tools used (OGGM, random forest).

R: Thanks for your comment. The specific paragraph will be reorganised according to your suggestions.

Ln 91 onward: The preceding paragraph talks about different sources of uncertainty in the modeling chain in current literature (beyond what the authors have explored in this work). Various statements in this study imply that it focuses on comprehensive sources of uncertainties across the modeling chain. In reality, it only considers two sources i.e., model input data (glacier outlines, thickness) and atmospheric forcing (historical data, GCMs, climate scenarios, and bias-correction methods). I would not consider this encompassing "*uncertainty sources in the modeling chain*" (e.g., Section 5.1 heading). The full chain will include considerations such as model workflow, parameter space, MB or ice flow schemes, etc. (this is duly mentioned as a study limitation in Ln 517).

I recommend removing the term "glacier modeling chain" here and in other instances, including the abstract, and explicitly mention that various configurations of model input and forcing data are used to identify the dominant

sources of uncertainty. This is an extensive assessment with these two broad categories as it is, so there is no need to overstate the study's scope.

**R: We appreciate the insightful feedback and acknowledge the concerns raised about the term 'glacier modelling chain'. After careful consideration, we agree that the term may not accurately reflect the scope of our study. We will remove the term 'glacier modelling chain' and explicitly clarify in various sections, including the abstract, that our focus is on the impact of different sources of 'data' uncertainty. The revised version will better emphasise our comprehensive assessment within the specified categories of model inputs and forcing data, without overstating the scope of the study.**

**Study area**

Ln 103: What is meant by pristine environment here? Is this referring to glaciers or the water resources?

**R: The specific sentence will be removed.**

Ln 111: These nine hydrological zones are referred to extensively throughout the paper. Please fully name all of them here.

**R: The nine hydrological zones will be defined in the indicated paragraph. Particular attention will be given to the northern zones of PPY and PCA which were only defined in Fig. 1.**

Ln 111: I understand how the nine hydrological zones were identified and demarcated, but can you please elaborate how the 847 catchments were selected. Was having a 0.1% glacier area of the total catchment area the only criteria? Why was this threshold selected?

**R: Thanks for the question. We selected 847 glacierised catchments, each with at least one glacier and a glacier area greater than 0.1%. The 0.1% was selected as a conservative threshold for drought buffering (please see Fig. 3 in Ultee et al. 2022; cited in main text). This clarification will be added to the main text.**

Ln 113: Can you please explain "*high explanatory power of recent glacier change*" or reword this statement.

**R: The statement will be clarified: "that shown a strong capacity to reproduce recent glacier changes"**

Ln 123: "*are characterized by many small catchments*". Replace characterize by 'which hosts many small catchments' or something similar.

**R: Thanks for the suggestion. This will be changed as suggested ("which hosts many small catchments").**

**Methods**

Ln 142: Please mention the upper and lower thresholds here in brackets.

**R: This will be added to the sentence in brackets [0, 2 ºC]**

Ln 146: OGGM is a flowline model so what is meant by "local grid" of 10 – 200m? Is it referring to the spacing between cross-sections across the centerline?

**R: Yes, in the context of OGGM, the "local grid" refers to the spacing between grid points on the glacier surface. This grid is used to obtain the model glacier geometry by overlaying glacier inventory outlines and NASADEM elevation data. The resolution of the grid varies with the glacier size, ranging from 10 to 200 m. Glaciers are then segmented into elevation bands following the algorithm proposed by Werder et al. (2020), each of which covers an elevation difference of 30 m. This response will be used to clarify this procedure in Section 3.1.**

Ln 151: If the precipitation factor is set a single value of 1, then it is no longer accounting for biases due to topography, missing processes etc. (Ln 137). Also, what does it mean to "… *assess influence of different reference climates* …" Is this referring to the temperature sensitivity parameter as the precipitation factor is set to 1 for all cases.

**R: Thanks for the comment. It is true that setting the precipitation factor (Pf) to 1 no longer accounts for the missing local processes, and therefore these processes are included in the modelling framework with the temperature sensitivity parameter. Regarding the second point, the statement also includes the historical precipitation of the reference climate dataset, as it influences, for example, the interannual variability of the glacier mass balance. The decision to use Pf = 1 is based on the fact that some of the regional climate datasets used in this study (e.g. PMET and CR2MET) already include a bias correction process to correct for potential underestimation of precipitation. Unlike many previous studies which strongly "correct" the input datasets before using them to drive the impact model, we chose to incorporate the uncertainty of the driving dataset in our evaluation. This will be added to the discussion on model calibration.**

Ln 163: Replace "*new value*" with initial value or first guess.

**R: The "new value" will be replaced by "initial value"**

Ln 172: Is the simulated glacier volume set to match the input volume accumulated over each of the nine hydrological zones and not the individual 847 sub-catchments?

**R: Yes, please see the reply on Volume-Area scaling (VAS) below**

Ln 187: Remove "it" after compared.

**R: Thanks! The "it" will be removed.**

Ln 191: This might be a typographical error, what is N°1?

**R: Nº1 refers to the first step. This will be clarified.**

Ln 201: What does "*latitudinal patterns in terms of area*" mean?

**R: The specific sentence will be replaced by "both datasets show similar areas across different latitudes"**

Ln 209: It helps to keep the language simple. Can this be rephrased as M22 having 13% larger volume than F19… or something along those lines.

**R: Thanks for the suggestion. The sentence will be modified as suggested ("In the southern Andes, Hock et al. (2023) reported that M22 had 13% more total ice volume than F19")**

Ln 210: "Both alternatives" - consider replacing with the 'two volume data sources' or something similar.

**R: Thanks, we will follow this suggestion.**

Ln 213: By dataset do you mean the 4 gridded products?

**R: This paragraph refers to the two ice volume datasets. To clarify this, we will replace "datasets" with "volume data sources", as suggested in the previous comment.**

Ln 213: VAS was done for the 9 hydrological zones; can you explain here why not for the 847 individual catchments?

**R: Thanks for the question. Considering that most catchments have only a few glaciers, and the theoretical basis of VAS is determined for samples of glaciers spanning a wide range of sizes (and is validated by observations covering a wide range of sizes; Bahr et al. 2015), we decided to calculate the VAS parameters using the hydrological zones.**

**Bahr, David B., W. Tad Pfeffer, and Georg Kaser. "A review of volume-area scaling of glaciers." Reviews of Geophysics 53.1 (2015): 95-140.**

Ln 228: Can you please elaborate on why/how these specific GCMs were selected? Was this a subset from a larger initial pool and the 10 GCMs were selected based on their TCR and ECS?

**R: Thanks for the question. We will add a sentence to clarify this point: "Using the full CMIP6 ensemble, the selection of the 10 GCMs was based on the recommendations of Hausfather et al. (2022), who suggest focusing on a subset of GCMs that are most consistent with the assessed warming projections of the Sixth Assessment Report (AR6)." For more details on the source of the full ensemble, please see the "Data availability" section.**

Ln 234 – 236: This is rather confusing: all GCMs have ECS falling in "very likely" range but only 80% in "likely"? Is the very likely range broader than the likely range? The "likely' range in *Hausfather22* was narrower (1.4 - 2.2 C).

**R: Thanks for the comment, it gives us the opportunity to clarify this point. Hausfather et al. 22 recommended screening out models with a TCR that lies outside the 'likely' range of 1.4–2.2 °C, or alternatively using a "likely" ECS of 2.5-4°C, which also reproduces the AR6 results well. In view of these alternatives, the sentence relating to ECS will be removed from the main text to improve the clarity of the GCM selection.**

Ln 257: It is somewhat unclear where, when, and why catchments and hydrological zones are considered separately, e.g., volume-area scaling was done at the hydrological zone scale, glacio-hydrological signatures were assessed at catchment scale (Ln 288), etc.

**R: Thanks for the comment. Our initial intention was to conduct the complete study using the catchment scale only. However, in order to present the glacio-hydrological projections in Section 4.2 (Fig 8, 9 + Fig S1 and S2) we needed a clear level of aggregation (i.e., hydrological zones). Additionally, the theoretical basis of VAS is determined for samples of glaciers spanning a wide range of sizes, and therefore, the proposed zones were more appropriate than the catchments (please see the previous response about VAS procedure).**

Ln 259: Not sure I understand what is meant by area/volume aggregation based on terminus location.

**R: Thanks for the comment. This will be clarified in the specific paragraph: "For area and volume, we calculated the relative and absolute differences for each catchment and hydrological zone defined in Fig. 1. To calculate these differences, we aggregated glacier area and volume for a given catchment by selecting all glaciers with their terminus location within that catchment. It is assumed that, if the inventory outlines are correct, all the water flowing out of the glacier will flow via its terminus".**

Ln 261: This lapse rate was used because it is default in OGGM and also used within this domain in the past studies (Ln 141). Table 1 however also shows that literature used different lapse rates depending on the region, e.g., 6.5 for NPI and 5.8 for GCN. Is there a north-to-south gradient in the mean annual lapse rates? Does that affect the downscaling of ERA5 data? Also, is this downscaling step the same as what is mentioned in Ln 141? What about total precipitation, how was that handled when going from a quarter degree to 0.05˚ in ERA5?

**R: Thank you for the question, it gives us the opportunity to elaborate on our decision to continue using a constant environmental lapse rate of 6.5 ºC km$^{-1}$ to downscale the temperature. Although recent studies in Patagonia have demonstrated the variability of this value (Bravo et al. 2019 in SPI), recent efforts during the development of the PMET (Aguayo et al. 2024; citation will be updated in the text) have shown that seasonal and regional variations of this value in Patagonia do not significantly improve performance (results based on > 100 meteorological stations). Using a lapse rate of 6.5 ºC km$^{-1}$, all reference climates used in this study are able to achieve mean biases of less than 1 ºC (Fig. 5 in Aguayo et al. 2024). No downscaling method was used for precipitation. This information will be added to the main text.**

**Bravo, C., Quincey, D. J., Ross, A. N., Rivera, A., Brock, B., Miles, E., & Silva, A. (2019). Air temperature characteristics, distribution, and impact on modeled ablation for the South Patagonia Icefield. Journal of Geophysical Research: Atmospheres, 124(2), https://doi.org/10.1029/2018JD028857, 907-925.**

**Aguayo, R., León-Muñoz, J., Aguayo, M., Baez-Villanueva, O. M., Zambrano-Bigiarini, M., Fernández, A., and Jacques-Coper, M.: PatagoniaMet: A multi-source hydrometeorological dataset for Western Patagonia, Sci Data, 11, 6, https://doi.org/10.1038/s41597-023-02828-2, 2024.**

263: Please mention these default parameters either here or in supplementary material. I suppose Ln 153-154 mentions a couple of these, but please enlist all the parameters for reproducibility. Model "defaults" can change over time as new information becomes available.

**R: The parameters will be added here ($T_{melt}$ = -1 ºC, $T_{solid}$ = 0 ºC and $T_{liquid}$ = 2 ºC).**

Ln 264, 273, 361, etc.: "*Glacierized grid cells*" – to confirm, these grid cell-based computations are only done for input & forcing data because OGGM is a flowline model and the output information (area, volume, melt) will be for a specific glacier?

**R: Yes, OGGM uses the climate data from the nearest grid cell to generate outputs for each glacier. In the case of air temperature, the raw climate data is downscaled to the glacier surface (elevation bands) using a constant lapse rate.**

Ln 275: To clarify, when uncertainty metrics are computed all glaciers are considered (Ln 256), but for hydrological assessments only this subset is considered? Section 4.1 is for all the glaciers in RGI and Section 4.2 is for these ~2000 glaciers?

**R: Yes, you are correct. Due to computational limitations, we used the OGGM model to estimate the evolution of only the glaciers with an area > 1 km² (~2,000 glaciers) (Section 3.4). The comparative analysis (Section 3.2) was performed considering all glaciers in RGI6 and RGI7.**

Ln 278 – 279: Up till this point, I do not think historical (outline, volume, ref. climate) vs future (scenarios, GCMs, BCM) sources are explicitly mentioned and separated in the text. So, the "*2 . 2 . 4*" reference is not clear.

**R: Thanks for the comment. We agree that it is important to explicitly mention this here: "For each glacier, we evaluated 16 scenarios generated by the historical sources of uncertainty: glacier outlines (n = 2), volume datasets (n = 2) and reference climates (n = 4). These scenarios were used to project the future evolution given by different GCMs (n = 10), future scenarios (n = 4), and bias correction methods (n = 3), resulting in 120 future scenarios for each historical simulation (a total of 1920 potential scenarios; Fig. 2)"**

Ln 283-284: I understand on- and off-glacier liquid precipitation, and "on-glacier" melt. But what is off glacier melt? Is it referring to snow on non-glaciated areas? Also, the term on-glacier melt is unusual, I believe this is referring to direct melt from glacier?

**R: The melt off-glacier melt corresponds to snow melt on areas that are now glacier-free (i.e. 0 at the start of the simulation; in our case 1980). The term "melt on glacier" will be renamed to glacier melt and clarified in this text: "we also extracted the melt on glacier (hereafter glacier melt), which is the sum of ice and seasonal snow melt on the glacier (Fig. 2c)".**

Ln 284: It seems this line and onward is now referring to the full 1920 scenarios and not the 16 historical ones? How is climate uncertainty influence being overestimated? What is meant by 'climate' here, and what is precipitation reduction?

**R: Thanks for the questions. We will divide the specific paragraph into two: one for the scenarios and the following for the variables extracted from OGGM. The second paragraph will start with: "For all 1920 scenarios, we extracted…"**

**Regarding the second point, this will be clarified in the text: "To disaggregate the impact of projected precipitation changes, we also extracted the melt on glacier (hereafter glacier melt)…"**

I suppose "glacier runoff" here is the sum of glacial melt and liquid precipitation on glacier? Please consider rephrasing the 'melt on glacier' with an equivalent term commonly used in literature.

**R: Yes, at the beginning of the simulation (i.e., year 1980) glacier runoff is the sum of glacier melt and liquid precipitation. As the glacier retreats, off-glacier melt (seasonal snow) and liquid precipitation are also considered (glacier runoff corresponds to all water originating from the initially glacierized area).**

**The term "melt on glacier" will be renamed to glacier melt and clarified in this text: "we also extracted the melt on glacier (hereafter glacier melt), which is the sum of ice and seasonal snow melt on the glacier (Fig. 2c)".**

Ln 286: Is this referring to the melt and precipitation time series? Also, I do not understand the meaning of 'according to glacier terminus' here.

**R: In the previous comments, we have defined and clarified the glacier runoff and melt time series. The glacier terminus reference point will be clarified in Section 3.3, and a reference to that clarification will be included here: "As in the comparative analysis (Section 3.3), the time series were initially aggregated at the catchment scale according to the location of the glacier terminus".**

Ln 288: Now this assessment is at catchment scale and not hydrological zone scale, again it gets fuzzy where hydrological zones vs catchments are considered. Also, aren't SSP-based scenarios 4, and n=1920 the total number of scenarios?

**R: As mentioned above, our original intention was to conduct the entire study at the catchment scale only. However, in order to present the glacio-hydrological projections in Section 4.2 (Figs. 8, 9 + Figs. S1 and S2), we needed a clear level of aggregation (i.e. hydrological zones). In addition, the theoretical basis of VAS is determined for samples of glaciers covering a wide range of sizes, and therefore the proposed zones were more appropriate than catchments. On the other hand, in order to identify clear patterns using the permutation feature importance of RF regression models, we need to have a representative number of samples greater than just 9 hydrological zones. We hope that this clarification will help to understand our decisions that prevented us from using only one scale of analysis.**

**Results**

Ln 321-322: Can you please make the statements explicit here, e.g., RGI7 has 4% and 15% greater area than RGI6. The "*showed positive/negative difference*" is perhaps not the best way to state this.

**R: The specific sentence will be rephrased: "showed increases ranging from 4% to 15% relative to RGI6". We agree that "showed positive/negative difference" is not the clearest way to show the differences and therefore we will rephrase similar sentences.**

Ln 325: There is only a 1–year difference in the acquisition date for most glaciers from RGI6 to RGI7 (Fig. 3c) – did that make such a large difference in terms of area reduction?

**R: Although a detailed comparison between the different sources of data uncertainty is beyond the scope of this study, the differences reported here are similar to the detailed comparison made by Zalazar et al. (2020) (see Fig. 11 there). For example, Zalazar et al. (2020) showed that the largest absolute differences are in the Patagonian Icefields, and some (one-degree) latitudinal bands can show relative differences of more than 50% in the Patagonian Andes. It should also be noted that improved outlines and corrections from local inventories in RGI7 contribute to the observed differences in glacier area.**

Ln 335: Just for clarification, both M22 and F19 have the same glacier outlines taken from RGIv6?

**R: Yes, in order to make a proper comparison we use RGI6 in both ice volume datasets. This was mentioned in caption of Fig. 4 and will be added to the indicated sentence.**

Ln 336: Fig. 4b is the M22 minus F19, correct? Most of the area is in yellow shades visually so M22 shows less volume. Perhaps it is the colorbar that needs to be changed.

**R: Yes, that's correct. Panels B and C are differences in percentage relative to F19 ((M22 - F19) / F19). We will adjust the range of the colormap from [-100, 100] to [-50, 50] to have more visible differences.**

Ln 351: Replace "in" with "over 51% of the glaciated area".

**R: This will be changed as suggested.**

Ln 352: Throughout the paper, the reference to glacier/glaciated area and catchment area together is very confusing, for example this line mentions 51% of glacier area, 22% of the catchment area for the precipitation reference and 95% of the glacier area, 99% of the catchment area for temperature reference. Please consider presenting this information in some other way (perhaps focus on one of these only).

**R: Thanks for the comment. This was one of the general comments of Reviewer 1. The percentages related to catchment area will be removed from the main text in order to the information clearer to the future reader.**

Ln 356: Similar patterns in terms of spatial patterns?

**R: Thanks for the question, the wording was not clear. The sentence will be clarified: "In relation to PMET, CR2MET and MSWEP showed a difference of nearly -50% in solid precipitation, while ERA5 showed smaller differences close to -20% (Fig. 5c)."**

Ln 356: The spatial resolution of the native data is quite different (0.25 for ERA5, 0.05 for PMET/CR2MET), does that come into play at all?

**R: Please see our previous response about the spatial resolution of the climate data (use of lapse rates)**

Ln 367: Also, are there other characteristics that create these latitudinal differences in (a) precipitation change sign and (b) model agreement/disagreement?

**R: The projected changes in precipitation follow a latitudinal pattern which generates a low agreement in the intermediate zone (SPI and GCN) due to the fact that it is a transitional zone (ΔPP = 0%). These projections are consistent with historical trends that has been attributed to the Southern Annular Mode (SAM), an index that describes the movement of the low-pressure belt that generates westerly winds (Fogt et al. 2020). This index has recorded a transition to a positive phase, causing a decrease in the intensity of westerly winds at mid-latitudes, which generates a significant part of the decrease (increase) in precipitation in northern (southern) Patagonia.**

**Fogt, Ryan L., and Gareth J. Marshall. "The Southern Annular Mode: variability, trends, and climate impacts across the Southern Hemisphere." Wiley Interdisciplinary Reviews: Climate Change 11.4 (2020): e652.**

Ln 370: Why/how is ice volume relevant to climate projections? This is a rather obtuse statement, talking about GCM model disagreement and then ice volume estimates.

**R: The specific sentence ("areas characterized by a high ice volume") will be removed from the paragraph.**

Ln 374: What do you mean by the "main catchments".

**R: The main catchments correspond to the catchments that have an area greater than 5,000 km$^2$ (Fig. 1). All maps have the following sentence to indicate the main catchments: "The names in grey correspond to the names of the main catchments"**

Ln 405: What do you mean by the prolongation of the mass loss? Is this referring to Fig. 8b?

**R: Yes, but more generally to the whole of Fig. 8, as all panels refer to volume loss, which will be clarified in the specific sentence (volume instead of mass): "Projections from OGGM indicate a prolongation of the glacier volume loss of recent decades (Fig. 8)."**

Ln 406: Again, please reconsider the discussion related to catchment and glacier area. It is surprising that 18% of glacier area is equivalent to 43% of the catchment area. Also, how is catchment losing volume (it is only the glaciers that will be losing the volume).

**R: Thanks for the comment. This was one of the general comments of Reviewer 1. The percentages related to catchment area will be removed from the main text in order to the information clearer to the future reader. In this case, the difference between 18% and 43% is explained by the fact that the glaciers are located in the northern area which is characterised by the largest catchments (Petrohue, Puelo, Yelcho, Palena, etc.)**

Ln 417: Is this the maximum range across all the basins? It seems some basins have less or more spread across the scenarios (e.g., SPI-N). Again, can you comment on the difference between the hydrological zones here or in discussion.

**R: Thanks for the question. We will address this by calculating the differences across hydrological zones (in this case, specific mass balance) and discussing the spatial differences as part the proposed new structure of the discussion section. This will provide deeper insights into the observed differences between sub-regions and scenarios.**

Ln 427: 61% of the total catchment area contains 30% of the total glacier area i.e., the remaining 70% of the glaciated area in the ~40% of the catchment area? Here it is best to talk about glacier area (because it is talking about glacial melt), catchment area reference is somewhat misleading.

**R: We agree that the catchment area could be misleading, so the reference to catchment area will be removed as before.**

Ln 428: Is this melt now talking about all the glaciated area in the full study domain?

**R: Yes, this will be clarified in the specific sentence.**

Ln 430: Please consider rephrasing "the evolution of the melt on glacier…". Do you mean changes in melt rates?

**R: Thanks for the question. This will be clarified in the main text: "The projected trajectories of glacier melt varied slightly among emissions scenarios"**

Ln 435: This is referring to Fig. 9e for SPI, correct?

**R: Yes, we will correct the reference to the panel figure.**

Ln 436: Which panel of Fig. 5 is this referring to? Please rephrase "*melt on glacier evolution*".

**R: The reference to Fig. 5 was incorrectly located there and has been removed: "Relative to glacier runoff (Fig. S2), the uncertainty (i.e., standard deviation) in the glacier melt was lower due to a lower influence of the reference climate"**

Ln 455: Remove comma after 'contribution'.

**R: Thanks for the comments. The comma after 'contribution' will be moved before 'metrics': "This was especially clear for (…) metrics, where the reference climate accumulated…"**

Ln 468: What is meant by lower importance of climate here? Reference climate?

**R: Yes, this will be clarified in the revised version ("The lower importance of the reference climate was limited to…").**

Ln 470: Can you explain why there are differences between seasonal contribution, variability and shift in Fig. 10? What are the mechanisms causing these differences in sources of uncertainty.

**R: Before addressing the differences observed in Fig. 10, it is important to acknowledge an error in the calculation of seasonal variability. The seasonal variability was incorrectly based on the full period rather**

**than the reference period, which increases the importance of future sources of uncertainty such as GCM, SSP, and BCM (bias correction method). The results and discussion section will be updated using the corrected values.**

**Considering the updated Fig. 10, the future sources of uncertainty do not contribute to the uncertainty in seasonal contribution and variability. In both cases, the reference climate accumulated more than 60% of the feature importance, followed by the glacier inventory and the volume. There are not clear differences in these metrics between glacier runoff and melt. The seasonal shift, which is the absolute change in summer contribution (DJF) between the reference and future periods, shows a different pattern. For this metric, the reference climate shows a similar importance to the SSPs and GCMs. It is important to note that the greatest importance of the bias correction method is for this metric. Mechanisms that underscore the importance of the reference climate as a major source of uncertainty are likely to include its role in defining the baseline conditions against which future changes are assessed. In addition, the influence of the reference climate on temperature and precipitation patterns directly affects seasonal glacier response (melt / accumulation), contributing to its importance in determining seasonal metrics.**

**Discussion**

Overall, the discussion section needs some attention from the authors. This is not a study that can be easily replicated for other domains in terms of time commitment and computing resources needed. That makes it important to highlight the big picture findings, i.e., statements that provide insights on how to interpret the results for other regions (Are the findings applicable globally? If yes, then how was the conclusion reached? If not, then what are the differences or other cases that might need to be considered?).

**R: Both reviewers have raised concerns about the discussion section. To address this, we have proposed a new structure for this section. This revised structure aims to minimise repetition of results and to consolidate previous sections on uncertainty. In addition, we plan to incorporate the suggestions of both reviewers by focusing more on the implications of the results, rather than simply comparing them with other studies. This approach will improve the clarity and coherence of the discussion and allow readers to better understand the significance of our findings.**

**The proposed new structure is as follows: Section 5.1 ("Changing glacier hydrology") will present regional projections of changing glacier hydrology and compare them with results from previous studies. Section 5.2 ("Hydrological importance of data uncertainty") will discuss the hydrological importance of data uncertainty, summarising comparisons and highlighting the importance of sources of uncertainty, emphasising spatial differences and the importance of domain characteristics, as suggested by reviewer 2. In addition, the importance of model calibration is addressed, as suggested by Reviewer 1. Finally, Section 5.3 ("Limitations and global implications") discusses limitations such as unconsidered sources and potential global implications suggested by both reviewers.**

Also, for this specific domain, the authors should discuss the differences across the various hydrological zones e.g., how does domain characteristics in terms of climate, topography, glacier size, etc. affect the six sources of uncertainty and the ten runoff metrics?

**R: We appreciate the suggestion for the discussion. As part of the newly proposed Section 5.2, "Hydrological significance of data uncertainty", we will explore the spatial differences between different hydrological zones, with particular emphasis on climate. Given the hydrological importance of climate (Fig. 10) and its diversity in our study area, we will analyse the differences between historical reference climates and their impact on hydrological metrics. In addition, we will discuss the potential implications of our findings for other high mountain regions, which often face challenges in constraining gridded precipitation products due to a low density of gauging stations (Section 5.3). This will provide valuable insights into the spatial variability of hydrological dynamics and have implications for glacier modelling and water resource management in similar regions.**

Ln 484-485: This statement implies that the difference in acquisition year played a significant role in glacier area, while in Fig. 3d there is only a 1-year difference between the two RGI versions. Are you saying there was a large area loss between 2000 and 2001?

**R: In addition to the previous response regarding differences in glacier area between the inventories (results consistent with Zalazar et al. 2020), we will highlight that improved outlines and corrections from local inventories in RGI7 also contribute to the observed differences between inventories.**

Ln 486 – 488: The sentence from "*While the 69% of the total catchment ... estimates*" is unclear, please clarify and rephrase it. Also in Ln 488, what observational data are you referring to (glacier outlines or climate)?

**R: The specific sentence will be moved (and rephrased) to the results to avoid repetition in the discussion section: "In terms of glacier area, 69% showed a relative difference of less than ± 10% between the two glacier inventories, while only 27% showed differences below this threshold when considering both ice volume estimates"**

**In Ln 488, we are referring to ice thickness estimates. This will be rephrased for clarity: "The use of direct or indirect observations of ice thickness, such as those based on ground-penetrating radar or airborne surveys, can help to select a better dataset for the study area, potentially reducing the data uncertainty. However, these observations are spatially and temporally scarce in the Patagonian Andes."**

Ln 491 – 493: Not sure what this sentence is getting at.

**R: The specific sentence will be clarified and rephrased: "Furthermore, the generation of large-scale ice thickness datasets requires the compilation of numerous datasets derived from different acquisition dates, often spanning years or even decades, which hinders regional validation".**

Ln 537: What is meant by "*continue changing*"?

**R: The use of "continue changing" suggests an ongoing and evolving process, driven by the sustained loss of glacier mass (historical and projected) in the region. The sentence will be clarified.**

Ln 539: Just to clarify, 43% of the catchment will lose 80% of their glacial volume? Also, there is latitudinal dependence (North-South divide). Can you talk more about that?

**R: Yes, considering the mean derived from the full set of SSP scenarios (n = 1920), 18% of the total glacier area will lose more than 80% of its current volume this century (the use of % of catchment area will be removed from the manuscript to avoid adding unnecessary complexity to the study). The regional projections and their spatial differences will be discussed as part of the proposed new structure of the discussion.**

Ln 545: How was this conclusion reached? Is this talking about *Rounce et al. (2023)* or the current study?

**R: The conclusion was reached by comparing the uncertainty (i.e. error bars in Fig. S4) associated with the volume loss in 2100 between our study and Rounce et al. (2023). Although Rounce et al. (2023) only considered GCM and SSP as sources of data uncertainty, the error bars, which are calculated using one standard deviation, have surprisingly similar values in both cases (Fig S4). We recognize that this deduction requires further clarification, and we will provide additional context and explanation in the revised version of the conclusion.**

Ln 565-566: Statement regarding peak water already reached – can this be supported with observational data for the region (at least that can help understand the historical metrics between 1980 – 2023, if the data is available)?

**R: Thank you for the question. Ground-based validation is very important, but in high mountain areas the availability of ground-based data is generally limited, and the Patagonian Andes are no exception. Despite recent efforts during the development of PMET-obs (the ground-based alternative to PMET), only 9 (out of 109) catchments with stream gauges have more than 10% glacier area, making it difficult to properly isolate the glacier influence. Nevertheless, it is planned to use these data for hydrological modelling in an ongoing effort, which we expect to be available as a preprint this autumn (March - April).**

**Conclusions**

Much of this reads like results or discussion section. For example, the second bullet can be moved to results.

**R: We will re-evaluate the distribution of information to ensure a more coherent structure. The main findings of the study will be presented and highlighted in the results section, and the conclusions section will focus on summarising the key takeaways and implications of these findings.**

Ln 629: This study did not provide insights into "*local calibration choices*". As the authors rightly point out in the following sentence, future studies need to look into MB schemes, parameterizations, etc. This study is fairly comprehensive as it is, so the authors should try to highlight the significance of the work done and how it can be interpreted for other domains.

**R: Thanks for your comment. We agree with this point, and therefore we will replace "local calibration choices" by "impacts of data uncertainty".**

**Figures**

Fig. 1 and throughout: Can you please make the delineation between the 9 hydrological zones more prominent (e.g., in thick black line). The zone labels in 1b and c are not visible.

**R: We will change the transparency of the delineation lines in order to make them more prominent. We will also change the colours of the zone labels in Fig. 1b,c**

Also, it is better not to use sequential colormap for discrete categories. For example, it is hard to see the difference between <500 and <1500 in 1c. Readers should be able to extract this information quickly.

**R: Thanks for the suggestions. We will replace the discrete categories in b) and c) with sequential colormaps.**

Fig. 2: This flowchart is hard to follow, can you provide step numbers along the way (preferably in same sequence as the text from Ln 162).

**R: Thanks for the suggestion, we will add roman numerals in Fig. 2 to refer to the calibration steps in Section 3.1**

Fig. 3: Please consider using a more distinct diverging colormap for 3a, with white in the middle. It seems most of the regions are around yellow, so there is no (visible) difference between RGI 6 and 7. Also the bar colors in panel b are barely visible. Perhaps remove the grey background and make all the colors in darker shades.

**R: Thanks for the suggestions. We tried to use to white in the colormap of Fig 3b, but the contrast with the background was not adequate. We will adjust the range of the colormap from [-100, 100] to [-50, 50] to have more visible differences in the northern and southern zones. We will also add black outlines and a zero line to the bar plot in Fig 3b to prevent the bars being confused with the background.**

Fig. 4: For panel b, please see same comment as before. For panel a, consider a discrete colorbar with set ranges (0-20, 20-40, etc.). It's very hard to see the differences between the domains.

**R: Thank you for your suggestions. For Panel B, we will also adjust the range of the colour map (as in Fig. 3). We agree that the different order of magnitude makes it difficult to analyse the differences in volume between catchments. As an alternative to the proposed discrete colourmap, we will use a log scale for Panel A to address this issue.**

Fig. 5, 6, 7: Same as 4a. I do not see the dotted line for the mean value mentioned in the caption.

**R: The dotted lines were places within the boxplots to indicate the mean values. Considering that the violin plots already show the distribution of the values, we have decided to remove the dotted lines in the box plots.**

Fig. 6: The dots in panel a are not visible.

**R: Similar to Fig. 7, where the low agreement is indicated with black outlines, we will replace the dots in Panel A with black outlines ("The catchments with black outlines indicate low model agreement, where less than 80% of the models agree on the sign of the change.")**

---

## Author Response (AR1)

**Editor**

Dear Rodrigo Aguayo, and co-authors,

First of all I want to thank you and your co-authors for the dedicated and thorough point-by-point answers to all review comments and the respectively planned actions. In the following, I only want to highlight some of the major changes that you indicate.

- According to one major request by reviewer #1 (supported by reviewer #2), you plan to avoid the catchment terminology in your revised manuscript. I agree that keeping the glacier focus in your analysis will make you manuscript more accessible.

**R: We acknowledge the concerns raised about the introduction of "catchments" and the resulting confusion for the reader. We recognise the importance of clarity in our presentation, and therefore we agree that the use of "% of catchment area" in the comparison between the different sources of uncertainty may add unnecessary complexity to the study. Consequently, we have decided to remove these values from the manuscript, and we have changed the title of the manuscript to "Unravelling the sources of uncertainty in glacier runoff projections in the Patagonian Andes".**

**In addition, we have simplified the presentation in the revised version by i) focusing our analysis only on 'glacier area (at inventory date)', ii) presenting the relative differences between the different sources of uncertainty in terms of glacier area (%), and iii) adding glacier area-weighted averages to summarise the differences. These changes have been incorporated into Figures 5-7 and 10 and will contribute to a clearer and more concise presentation of our results. Finally, we have retained the level of aggregation of hydrological zones in order to examine spatial differences over a wide area encompassing different climates and geographical features (Figs. 8, 9, S1 and S2).**

- Both reviewers have requested a re-organisation of the discussion section to make it more coherent and concise. Both also request a focus on implications. You suggest a new structure for the discussion with 3 sub-sections. You briefly outline their content but it still remain rather vague.

**R: Thanks for the comment. In the revised version, the new structure aims to minimise repetition of results and to consolidate previous sections on uncertainty. In addition, we have incorporated the suggestions of both reviewers by focusing more on the implications of the results based on current limitations. This approach will improve the clarity and coherence of the discussion, and we hope that it will allow readers to better understand the significance of our results and the current challenges in assessing uncertainty in large-scale glacier models. The new structure of the discussion in the revised version is as follows:**

**- 5.1 Hydrological response of Patagonian glaciers to climate change: This section summarises the regional glacier projections and their associated uncertainties. In this section we have maintained the comparison of glacier runoff with results from previous studies.**

**- 5.2 Hydrological importance of data uncertainty: This section links the analysis of sources of uncertainty with their influence on different glacio-hydrological signatures, emphasising spatial differences due to domain characteristics, as suggested by reviewer 2.**

**- 5.3 Influence of model calibration: As the calibration approach varies between glacier models and has a direct influence on glacier discharge (through the precipitation factor), we have added a section to discuss this issue, as suggested by Reviewer 1.**

**- 5.4 Limitations and global implications: This section discusses limitations such as unconsidered sources and potential global implications suggested by both reviewers.**

- I am glad that you intend to follow the 2nd reviewer's suggestion on partitioning your uncertainty assessment into two over-arching categories: model inputs & forcing data. I am convinced that these categories will make it more easy to distill your main conclusions.

**R: In the revised version, we have incorporated the suggestion to partition our uncertainty assessment into two overarching categories. However, the proposed categories aggregate the sources of uncertainty by time period (historical and future data), instead of model inputs and forcing data. We believe that these categories are better suited to understanding how the historical sources of uncertainty are projected into the future, given that the primary objective of glacio-hydrological modelling studies has been to assess the future impacts of climate change.**

- Concerning your selection of 10 GCMs from the CMIP6 ensemble, your explanation refers to the AR6 decision criteria using global temperature metrics. Yet I wonder about the performance of the selected GCMs over South America. Please provide any information. Are they well suited for this region?

**R: Thanks for the question, it gives us the opportunity to examine the regional performance of the selected GCMs based on recent studies. In Chile, we have two studies that have investigated the adequacy of large ensembles of GCMs (Salazar et al., 2024; Gateño et al., 2024).**

**Given that raw GCM time series are bias-corrected using three different techniques as part of the sources of uncertainty, we will focus on Gateño's study, which proposes a framework that goes beyond bias-related metrics to include metrics related to the ability of GCMs to reproduce teleconnection responses to the El Niño Southern Oscillation (ENSO) and the Southern Annular Mode (SAM), which can affect regional climate variability and trends. For example, the SAM has shown a transition to a positive phase, causing a decrease in the intensity of mid-latitude westerlies, which generates a significant part of the decrease (increase) in precipitation in northern (southern) Patagonia.**

**Specifically, the GCM information in Table S2 was complemented with regional Past Performance Indexes for precipitation ($PPI_{PP}$) and temperature ($PPI_{T2M}$). In addition, we have included the selected GCMs from the screening approach based on the analysis of Gateño et al. All the selected GCMs showed an adequate performance in reproducing temperature, while only half of them showed a good performance for precipitation. Out of the 10 GCMs selected, 4 are included in the screening recommendation of Gateño et al.**

**(initial pool of 27 GCMs), highlighting the appropriateness of our selection. This information has been included in the discussion section as a recommendation to reduce the future overall uncertainty.**

**Salazar, Á., Thatcher, M., Goubanova, K. et al. CMIP6 precipitation and temperature projections for Chile. Clim Dyn 62, 2475–2498 (2024). DOI: 10.1007/s00382-023-07034-9**

**Gateño, F., Mendoza, P.A., Vásquez, N. et al. Screening CMIP6 models for Chile based on past performance and code genealogy. Climatic Change 177, 87 (2024). DOI: 10.1007/s10584-024-03742-1**

- Please also consider my initial comments from the access review in your revised manuscript. I am very pleased that you intend to include a paragraph on the influence of model calibration on your uncertainty analysis. In how far is the high importance of the historical sources of uncertainty a result of the model calibration. This comment is in line with request raised by reviewers #2.

**R: Thanks for the comment, we have added a new section to the discussion entitled "Influence of model calibration" (Section 5.3). In this section we address the complexities of calibrating large-scale glacier models. We highlight GloGEM's sequential calibration approach and its role in mitigating sensitivity to climate forcing and draw insights from regional studies to underline the importance of reference climate choices. With regard to the scaling effect of the precipitation factor on glacier runoff, our decision not to correct historical climate datasets aims to capture the full range of potential precipitation values while utilising existing bias correction procedures. In addition, we discuss the potential for future studies to refine uncertainty assessment through ensemble meteorological datasets and explore recent advances in model calibration techniques, including Bayesian inference methods.**

Altogether I gladly retain your manuscript for consideration in TC. Based on the brief summary of the main actions above, I suggest that your manuscript undergoes a second review round after you revised your manuscript according to your plans.

All the best,

The editor, Johannes Fürst

**Anonymous Referee #1**

In the manuscript "Assessing the glacier projection uncertainties in the Patagonian Andes (4-56 S) from a catchment perspective", a large set of glacier model simulations obtained with the OGGM model is used to assess the uncertainty in glacier runoff and glacier melt projections. The study looks at the effect of different glacier outlines, different glacier volume estimates, various historical climate datasets, different GCM, different emission scenarios and different bias correction methods. Each of these different datasets are discussed and it is shown how they vary, mainly focusing on their spatial patterns. Using a random forest regression method, the relative importance of each of these "model choice" uncertainties is examined. The study concludes that the reference climate is the most important source of uncertainty for a range of glacio-hydrological signatures, even for signatures that represent a signal beyond the reference period.

Overall, I think that the study presents an impressive amount of model run comparisons at a large regional scale and shows clear insights into spatial differences of glacier volume and runoff changes in the Patagonian Andes. Moreover, the effect of each source of uncertainty on the glacio-hydrological signatures nicely illustrates their variable importance for different aspects of the change in glacier runoff. However, I feel that some parts of the manuscript could be improved, such as the use of "catchments", and the discussion section, which should describe more the implications and possible hypotheses of the findings, rather than only a summary of the findings and comparison with other studies. Please find below a list of more detailed comments, also explaining these two examples.

**R: Thank you for your thoughtful and constructive feedback on our manuscript. We appreciate your positive assessment of the comprehensive model run comparisons and the insights they provide into spatial variations in glacier volume and runoff changes within the Patagonian Andes. We will carefully consider your suggestions for improvement, particularly regarding the use of "% of catchment area" and the discussion section. We recognise the importance of delving deeper into the implications of our results and proposing potential hypotheses, rather than simply summarising and comparing them with other studies.**

**To address these concerns, we have revised the discussion section to provide a more thorough exploration of the implications of our findings. In addition, we have chosen to prioritise the use of 'glacier area (at inventory date)' over '% of catchment area' throughout the manuscript to improve clarity for future readers. Detailed responses to these general comments are provided below.**

**Abstract**

The introduction of "catchments" starts here in the abstract. Without stating what "catchment" represents in this study, it is quite hard to follow. The reader doesn't know if the study is about a few catchments, or many, and how they are delineated/defined. Accordingly, all the statements with "xxx% of the catchment area" are difficult to interpret. On a more general note, it did not became clear why catchments are used in the study? Most of the results are rather described on the basis of the hydrological zones. In the study area description, it is noted "847 catchments were selected" without any further information. On the basis of what were these catchments defined? Since the study is not using any downstream information or non-glacierized catchment info ("only using glacierized grid cells"), I think the "catchment" part should not be part of this study. Results could then be just presented as xx% of glaciers show this and that, while also keeping the aggregation level of the hydrological zones.

**R: We acknowledge the concerns raised about the introduction of "catchments" and the resulting confusion for the reader. We recognise the importance of clarity in our presentation, and therefore we agree that the use of "% of catchment area" in the comparison between the different sources of uncertainty may add unnecessary complexity to the study. Consequently, we have decided to remove these values from the manuscript, and we have changed the title of the manuscript to "Unravelling the sources of uncertainty in glacier runoff projections in the Patagonian Andes".**

**In addition, we have simplified the presentation in the revised version by i) focusing our analysis only on 'glacier area (at inventory date)', ii) presenting the relative differences between the different sources of uncertainty in terms of glacier area (%), and iii) adding glacier area-weighted averages to summarise the differences. These changes have been incorporated into Figures 5-7 and 10 and will contribute to a clearer and more concise presentation of our results. Finally, we have retained the level of aggregation of hydrological zones in order to examine spatial differences over a wide area encompassing different climates and geographical features (Figs. 8, 9, S1 and S2).**

**Introduction**

L44 undergoing "a" shift – shouldn't shift not be an increase, as otherwise there is no increase in risk? Or does it relate to larger volumes that are stored before bursting out?

**R: The specific sentence on GLOFs has been removed to improve the continuity with the following paragraph.**

L65 "have had to be used" – have been used? Also check the "but" which is more an additional problem than a contrast?

**R: The specific sentence has been removed**

I would suggest to move table 1 to the SI. The abbreviations in the first column are not yet clear and in general the aim/message of the table did not become clear in the introduction

**R: Thanks for the comment. Table 1 has been moved to the Supplementary Material. The abbreviations in the first column have been defined in the table caption.**

L93 "to project the evolution of the glacier area" – the model is not only used for projecting the area?

**R: The specific sentence says, "to project the evolution of each glacier (area > 1 km$^2$)". The glacier model outputs used are not only the glacier area, but also the volume and hydrological outputs (glacier runoff and melt from formerly glacierised areas, see Section 3.4).**

**Study area**

L104-105 "crucial", "essential" in the same sentence, maybe one is enough?

**R: The sentence has been summarised: "the seasonal melting of glaciers is essential for the long-term sustainability of the local ecosystems and coastal human populations".**

**Methods**

Step 1 of the calibration procedure – I wondered if it should be discussed what the effect is of only calibrating the melt factor/temperature sensitivity using geodetic mass balances. If the precipitation is off (e.g. too little) then the temperature sensitivity might be smaller, to compensate?

**R: It is true that the calibrated temperature sensitivity parameter depends strongly on the assumed precipitation, and thus on the chosen precipitation factor (Pf; see e.g. Schuster et al. (2023). However, for most glaciers we have only one robust observation of mass change, i.e., the 20-year geodetic mass-balance average. Therefore, we decided to set the Pf to 1, as some regional climate datasets used in this study (e.g. PMET and CR2MET) already include a bias correction process to correct for potential precipitation underestimation, and therefore we expect the range of "true" precipitation to be covered by the different products. Based on this comment, the importance of the model calibration in large-scale glacier models has been added to the discussion (please see Section 5.3: Influence of model calibration).**

Step 2 – the description is a little bit unclear, is the gradient adjusted? Or what is meant with "a mass balance residual to add"?

**R: Sorry for the confusion, the gradient of the mass balance is not adjusted. In fact, the whole mass balance profile is shifted so that the resulting apparent mass balance is zero, to satisfy the equilibrium assumption. We have clarified this in the manuscript (step 2).**

Step 3 – maybe the equation could be given here? I think it is similarly known as equation 1, but helps the reader understanding what is needed for the inversion and at what scales it is applied

**R: Thanks for the suggestion. We think that the equations for the inversion are not that relevant for the paper and are already very well documented in Maussion et al. 2019. However, we have added the equation for the ice velocity in Section 3.1 to illustrate the control of the ice creep parameter A (which is calibrated in this step).**

Step 4 – what is a "constant mass balance run"?

**R: A constant mass balance run means that we define a mass balance over a period of time (e.g. the average mass balance of all the years in the period), this is our constant mass balance. Then we use this constant mass balance and let the glacier evolve for a few years. We have clarified this concept in step 4.**

Figure 2 – I think it would be helpful to add the number of the steps in the figure. And what is the arrow from reference climate to bed inversion? It may refer to one step before that?

**R: Thanks for the suggestion, we have added roman numerals in Fig. 2 to refer to the calibration steps in Section 3.1**

L259 "according to the location of the glacier terminus" – I am not sure to understand this addition? Maybe some more explanation of glaciers that are crossing catchment/hydrological zone borders could be added here

**R: Thanks for the comment. This has been clarified according to your suggestion: "For area and volume, we calculated the relative and absolute differences for each catchment and hydrological zone defined in Fig. 1. To calculate these differences, we aggregated glacier area and volume for a given catchment by selecting all glaciers with their terminus location within that catchment. It is assumed that, if the inventory outlines are correct, all the water flowing out of the glacier will flow via its terminus.".**

L291 here the text could benefit from some explanation/ careful discussion how streamflow metrics can be used to apply on aggregated glacier runoff data (i.e. glacier runoff is not the same as downstream streamflow and so their effect on the aquatic ecosystem is not 1:1 comparable)

**R: Thanks for the comment. We have added a sentence to highlight the limitations of our analysis: "However, our analysis of glacier runoff should not be considered as downstream streamflow because our simulations only considered the initially glacierised area and did not include the interaction with other hydrological fluxes (e.g., evaporation and infiltration)"**

Table 2 misses information about which period was used to calculate the signatures, apart from the ones that explicitly state "ref and future period"

**R: We have added a "Period" column for each signature.**

Header of 3.5 – maybe choose the titles of section 3.3 and 3.5 in such a way that it is more clear what their different content is

**R: The original header of 3.5 ("Uncertainty analysis") has been renamed to "Hydrological importance of sources of uncertainty"**

L299 – where comes 329 from?

**R: Out of 847 catchments, only 329 catchments have one or more glaciers in both inventories (RGI6 and RGI7). This has been clarified in the text.**

In the calculation of RMSE, what is the baseline? i.e. how is RMSE calculated?

**R: Thank you for your question. The permutation feature importance measures the change in model performance (in this case, the Root Mean Square Error; RMSE) after the values of a single model feature have been permuted (also known as shuffled), with more important features resulting in greater decreases in performance when permuted. The baseline corresponds to the model performance before the permutation of the model features (in this case, the six sources of uncertainty). This has been clarified in Section 3.5.**

**Results**

Figure 2: possibly remove the blue/gray background so the results are better visible

**R: We think that this comment refers to Fig 3, as there are no results in Fig 2. We have added black outlines and a zero line to the bar plot in Fig 3b to prevent the bars being confused with the background.**

L335 – why was the glacier thickness divided by the catchment area, while all other variables are focused on the glaciers/glacier grid cells?

**R: In the case of ice volume, which is derived from ice thickness and glacier outline, we decided to normalise the value by catchment area to facilitate the comparison between catchments - hydrological zones.**

L374 – maybe I missed it, but what are "the main catchments"?

**R: The main catchments correspond to the catchments that have an area greater than 5,000 km$^2$ (Fig. 1), which account for the 68% of the total catchment area. All maps have the following sentence to indicate the main catchments: "The names in grey correspond to the names of the main catchments"**

L408 "historical sources" – climate?

**R: Yes. This has been clarified in the text ("Considering the prolongation of historical climate conditions, 26% ± 9% of the total glacier ice is committed to melt in the long term")**

L435-L440 Does isolating the melt from glacier runoff result in more or less uncertainty? There is a hint that melt only has an effect of temperature, but it also states that precipitation can compensate for the change in melt? The 60% suggests that 40% of the glacier runoff is generated from off-glacier melt, or liquid precipitation on+off glacier?

**R: Compared to total runoff (Fig. S2), the uncertainty (standard deviation of the annual glacier melt) was reduced because the reference climate, which is the main source of climate uncertainty, had less influence, as the strongly past climate-dependent liquid precipitation is only included in the glacier runoff components (on- and off-glacier melt and on- and off-glacier liquid precipitation). This is mentioned in the subsection 5.1 ("Hydrological response of Patagonian glaciers to climate change"). Therefore, the 60% suggests that 40% of the glacier runoff is generated by off-glacier melt and on- and off-glacier liquid precipitation.**

**Discussion**

For the discussion in general, I found it sometimes hard to follow what was described in the different parts. The first part discusses the uncertainties, but the other two parts as well. Maybe by less repetition of the results, and more focusing on the implications of the results would help in restructuring the discussion. I think the comparison with other studies is mostly well described, but would also benefit from an additional thought on what are the implications of that it agrees well or not.

**R: Both reviewers raised concerns about the discussion section. To address this, we have proposed a new structure for this section. This revised structure aims to minimise repetition of results and to consolidate previous sections on uncertainty. In addition, we have incorporated the suggestions of both reviewers by focusing more on the implications of the results based on current limitations. This approach will improve**

**the clarity and coherence of the discussion, and we hope that it will allow readers to better understand the significance of our results and the current challenges in assessing uncertainty in large-scale glacier models.**

**The new structure of the discussion in the revised version is as follows:**

**- 5.1 Hydrological response of Patagonian glaciers to climate change: This section summarises the regional glacier projections and their associated uncertainties. In this section we have maintained the comparison of glacier runoff with results from previous studies.**

**- 5.2 Hydrological importance of data uncertainty: This section links the analysis of sources of uncertainty with their influence on different glacio-hydrological signatures, emphasising spatial differences due to domain characteristics, as suggested by reviewer 2.**

**- 5.3 Influence of model calibration: As the calibration approach varies between glacier models and has a direct influence on glacier discharge (through the precipitation factor), we have added a section to discuss this issue, as suggested by Reviewer 1.**

**- 5.4 Limitations and global implications: This section discusses limitations such as unconsidered sources and potential global implications suggested by both reviewers.**

I was wondering if there should be maybe a discussion on the way such glacier models are calibrated? If all model runs are calibrated equally well (RGI areas and glacier volumes are used for calibrating), then how come the results are so different, especially regarding reference climates? These parameters propogate in all other simulations, right? This must then come from processes that are not captured when looking at annual and long-term metrics only when calibrating model parameters? Or maybe there are other processes that are relevant to discuss for improving glacier modelling?

**R: Thanks for the comment, we have added a new section to the discussion entitled "Influence of model calibration" (Section 5.3). In this section we address the complexities of calibrating large-scale glacier models. We highlight GloGEM's sequential calibration approach and its role in mitigating sensitivity to climate forcing and draw insights from regional studies to underline the importance of reference climate choices. With regard to the scaling effect of the precipitation factor on glacier runoff, our decision not to correct historical climate datasets aims to capture the full range of potential precipitation values while utilising existing bias correction procedures. In addition, we discuss the potential for future studies to refine uncertainty assessment through ensemble meteorological datasets and explore recent advances in model calibration techniques, including Bayesian inference methods.**

L484 "acquisition dates" – this sentence could benefit from more explanation about what the difference is in both RGI outlines with respect to acquisition dates.

**R. Thanks for the comment. The specific sentence was removed from the discussion to avoid repetition with the results (subsection 4.1.1): "These regional differences may be due to several factors, including improved outlines and corrections from local inventories and differences in acquisition dates (Fig. 3d) …"**

L487 "this threshold" – do you mean a 10% threshold for volume?

**R: Yes, the sentence compared the relative differences between different sources of glacier inventory and ice thickness estimates. The specific sentence has been removed from the discussion in the revised version.**

L494 – Isn't this paragraph suggestion a contradiction? It would help starting the paragraph like that.

**R: The main topic of the specific paragraph was the uncertainty associated with the reference climate (in specific precipitation). In the revised version, the content of this paragraph has been summarised and included in the "Limitations and potential implications" subsection. Specifically, the content of this paragraph is included in the context of sources of uncertainty that can be reduced using ground-based data.**

L510 – a space needs to be inserted before "The selected…."

**R: Thanks for the catch. The specific sentence ("The selected GCMs showed a high agreement in most hydrological zones…") was removed from the discussion to avoid repetition with the results.**

L512 "the ten selected….of the assessed warming – not sure how this sentence fits in?

**R: The specific sentence has been removed from the discussion to avoid repetition of the methodology/results.**

L526 Why are the "older" estimates named after the newer estimate and referred to as "recently"?

**R: Thanks for the question. The "recently" was misplaced there and has been removed from the sentence.**

L539 "of its current volume" – although it is clear that the study only discusses glaciers, the confusing use of "catchments" means that here there "glacier" needs to be added

**R: Thanks for the comment. The reference to "catchments" has been removed from the sentence to clarify the source of the volume loss.**

L594 – possibly add that Mackay et al. deals with glacio-hydrological modelling in Iceland

**R: Thanks for the comment. We have added this to clarify the scope of the antecedent.**

**Conclusions**

L610 – "differences", "different", "varied" in one sentence – consider rephrasing.

**R: The specific sentence has been rephrased: "The six sources of data uncertainty showed differences of varying magnitude".**

L613 isn't it a separate point, the one on reference climate being most important for uncertainty?

**R: We decided to aggregate the main points of the conclusions based on subsections of the Results: i) Analysis of sources of uncertainty, ii) Hydrological importance of data uncertainty, and iii) Limitations and potential implications.**

L622 "tended to converge towards an overall decrease" – decrease of what?

**R: Decrease of glacier runoff and melt. This has been clarified in the conclusions.**

L625-628 double?

**R: Yes, the specific sentence indicated that regardless of the spatial scale of interest (catchment or glacier area), the main source of uncertainty was the reference climate. This sentence has been clarified as the "catchment area" has been removed from the Results and Discussion sections**.

L629 – what is meant with "local"?

**R: The specific word has been removed from the sentence.**

L634 "relative contribution of non-glacial water sources" – shouldn't future studies not focus on the dynamics of these sources, rather than the relative contribution? And what should future studies do with these non-glacial water sources? The follow up sentence does not directly fit here, i.e. there is some gap between knowing a relative contribution and understanding other catchment stores. Maybe a general sentence on extending the scope from glaciers to downstream hydrology fits better?

**R: Thanks for the suggestion. We agree that a general sentence about broadening the scope is more appropriate than focusing only on non-glacial water sources. In the revised version, we have modified this as suggested: "future studies should address sources of uncertainty not considered in this study (…) and extend the scope from glaciers to downstream hydrology. Downstream hydrology can play a critical role in the seasonal and interannual water release during dry seasons (Drenkhan et al., 2022), attenuating the consequences of glacier shrinkage (e.g., Somers et al., 2019)"**

**Anonymous Referee #2**

In this manuscript the authors quantify the effects of historical and future glacial and climatic conditions on simulated glacier melt in the Patagonian Andes using the Open Global Glacier Model (OGGM) and Random Forest Regression approach. They focus on six categories, or sources of uncertainty, namely glacier outline inventories, ice thickness, historical climate, GCMs, bias correction methods, and emission scenarios. They examine the importance of each source on ten runoff metrics (e.g., peak water year and magnitude, interannual variability, seasonal contribution and variability, etc.) and conclude that the choice of reference climate is the predominant source of uncertainty.

The authors have undertaken a fairly comprehensive assessment, exploring 1920 cases across the six categories, to identify how the choice of input and forcing data affects the outcome in a glacialhydrological modeling workflow. This is overall a well-written paper, and the figures support the narrative well. The authors have demonstrated the methods clearly and the results are presented in a logical way.

The areas which require considerable improvement are the Discussion and Results sections which lack coherency and do not tie the various pieces together. There are differences across the hydrological zones/catchments and the ten glacier runoff metrics which require more nuanced discussion. Further, the authors need to explicitly highlight the implication/s of this work and explain if/how the findings here can be used for other glaciated regions of the world (i.e., does this study domain encompass a comprehensive set of climatic, glacial, and hydrological conditions to make general deductions).

I recommend moderate revisions before the manuscript is ready for acceptance. These revisions mostly require either clarification or further elaboration in the text. Comments below are provided section-wise and are not in order of importance.

**R: Thank you for your detailed review of our manuscript. We appreciate your positive assessment and valuable feedback. We have addressed the issues raised, in particular regarding the coherence of the Discussion and Results sections. Furthermore, we have explicitly emphasised the implications of our findings and their relevance to glaciated regions worldwide. These revisions aim to improve the clarity and comprehensiveness of our work.**

**Introduction**

Ln 61-62: Can you clarify the statements related to "*significantly increased streamflow*" and "*significant trends*". I presume the latter refers to statistical significance and the former to a large magnitude increase? Also linking with the previous statement, is it possible to extract information from these past work on how much (%) the streamflow has increased because of the accelerated glacial mass loss?

**R: Thanks for the opportunity to clarify these points. As you mentioned, the "significant" refers to the existence of a statistical significance. This has been clarified in the text (only Pasquini et al. 2021 used a statistical test): "recent studies have reported increased flows in rivers with important glacier cover (Masiokas et al., 2019; Vries et al., 2023), some of which have only begun to show significant trends ($p < 0.001$) in the last decade (e.g., Santa Cruz; Pasquini et al., 2021)"**

**To our knowledge, there have been no comprehensive studies to date that would allow us to determine the percentage of streamflow that has increased as a result of accelerated glacier mass loss. We are addressing these important questions in an ongoing study that we expect to be available as a preprint this autumn.**

Ln 67-68: Is this referring to overestimation of precipitation? Also, what does "*diverged towards*" mean?

**R: Yes, this has been clarified in the introduction: "the different approaches and data sources have overestimated the precipitation according to numerical simulations of regional moisture fluxes."**

Ln 70 onward: I am somewhat unsure about the purpose of this table (and generally this paragraph). The table summarizes past mass balance assessments; however, the current study is not considering various MB schemes or calibration processes as a source of uncertainty. This table seems superfluous and if the authors want to keep it, please consider moving to the supplementary material. Instead, please expand on the two Patagonian glacier hydrological contribution studies (*Mernild 2017* and *Caro 2024*). That will link better with the preceding paragraph and the overall theme of the paper.

**R: Thanks for the suggestion. Table 1 has been moved to the Supplementary Material, and we have also added a few sentences about the two previous hydrological studies as suggested.**

Ln 89: "… *adding additional data to the calibration*…" What type of additional data is this referring to?

**R: In parenthesis, we have added the most frequent complementary data used by glacio-hydrological models (snow cover area and glacier mass change; Van Tiel et al., 2020)**

Ln 91: Please explicitly mention the six sources of uncertainty here, bringing information in Ln 94 – 96 earlier, and then mention the tools used (OGGM, random forest).

**R: Thanks for your comment. The specific paragraph has been reorganised according to your suggestions.**

Ln 91 onward: The preceding paragraph talks about different sources of uncertainty in the modeling chain in current literature (beyond what the authors have explored in this work). Various statements in this study imply that it focuses on comprehensive sources of uncertainties across the modeling chain. In reality, it only considers two sources i.e., model input data (glacier outlines, thickness) and atmospheric forcing (historical data, GCMs, climate scenarios, and bias-correction methods). I would not consider this encompassing "*uncertainty sources in the modeling chain*" (e.g., Section 5.1 heading). The full chain will include considerations such as model workflow, parameter space, MB or ice flow schemes, etc. (this is duly mentioned as a study limitation in Ln 517).

I recommend removing the term "glacier modeling chain" here and in other instances, including the abstract, and explicitly mention that various configurations of model input and forcing data are used to identify the dominant sources of uncertainty. This is an extensive assessment with these two broad categories as it is, so there is no need to overstate the study's scope.

**R: We appreciate the insightful feedback and acknowledge the concerns raised about the term 'glacier modelling chain'. After careful consideration, we agree that the term may not accurately reflect the scope of our study. We will remove the term 'glacier modelling chain' and explicitly clarify in various sections,**

**including the abstract, that our focus is on the impact of different sources of 'data' uncertainty. In the revised version, we have emphasised our assessment within the specified categories of model inputs (by time period, historical and future), without overstating the scope of the study.**

**Study area**

Ln 103: What is meant by pristine environment here? Is this referring to glaciers or the water resources?

**R: The specific sentence has been removed.**

Ln 111: These nine hydrological zones are referred to extensively throughout the paper. Please fully name all of them here.

**R: The nine hydrological zones have been defined in the indicated paragraph. Particular attention has been given to the northern zones of PPY and PCA which were only defined in Fig. 1.**

Ln 111: I understand how the nine hydrological zones were identified and demarcated, but can you please elaborate how the 847 catchments were selected. Was having a 0.1% glacier area of the total catchment area the only criteria? Why was this threshold selected?

**R: Thanks for the question. We selected 847 glacierised catchments, each with at least one glacier and a glacier area greater than 0.1%. The 0.1% was selected as a conservative threshold for drought buffering (please see Fig. 3 in Ultee et al. 2022; cited in main text). This clarification has been added to the study area section.**

Ln 113: Can you please explain "*high explanatory power of recent glacier change*" or reword this statement.

**R: The statement has been clarified: "that shown a strong capacity to reproduce recent glacier changes"**

Ln 123: "*are characterized by many small catchments*". Replace characterize by 'which hosts many small catchments' or something similar.

**R: Thanks for the suggestion. This has been changed as suggested ("which hosts many small catchments").**

**Methods**

Ln 142: Please mention the upper and lower thresholds here in brackets.

**R: The thresholds have been added to the sentence in brackets [0, 2 ºC]**

Ln 146: OGGM is a flowline model so what is meant by "local grid" of 10 – 200m? Is it referring to the spacing between cross-sections across the centerline?

**R: Yes, in the context of OGGM, the "local grid" refers to the spacing between grid points on the glacier surface. This grid is used to obtain the model glacier geometry by overlaying glacier inventory outlines and**

**NASADEM elevation data. The resolution of the grid varies with the glacier size, ranging from 10 to 200 m. Glaciers are then segmented into elevation bands following the algorithm proposed by Werder et al. (2020), each of which covers an elevation difference of 30 m. This procedure has been clarified in Section 3.1.**

Ln 151: If the precipitation factor is set a single value of 1, then it is no longer accounting for biases due to topography, missing processes etc. (Ln 137). Also, what does it mean to "… *assess influence of different reference climates* …" Is this referring to the temperature sensitivity parameter as the precipitation factor is set to 1 for all cases.

**R: Thanks for the comment. It is true that setting the precipitation factor (Pf) to 1 no longer accounts for the missing local processes, and therefore these processes are included in the modelling framework with the temperature sensitivity parameter. Regarding the second point, the statement also includes the historical precipitation of the reference climate dataset, as it influences, for example, the interannual variability of the glacier mass balance. The decision to use Pf = 1 is based on the fact that some of the regional climate datasets used in this study (e.g. PMET and CR2MET) already include a bias correction process to correct for potential underestimation of precipitation. Unlike many previous studies which strongly "correct" the input datasets before using them to drive the impact model, we chose to incorporate the uncertainty of the driving dataset in our evaluation. This has been added to the new discussion entitled ("Influence of model calibration").**

Ln 163: Replace "*new value*" with initial value or first guess.

**R: The "new value" has been replaced by "initial value"**

Ln 172: Is the simulated glacier volume set to match the input volume accumulated over each of the nine hydrological zones and not the individual 847 sub-catchments?

**R: Yes, please see the reply on Volume-Area scaling (VAS) below**

Ln 187: Remove "it" after compared.

**R: Thanks! The "it" has been removed.**

Ln 191: This might be a typographical error, what is N°1?

**R: Nº1 refers to the first step. This has been clarified ("If not, a new μ* is defined and the process starts again from the beginning").**

Ln 201: What does "*latitudinal patterns in terms of area*" mean?

**R: The specific sentence has been replaced by "both datasets show similar areas across different latitudes"**

Ln 209: It helps to keep the language simple. Can this be rephrased as M22 having 13% larger volume than F19… or something along those lines.

**R: Thanks for the suggestion. The sentence has been modified as suggested.**

Ln 210: "Both alternatives" - consider replacing with the 'two volume data sources' or something similar.

**R: Thanks for the suggestion. We have replaced "both alternatives" with "two volume data sources" in this paragraph.**

Ln 213: By dataset do you mean the 4 gridded products?

**R: This paragraph refers to the two ice volume datasets. To clarify this, we have replaced "datasets" with "volume data sources", as suggested in the previous comment.**

Ln 213: VAS was done for the 9 hydrological zones; can you explain here why not for the 847 individual catchments?

**R: Thanks for the question. Considering that most catchments have only a few glaciers, and the theoretical basis of VAS is determined for samples of glaciers spanning a wide range of sizes (and is validated by observations covering a wide range of sizes; Bahr et al. 2015), we decided to calculate the VAS parameters using the hydrological zones.**

**Bahr, David B., W. Tad Pfeffer, and Georg Kaser. "A review of volume-area scaling of glaciers." Reviews of Geophysics 53.1 (2015): 95-140.**

Ln 228: Can you please elaborate on why/how these specific GCMs were selected? Was this a subset from a larger initial pool and the 10 GCMs were selected based on their TCR and ECS?

**R: Thanks for the question. The initial pool consisted of all GCMs that have at least one realisation (r1i1p1f1) in all four emissions scenarios (SSP). We have added a sentence to clarify this point: "Considering only GCMs with at least one output in all emission scenarios, the selection of the 10 GCMs was based on the recommendations of Hausfather et al. (2022), who suggest focusing on a subset of GCMs that are most consistent with the assessed warming projections of the Sixth Assessment Report (AR6)."**

**In the revised version of the discussion, we have also discussed the suitability of the GCM to reproduce seasonal cycles, monthly probabilistic distributions, spatial patterns of climatological means, and the ability of GCMs to reproduce teleconnection responses based on the analysis of Gateño et al. (2024). For more details, see the Editor's comment.**

**Gateño, F., Mendoza, P.A., Vásquez, N. et al. Screening CMIP6 models for Chile based on past performance and code genealogy. Climatic Change 177, 87 (2024). DOI: 10.1007/s10584-024-03742-1**

Ln 234 – 236: This is rather confusing: all GCMs have ECS falling in "very likely" range but only 80% in "likely"? Is the very likely range broader than the likely range? The "likely' range in *Hausfather22* was narrower (1.4 - 2.2 C).

**R: Thanks for the comment, it gives us the opportunity to clarify this point. Hausfather et al. 22 recommended screening out models with a TCR that lies outside the 'likely' range of 1.4–2.2 °C, or alternatively using a "likely" ECS of 2.5-4°C, which also reproduces the AR6 results well. In view of these**

**alternatives, the sentence relating to ECS has been removed from the main text to improve the clarity of the GCM selection.**

Ln 257: It is somewhat unclear where, when, and why catchments and hydrological zones are considered separately, e.g., volume-area scaling was done at the hydrological zone scale, glacio-hydrological signatures were assessed at catchment scale (Ln 288), etc.

**R: Thanks for the comment. Our initial intention was to conduct the complete study using the catchment scale only. However, in order to present the glacio-hydrological projections in Section 4.2 (Fig 8, 9 + Fig S1 and S2) we needed a clear level of aggregation (i.e., hydrological zones). Additionally, the theoretical basis of VAS is determined for samples of glaciers spanning a wide range of sizes, and therefore, the proposed zones were more appropriate than the catchments (please see the previous response about VAS procedure).**

Ln 259: Not sure I understand what is meant by area/volume aggregation based on terminus location.

**R: Thanks for the comment. This has been clarified in the specific paragraph: "For area and volume, we calculated the relative and absolute differences for each catchment and hydrological zone defined in Fig. 1. To calculate these differences, we aggregated glacier area and volume for a given catchment by selecting all glaciers with their terminus location within that catchment. It is assumed that, if the inventory outlines are correct, all the water flowing out of the glacier will flow via its terminus".**

Ln 261: This lapse rate was used because it is default in OGGM and also used within this domain in the past studies (Ln 141). Table 1 however also shows that literature used different lapse rates depending on the region, e.g., 6.5 for NPI and 5.8 for GCN. Is there a north-to-south gradient in the mean annual lapse rates? Does that affect the downscaling of ERA5 data? Also, is this downscaling step the same as what is mentioned in Ln 141? What about total precipitation, how was that handled when going from a quarter degree to 0.05° in ERA5?

**R: Thank you for the question, it gives us the opportunity to elaborate on our decision to continue using a constant environmental lapse rate of 6.5 °C km$^{-1}$ to downscale the temperature. Although recent studies in Patagonia have demonstrated the variability of this value (Bravo et al. 2019 in SPI), recent efforts during the development of the PMET (Aguayo et al. 2024; citation has been updated in the text) have shown that seasonal and regional variations of this value in Patagonia do not significantly improve performance (results based on > 100 meteorological stations). Using a lapse rate of 6.5 °C km$^{-1}$, all reference climates used in this study were able to achieve mean biases of less than 1 °C (Fig. 5 in Aguayo et al. 2024). No downscaling method was used for precipitation (this has been added to the main text).**

**Bravo, C., Quincey, D. J., Ross, A. N., Rivera, A., Brock, B., Miles, E., & Silva, A. (2019). Air temperature characteristics, distribution, and impact on modeled ablation for the South Patagonia Icefield. Journal of Geophysical Research: Atmospheres, 124(2), https://doi.org/10.1029/2018JD028857, 907-925.**

**Aguayo, R., León-Muñoz, J., Aguayo, M., Baez-Villanueva, O. M., Zambrano-Bigiarini, M., Fernández, A., and Jacques-Coper, M.: PatagoniaMet: A multi-source hydrometeorological dataset for Western Patagonia, Sci Data, 11, 6, https://doi.org/10.1038/s41597-023-02828-2, 2024.**

263: Please mention these default parameters either here or in supplementary material. I suppose Ln 153-154 mentions a couple of these, but please enlist all the parameters for reproducibility. Model "defaults" can change over time as new information becomes available.

**R: The parameters used in this study have been added ($T_{melt}$ = -1 ºC, $T_{solid}$ = 0 ºC and $T_{liquid}$ = 2 ºC).**

Ln 264, 273, 361, etc.: "*Glacierized grid cells*" – to confirm, these grid cell-based computations are only done for input & forcing data because OGGM is a flowline model and the output information (area, volume, melt) will be for a specific glacier?

**R: Yes, OGGM uses the climate data from the nearest grid cell to generate outputs for each glacier. In the case of air temperature, the raw climate data is downscaled to the glacier surface (elevation bands) using a constant lapse rate.**

Ln 275: To clarify, when uncertainty metrics are computed all glaciers are considered (Ln 256), but for hydrological assessments only this subset is considered? Section 4.1 is for all the glaciers in RGI and Section 4.2 is for these ~2000 glaciers?

**R: Yes, you are correct. Due to computational limitations, we used the OGGM model to estimate the evolution of only the glaciers with an area > 1 km² (~2,000 glaciers) (Section 3.4). The comparative analysis (Section 3.2) was performed considering all glaciers in RGI6 and RGI7.**

Ln 278 – 279: Up till this point, I do not think historical (outline, volume, ref. climate) vs future (scenarios, GCMs, BCM) sources are explicitly mentioned and separated in the text. So, the "*2 . 2 . 4*" reference is not clear.

**R: Thanks for the comment. We agree that it is important to explicitly mention this here: "For each glacier, we evaluated 16 scenarios generated by the historical sources of uncertainty: glacier outlines (n = 2), volume datasets (n = 2) and reference climates (n = 4). These scenarios were used to project the future evolution given by different GCMs (n = 10), future scenarios (n = 4), and bias correction methods (n = 3), resulting in 120 future scenarios for each historical simulation (a total of 1920 potential scenarios; Fig. 2)"**

Ln 283-284: I understand on- and off-glacier liquid precipitation, and "on-glacier" melt. But what is off glacier melt? Is it referring to snow on non-glaciated areas? Also, the term on-glacier melt is unusual, I believe this is referring to direct melt from glacier?

**R: The melt off-glacier melt corresponds to snow melt on areas that are now glacier-free (i.e. 0 at the start of the simulation; in our case 1980). The term "melt on glacier" has been renamed to glacier melt and clarified in this text: "we also extracted the melt on glacier (hereafter glacier melt), which is the sum of ice and seasonal snow melt on the glacier (Fig. 2c)".**

Ln 284: It seems this line and onward is now referring to the full 1920 scenarios and not the 16 historical ones? How is climate uncertainty influence being overestimated? What is meant by 'climate' here, and what is precipitation reduction?

R: Thanks for the questions. We have divided the specific paragraph into two: one for the scenarios and the following for the variables extracted from OGGM. The second paragraph will start with: "For all 1920 scenarios, we extracted…"

Regarding the second point, this has been clarified in the text: "To disaggregate the impact of projected precipitation changes, we also extracted the melt on glacier (hereafter glacier melt)…"

I suppose "glacier runoff" here is the sum of glacial melt and liquid precipitation on glacier? Please consider rephrasing the 'melt on glacier' with an equivalent term commonly used in literature.

R: Yes, at the beginning of the simulation (i.e., year 1980) glacier runoff is the sum of glacier melt and liquid precipitation. As the glacier retreats, off-glacier melt (seasonal snow) and liquid precipitation are also considered (glacier runoff corresponds to all water originating from the initially glacierized area).

The term "melt on glacier" has been renamed to glacier melt and clarified in this text: "we also extracted the melt on glacier (hereafter glacier melt), which is the sum of ice and seasonal snow melt on the glacier (Fig. 2c)".

Ln 286: Is this referring to the melt and precipitation time series? Also, I do not understand the meaning of 'according to glacier terminus' here.

R: In the previous responses, we have defined and clarified the glacier runoff and melt time series. The glacier terminus reference point will be clarified in Section 3.3, and a reference to that clarification has been included here: "As in the comparative analysis (Section 3.3), the time series were initially aggregated at the catchment scale according to the location of the glacier terminus".

Ln 288: Now this assessment is at catchment scale and not hydrological zone scale, again it gets fuzzy where hydrological zones vs catchments are considered. Also, aren't SSP-based scenarios 4, and n=1920 the total number of scenarios?

R: As mentioned above, our original intention was to conduct the entire study at the catchment scale only. However, in order to present the glacio-hydrological projections in Section 4.2 (Figs. 8, 9 + Figs. S1 and S2), we needed a clear level of aggregation (i.e. hydrological zones). In addition, the theoretical basis of VAS is determined for samples of glaciers covering a wide range of sizes, and therefore the proposed zones were more appropriate than catchments. On the other hand, in order to identify clear patterns using the permutation feature importance of RF regression models, we need to have a representative number of samples greater than just 9 hydrological zones. We hope that this clarification will help to understand our decisions that prevented us from using only one scale of analysis.

Results

Ln 321-322: Can you please make the statements explicit here, e.g., RGI7 has 4% and 15% greater area than RGI6. The "*showed positive/negative difference*" is perhaps not the best way to state this.

**R: The specific sentence has been rephrased: "showed increases ranging from 4% to 15% relative to RGI6".
We agree that "showed positive/negative difference" is not the clearest way to show the differences and
therefore similar sentences have been rephrased.**

Ln 325: There is only a 1–year difference in the acquisition date for most glaciers from RGI6 to RGI7 (Fig. 3c) –
did that make such a large difference in terms of area reduction?

**R: Although a detailed comparison between the different sources of data uncertainty is beyond the scope of
this study, the differences reported here are similar to the detailed comparison made by Zalazar et al. (2020)
(see Fig. 11 there; cited in the main text). For example, Zalazar et al. (2020) showed that the largest absolute
differences are in the Patagonian Icefields, and some (one-degree) latitudinal bands can show relative
differences of more than 50% in the Patagonian Andes. It should be noted that improved outlines and
corrections from local inventories in RGI7 also contribute to the observed differences in glacier area.**

Ln 335: Just for clarification, both M22 and F19 have the same glacier outlines taken from RGIv6?

**R: Yes, in order to make a proper comparison we use RGI6 in both ice volume datasets. This was mentioned
in caption of Fig. 4 and has been added to the indicated sentence.**

Ln 336: Fig. 4b is the M22 minus F19, correct? Most of the area is in yellow shades visually so M22 shows less
volume. Perhaps it is the colorbar that needs to be changed.

**R: Yes, that's correct. Panels B and C are differences in percentage relative to F19 ((M22 - F19) / F19). We
have adjusted the range of the colormap from [-100, 100] to [-50, 50] to have more visible differences.**

Ln 351: Replace "in" with "over 51% of the glaciated area".

**R: Changed as suggested.**

Ln 352: Throughout the paper, the reference to glacier/glaciated area and catchment area together is very confusing,
for example this line mentions 51% of glacier area, 22% of the catchment area for the precipitation reference and
95% of the glacier area, 99% of the catchment area for temperature reference. Please consider presenting this
information in some other way (perhaps focus on one of these only).

**R: Thanks for the comment. This was one of the general comments of Reviewer 1. The percentages related
to catchment area have been removed from the main text in order to the information clearer to the future
reader.**

Ln 356: Similar patterns in terms of spatial patterns?

**R: Thanks for the question. The specific sentence has been removed"**

Ln 356: The spatial resolution of the native data is quite different (0.25 for ERA5, 0.05 for PMET/CR2MET), does
that come into play at all?

**R: Please see our previous response about the spatial resolution of the climate data (use of lapse rates)**

Ln 367: Also, are there other characteristics that create these latitudinal differences in (a) precipitation change sign and (b) model agreement/disagreement?

**R: The projected changes in precipitation follow a latitudinal pattern which generates a low agreement in the intermediate zone (SPI and GCN) due to the fact that it is a transitional zone ($\Delta PP = 0\%$). These projections are consistent with historical trends that has been attributed to the Southern Annular Mode (SAM), an index that describes the movement of the low-pressure belt that generates westerly winds (Fogt et al. 2020). This index has recorded a transition to a positive phase, causing a decrease in the intensity of westerly winds at mid-latitudes, which generates a significant part of the decrease (increase) in precipitation in northern (southern) Patagonia.**

**Fogt, Ryan L., and Gareth J. Marshall. "The Southern Annular Mode: variability, trends, and climate impacts across the Southern Hemisphere." Wiley Interdisciplinary Reviews: Climate Change 11.4 (2020): e652.**

Ln 370: Why/how is ice volume relevant to climate projections? This is a rather obtuse statement, talking about GCM model disagreement and then ice volume estimates.

**R: The specific sentence ("areas characterized by a high ice volume") has been removed from the paragraph.**

Ln 374: What do you mean by the "main catchments".

**R: The main catchments correspond to the catchments that have an area greater than 5,000 km² (Fig. 1). All maps have the following sentence to indicate the main catchments: "The names in grey correspond to the names of the main catchments"**

Ln 405: What do you mean by the prolongation of the mass loss? Is this referring to Fig. 8b?

**R: Yes, but more generally to the whole of Fig. 8 (all panels refer to volume loss), which has been clarified in the specific sentence (volume instead of mass): "Projections from OGGM indicate that the glacier volume loss of recent decades will continue (Fig. 8)."**

Ln 406: Again, please reconsider the discussion related to catchment and glacier area. It is surprising that 18% of glacier area is equivalent to 43% of the catchment area. Also, how is catchment losing volume (it is only the glaciers that will be losing the volume).

**R: Thanks for the comment. This was one of the general comments of Reviewer 1. The percentages related to catchment area have been removed from the main text in order to the information clearer to the future reader. In this case, the difference between 18% and 43% is explained by the fact that the glaciers are located in the northern area which is characterised by the largest catchments (Petrohue, Puelo, Yelcho, Palena, etc.)**

Ln 417: Is this the maximum range across all the basins? It seems some basins have less or more spread across the scenarios (e.g., SPI-N). Again, can you comment on the difference between the hydrological zones here or in discussion.

**R: Thanks for the question. The referred differences in NPI-E between contrasting scenarios are not the maximum across hydrological zones (see Table R1). The greatest differences are found in the SPI (e.g., NPI-N and SPI-C) as you noticed. To simplify the discussion in the revised version, we have omitted the specific example of NPI-E and focused instead on broader patterns across hydrological zones (please see section 5.2).**

**Table 1. Mean specific mass balance for the period 2070-2099 for different hydrological zones.**

| Hydrological zone | Specific mass balance (kg m$^{-2}$) | |
|---|---|---|
| | SSP1-2.6 | SSP5-8.5 |
| PCA | -362 | -1524 |
| PPY | -312 | -1491 |
| NPI-E | -476 | -1712 |
| NPI-W | -1484 | -3083 |
| SPI-N | -1828 | -4196 |
| SPI-C | -1203 | -2914 |
| SPI-S | -794 | -1996 |
| GCN | -683 | -1643 |
| CDI | -313 | -955 |

Ln 427: 61% of the total catchment area contains 30% of the total glacier area i.e., the remaining 70% of the glaciated area in the ~40% of the catchment area? Here it is best to talk about glacier area (because it is talking about glacial melt), catchment area reference is somewhat misleading.

**R: We agree that the catchment area could be misleading, so the reference to catchment area has been removed as previously mentioned.**

Ln 428: Is this melt now talking about all the glaciated area in the full study domain?

**R: Yes, this has been clarified in the specific sentence.**

Ln 430: Please consider rephrasing "the evolution of the melt on glacier…". Do you mean changes in melt rates?

**R: Thanks for the question. This has been clarified in the main text: "The projected trajectories of glacier melt varied slightly among emissions scenarios"**

Ln 435: This is referring to Fig. 9e for SPI, correct?

**R: Yes, the reference to the panel figure has been corrected.**

Ln 436: Which panel of Fig. 5 is this referring to? Please rephrase "*melt on glacier evolution*".

**R: The reference to Fig. 5 was incorrectly located there and has been removed.**

Ln 455: Remove comma after 'contribution'.

**R: Thanks for the comment. The comma after 'contribution' has been moved before 'metrics': "This was especially clear for (…) metrics, where the reference climate accumulated…"**

Ln 468: What is meant by lower importance of climate here? Reference climate?

**R: Yes, this has been clarified in the revised version ("The lower importance of the reference climate was limited to…").**

Ln 470: Can you explain why there are differences between seasonal contribution, variability and shift in Fig. 10? What are the mechanisms causing these differences in sources of uncertainty.

**R: Before addressing the differences observed in Fig. 10, it is important to acknowledge an error in the calculation of seasonal variability. The seasonal variability was incorrectly based on the full period rather than the reference period, which increases the importance of future sources of uncertainty such as GCM, SSP, and BCM (bias correction method). The results have been updated in the revised version.**

**Considering the updated Fig. 10 in the revised version, the future sources of uncertainty do not contribute to the uncertainty in seasonal contribution and variability. In both cases, the reference climate accumulated more than 60% of the importance, followed by the glacier inventory and the volume. There are not clear differences in these metrics between glacier runoff and melt. The seasonal shift, which is the absolute change in summer contribution (DJF) between the reference and future periods, shows a different pattern. For this metric, the reference climate shows a similar importance to the SSPs and GCMs. Mechanisms that underscore the importance of the reference climate as a major source of uncertainty are likely to include its role in defining the baseline conditions against which future changes are assessed. In addition, the influence of the reference climate on temperature and precipitation patterns directly affects seasonal glacier response (melt / accumulation), contributing to its importance in determining seasonal metrics. This analysis has been included in Section 5.2 ("Hydrological importance of data uncertainty")**

**Discussion**

Overall, the discussion section needs some attention from the authors. This is not a study that can be easily replicated for other domains in terms of time commitment and computing resources needed. That makes it important to highlight the big picture findings, i.e., statements that provide insights on how to interpret the results for other regions (Are the findings applicable globally? If yes, then how was the conclusion reached? If not, then what are the differences or other cases that might need to be considered?).

**R: Both reviewers raised concerns about the discussion section. To address this, we have proposed a new structure for this section. This revised structure aims to minimise repetition of results and to consolidate previous sections on uncertainty. In addition, we have incorporated the suggestions of both reviewers by focusing more on the implications of the results based on current limitations. This approach will improve the clarity and coherence of the discussion, and we hope that it will allow readers to better understand the significance of our results and the current challenges in assessing uncertainty in large-scale glacier models.**

**The new structure of the discussion in the revised version is as follows:**

**- 5.1 Hydrological response of Patagonian glaciers to climate change: This section summarises the regional glacier projections and their associated uncertainties. In this section we have maintained the comparison of glacier runoff with results from previous studies.**

**- 5.2 Hydrological importance of data uncertainty: This section links the analysis of sources of uncertainty with their influence on different glacio-hydrological signatures, emphasising spatial differences due to domain characteristics, as suggested by reviewer 2.**

**- 5.3 Influence of model calibration: As the calibration approach varies between glacier models and has a direct influence on glacier discharge (through the precipitation factor), we have added a section to discuss this issue, as suggested by Reviewer 1.**

**- 5.4 Limitations and global implications: This section discusses limitations such as unconsidered sources and potential global implications suggested by both reviewers.**

Also, for this specific domain, the authors should discuss the differences across the various hydrological zones e.g., how does domain characteristics in terms of climate, topography, glacier size, etc. affect the six sources of uncertainty and the ten runoff metrics?

**R: We have considered your suggestion and incorporated it into our discussion. Specifically, in Section 5.2 (Hydrological importance of data uncertainty), we have analysed the variations across different hydrological zones within the Patagonian Andes (Figs. S3 and S4). Our results show that, despite significant variability in climate, geography and glacier characteristics across the region, the influence of reference climate conditions is limited to specific zones. In the revised version, we have analysed these specific zones and the factors that could explain the greater importance of future sources of uncertainty, such as GCMs and SSPs.**

Ln 484-485: This statement implies that the difference in acquisition year played a significant role in glacier area, while in Fig. 3d there is only a 1-year difference between the two RGI versions. Are you saying there was a large area loss between 2000 and 2001?

**R: In addition to the previous response regarding differences in glacier area between the inventories (results consistent with Zalazar et al. 2020), we have highlighted in Section 4.1.1 that improved outlines and corrections from local inventories in RGI7 also contribute to the observed differences between inventories.**

Ln 486 – 488: The sentence from "*While the 69% of the total catchment … estimates*" is unclear, please clarify and rephrase it. Also in Ln 488, what observational data are you referring to (glacier outlines or climate)?

**R: The specific sentence has been removed and indirectly included in the discussion of the hydrological importance of data uncertainty (Section 5.2): "Despite the larger relative differences in glacier volume than in glacier inventories (11.1% vs. 4.0% of overall difference; Fig. 3 and 4), the selection of glacier inventory was more important for most metrics (Fig. 10)".**

**In Ln 488, we are referring to ice thickness estimates. This has been rephrased for clarity (and moved to subsection 5.3): "The use of observations of ice thickness, such as those based on ground-penetrating radar or airborne surveys, can help to select a better dataset for the study area. However, these observations are spatially and temporally scarce in the Patagonian Andes."**

Ln 491 – 493: Not sure what this sentence is getting at.

**R: The specific sentence has been clarified and rephrased: "Furthermore, the generation of large-scale ice thickness datasets requires the compilation of numerous datasets derived from different acquisition dates which hinders regional validation (subsection 5.3)".**

Ln 537: What is meant by "*continue changing*"?

**R: The use of "continue changing" suggests an ongoing and evolving process, driven by the sustained loss of glacier mass (historical and projected) in the region. The sentence has been removed in the revised version.**

Ln 539: Just to clarify, 43% of the catchment will lose 80% of their glacial volume? Also, there is latitudinal dependence (North-South divide). Can you talk more about that?

**R: Yes, considering the mean derived from the full set of SSP scenarios (n = 1920), 18% of the total glacier area will lose more than 80% of its current volume this century (the use of % of catchment area has been removed from the manuscript to avoid adding unnecessary complexity to the study).**

**Regarding the latitudinal dependence, it's interesting to note that while there are clear latitudinal patterns in precipitation projections (Fig. 6), no clear latitudinal pattern was observed in volume loss (Fig. R1). This suggests that volume loss is influenced by a variety of factors beyond just latitude-dependent variables. Initial volume, ice dynamics and climatological features all play an important role in shaping patterns of ice loss. This underscores the complexity of the processes driving changes in ice volume and highlights the need for a comprehensive understanding of the different sources of uncertainty.**

[Figure]

**Figure R1. Volume loss (in percentage) vs. Latitude. The size of the circles is proportional to the glacier volume.**

Ln 545: How was this conclusion reached? Is this talking about *Rounce et al. (2023)* or the current study?

**R: The conclusion was reached by comparing the uncertainty (i.e. error bars in Fig. S5) associated with the volume loss in 2100 between our study and Rounce et al. (2023). Although Rounce et al. (2023) only considered GCM and SSP as sources of data uncertainty, the error bars, which are calculated using one standard deviation, have surprisingly similar values in both cases (Fig S5). Nevertheless, we recognise that this statement requires more detailed analysis, given that Rounce et al. account for uncertainty in model parameters, and we have therefore removed this inference from the discussion.**

Ln 565-566: Statement regarding peak water already reached – can this be supported with observational data for the region (at least that can help understand the historical metrics between 1980 – 2023, if the data is available)?

**R: Thank you for the question. Ground-based validation is very important, but in high mountain areas the availability of ground-based data is generally limited, and the Patagonian Andes are no exception. Despite recent efforts during the development of PMET-obs (the ground-based alternative to PMET), only 9 (out of 109) catchments with stream gauges have more than 10% glacier area, making it difficult to properly isolate**

**the glacier influence. Nevertheless, it is planned to use these data for hydrological modelling in an ongoing effort, which we expect to be available as a preprint this summer (June - July).**

**Conclusions**

Much of this reads like results or discussion section. For example, the second bullet can be moved to results.

**R: We have re-evaluated the main points of the conclusions to ensure a more coherent structure. In the revised version, the main findings of the study are presented in the results section, and the conclusions section focuses on summarising the key findings, with one point per subsection of the Results.**

Ln 629: This study did not provide insights into "*local calibration choices*". As the authors rightly point out in the following sentence, future studies need to look into MB schemes, parameterizations, etc. This study is fairly comprehensive as it is, so the authors should try to highlight the significance of the work done and how it can be interpreted for other domains.

**R: Thanks for your comment. We agree with this point, and therefore we have replaced "local calibration choices" by "impacts of data uncertainty".**

**Figures**

Fig. 1 and throughout: Can you please make the delineation between the 9 hydrological zones more prominent (e.g., in thick black line). The zone labels in 1b and c are not visible.

**R: We have changed the transparency of the delineation lines in order to make them more prominent. We have also changed the colours of the zone labels in Fig. 1b,c**

Also, it is better not to use sequential colormap for discrete categories. For example, it is hard to see the difference between <500 and <1500 in 1c. Readers should be able to extract this information quickly.

**R: Thanks for the suggestion. We have replaced the discrete categories in (b,c) with sequential colormaps.**

Fig. 2: This flowchart is hard to follow, can you provide step numbers along the way (preferably in same sequence as the text from Ln 162).

**R: Thanks for the suggestion, we have added roman numerals in Fig. 2 to refer to the calibration steps in Section 3.1**

Fig. 3: Please consider using a more distinct diverging colormap for 3a, with white in the middle. It seems most of the regions are around yellow, so there is no (visible) difference between RGI 6 and 7. Also the bar colors in panel b are barely visible. Perhaps remove the grey background and make all the colors in darker shades.

**R: Thanks for the suggestions. We tried to use to white in the colormap of Fig 3b, but the contrast with the background was not adequate. We have adjusted the range of the colormap from [-100, 100] to [-50, 50] to**

**have more visible differences in the northern and southern zones. We have also added black outlines and a zero line to the bar plot in Fig 3b to prevent the bars being confused with the background.**

Fig. 4: For panel b, please see same comment as before. For panel a, consider a discrete colorbar with set ranges (0-20, 20-40, etc.). It's very hard to see the differences between the domains.

**R: Thank you for your suggestions. For Panel B, we have also adjusted the range of the colour map (as in Fig. 3). We agree that the different order of magnitude makes it difficult to analyse the differences in volume between catchments. As an alternative to the proposed discrete colourmap, we have used a log scale for Panel A to address this issue.**

Fig. 5, 6, 7: Same as 4a. I do not see the dotted line for the mean value mentioned in the caption.

**R: The dotted lines were places within the boxplots to indicate the mean values. Considering that the violin plots already show the distribution of the values, we have decided to remove the dotted lines in the box plots.**

Fig. 6: The dots in panel a are not visible.

**R: Similar to Fig. 7, where the low agreement is indicated with black outlines, we have replaced the dots in Panel A with black outlines ("The catchments with black outlines indicate low model agreement, where less than 80% of the models agree on the sign of the change.")**

---

## Referee Report (RR1)

The manuscript provides an in-depth evaluation of the uncertainty in glacio-hydrological modeling of the Patagonian Andes using the Open Global Glacier Model (OGGM). Through an extensive series of simulations (1920 in total), the authors have quantified how various model choices influence glacier melt and runoff projections. They examined six different model choices (referred to as sources of uncertainty), that relate to both historical datasets (different glacier outlines, glacier volumes, and reference climate data) and future climate forcings (various general circulation models, emission scenarios, and bias correction methods). Additionally, the relative importance of each source of uncertainty was evaluated using a random forest regression method. The analysis revealed that reference climate data is the most critical source of uncertainty, even for metrics related to future projections.

The authors have caried out a rather comprehensive assessment of model uncertainty, which is adequately documented in this manuscript. In response to the feedback during the first round of revision, the authors have reorganized the discussion section and improved the figures and general clarity throughout the text.

I deem the manuscript fit for publication after a minor, mainly textual, revision. Please consider the more detailed list of suggestions below.

**Abstract**

L24: area > 1km2 → with area > 1km2

L25: Here, I would suggest removing the mention of the number of catchments and hydrological zones. It does not add to the clarity of the abstract and it is not of key importance.

L25: Consider replacing "We used different glacier […]" by "As sources of uncertainty, we used different glacier […]" to prevent confusion about what is meant by *each source* in L30.

L30: "We used the permutation feature importance of random forest regression models to assess the relative importance of each source on the signatures of each catchment." → "We used the permutation feature importance of random forest regression models to assess the relative importance of each source of uncertainty on the signatures."

**Introduction**

L53-56: Having limited knowledge of South American geography, these lines appear confusing to me. How exactly are the 'Andean glaciers', 'Patagonian Andes' and, 'Southern Andes' related? You mention that the Glaciers in the Patagonian Andes are the dominant source of ice loss in the Southern Andes, and that all Andean glaciers combined are one of the largest contributors to sea level rise. Am I right to assume that the Patagonian glaciers thus make up most of the total number of glaciers in the Andes?

L54: mass loss → ice mass loss

L60: in rivers with important glacier cover → in rivers in glacierized catchments.
It is unclear what is meant by 'important glacier cover'. In addition, the introduction of the term 'catchments' in relation to rivers, at this stage, helps support your decision to study glacier evolution on catchment-scale rather than for each individual glacier.

L60-61: The subsentence 'some of which […]' downplays the rest of the sentence and seems contradicting to the first sentence of *Study Area* (L100). Consider rewriting to e.g., 'However, recent studies have reported increased flows in rivers in glacierized catchments (Masiokas et al., 2019; Vries et al., 2023), with a growing number of rivers showing significant trends (p < 0.01) in the last decade (e.g., Santa Cruz; Pasquini et al., 2021).

L82: have examined the influence of → have simultaneously examined the influence of OR have compared the influence of OR have examined the respective influence.

L88: Unclear what is meant by 'the climate model chain components', this could be more explicit by stating e.g., '[…] the main source of uncertainty was associated to climate forcing data […]'

L92-97: The scenarios were tested […] → The resulting 1920 scenarios were simulated […]

**Study Area**

L106-107: I suggest to mention here why you aggregate the catchments into nine hydrological zones. I.e., 'For this or that purpose/analysis, the catchments were aggregated into nine hydrological zones.' This could help prevent the confusion that was brought up during the initial revision. Otherwise, the reasoning needs to be more explicitly mentioned in the method section.

L108: '[…], that showed a strong capacity to reproduce recent glacier changes.' Technically, 'that' refers to spatial patterns in precipitation and temperature, which cannot explain glacier changes. Either rephrase to '[…], that showed a strong capacity to reproduce recent spatial variability in glacier changes.' OR ''[…]. Precipitation and temperature showed a strong capacity to reproduce recent glacier changes.' or similar.

Figure 1: It is unclear to my why the names of the main catchments are included in all the figures. In my opinion it makes the figures more cluttered and less obvious. If you are not explicitly referring to these names, I suggest to remove them.

**Methodology**

L128: In fact, you are using the same version at the model used by Marzeion et al. (2012), since you set Pf to 1 (i.e., ignore it) and include a Tspinup (which is similar to their Tbias), correct?

L136: The term 'positive degree-months' has not been introduced with respect to Eq. 1.

L154: It is unclear what 'this' refers to. Consider rephrasing to 'Not considering frontal ablation is an acknowledged shortcoming of our study […]'

L163: Although these lines are rephrased with respect to the initial submission, it is still unclear to me what is meant by the residual term. Based on Maussion et al. (2019), it appears that this is a time-based correction of the observed mass balance. Is that correct? I suggest to mention explicitly what you have done.

L173: The Tspinup appears very similar to Tbias which is more often used in mass balance equations like Eq. 1. In fact, in Appendix A the term temperature bias is also used. For consistency and compatibility with other studies, consider adding Tbias to Eq. 1 and mention in step i that this parameter is initially set to 0 for reference period simulations.

L195: I suggest to explicitly introduce the distinction between historical and future uncertainty sources from the start and ideally use the same or similar headers as in 4.1.

L224: PMET outperformed ERA5 […] → In an earlier study, PMET outperformed ERA5 […]

L247: This description of the different bias correction methods is rather abstract. Although I understand that (explaining) these methods is not the focus of the study, I suggest to clarify.

For instance (L246-248): '[…] method that combines quantile-based delta change and bias correction methods. Thus, it not only preserves the quantile changes predicted by climate projections, but also corrects the biases of modelled time series with respect to those of the reference time series.' Here, instead of explaining the method, the second sentence repeats the first sentence. In addition, 'but also corrects the biases of modelled time series with respect to those of the reference timeseries', is essentially the main aim of any bias correction methods, so it doesn't provide any real information on what distinguishes this method from the others.

L269: of → for

L282: I recommend to introduce the term 'commitment runs' here, as that term is used in the figures, but never explicitly coupled to these 16 runs.

L284: You also corrected annual glacier area, volume and specific mass balance for the additional 16 runs.

Table 1: first DJF → (December – February) OR (December, January, and February)

L306: As indicated by one of the earlier reviews, the term RMSE is confusing because this usually refers to an error between predicted and actual (observed) values (hence 'error'), while you are comparing a number of predictions to another prediction using one of the selected options for each source of uncertainty. Is that right?

In that case, does it matter which option (e.g., PMET as reference climate data set) is chosen as the initial selection? For example, imagine the projected glacier evolution to be very similar for ERA5, CR2MET and MSWEP, but very different for PMET: then the obtained total RMSE with respect to the PMET run would be larger than with respect to one of the others. I assume that the RMSE is always computed with respect to the previously selected option (instead of the initial one)? Maybe you can add a note on this in the text.

**Results**

L326: 'While […] both years).' This sentence reads confusing while you could just say that 84.7% of glacier area in RGI6 was acquired in 2000, the majority (~65%) of the data in RGI7 was acquired in 2001. There is no need to hide that on average, there is only one year difference between the acquisition dates in the two datasets. You come back to this in the discussion where you state that the inclusion of local data and different processing (correction) techniques are more important than the acquisition dates.

L336: The use of the term normalized thickness appears confusing to me because you haven't considered real ice thicknesses. Perhaps you can refer to it as normalized or scaled volumes, or 'ice volume per catchment area'.

L349: spatial climate diversity.

L352: When you refer to glacier area, is this based on RGI6, RIG7 or does it refer to the entire study domain?

L383: The term 'future climate uncertainty' is not properly introduced. At first, it was unclear to me that this refers to the combined product of four other sources of uncertainty. This section can easily be misunderstood as a preview of the discussion of the respective importance of the six sources of uncertainty on glacio-hydrological modeling. I suggest to introduce this term/analysis in section 3.3.

L405: The results suggest that ice loss will vary according to the different sources of uncertainty $\rightarrow$ The results indicate variable ice loss depending on model choices.

L407: Mention the term 'commitment run' in brackets.

Header 4.3: Use same header as in 3, Hydrological importance of sources of uncertainty.

L448: accumulated $\rightarrow$ explained

L455: 'Consistently, [...]' Wasn't this to be expected?

**Discussion**

L475-477: This sentence is rather long and the last subsentence is unclear.

L490: may have peaked (2021 +- 15) $\rightarrow$ may have peaked in 2021 (+-15 year).

L496: more important for $\rightarrow$ more important than the volume data source for

L519: what was the other methodology that Watanabe et al. (2019) used? Could any aspect of their methodology explain the different differences between introduced uncertainties?

L533: when comparing your projections to those of Rounce et al. (2023), do you consider only the runs with ERA5 forcing or a combination of all historical climate data sets? I believe Rounce uses ERA5.

End of 5.4: In 5.2 you concluded that historical sources of uncertainty (ref climate followed by glacier attributes) are more important than future sources of uncertainty, while most other studies focus on only the future sources of uncertainty. This could be emphasized more.

**Conclusion**

L581: 'six sources of data uncertainty' $\rightarrow$ 'six sources of uncertainty associated with model choices'.

**Appendix A**

L614: Here you refer to Tspinup as the temperature bias. See earlier comment.

**Supplementary Material**

Table S2: Longitude – Latitude $\rightarrow$ Resolution (lat – lon)

---

## Author Response (AR2)

The manuscript provides an in-depth evaluation of the uncertainty in glacio-hydrological modeling of the Patagonian Andes using the Open Global Glacier Model (OGGM). Through an extensive series of simulations (1920 in total), the authors have quantified how various model choices influence glacier melt and runoff projections. They examined six different model choices (referred to as sources of uncertainty), that relate to both historical datasets (different glacier outlines, glacier volumes, and reference climate data) and future climate forcings (various general circulation models, emission scenarios, and bias correction methods). Additionally, the relative importance of each source of uncertainty was evaluated using a random forest regression method. The analysis revealed that reference climate data is the most critical source of uncertainty, even for metrics related to future projections.

The authors have caried out a rather comprehensive assessment of model uncertainty, which is adequately documented in this manuscript. In response to the feedback during the first round of revision, the authors have reorganized the discussion section and improved the figures and general clarity throughout the text.

I deem the manuscript fit for publication after a minor, mainly textual, revision. Please consider the more detailed list of suggestions below.

**R. Thank you for your positive feedback. We have carefully reviewed and incorporated your detailed suggestions in the revised version. We appreciate your input and believe the improvements have enhanced the overall clarity and quality of the manuscript**

**Abstract**

L24: area > 1km2 → with area > 1km2

**R. Changed as suggested.**

L25: Here, I would suggest removing the mention of the number of catchments and hydrological zones. It does not add to the clarity of the abstract and it is not of key importance.

**R. We have made the suggested revision and removed the mention of the number of catchments and hydrological zones from the abstract**

L25: Consider replacing "We used different glacier […]" by "As sources of uncertainty, we used different glacier […]" to prevent confusion about what is meant by *each source* in L30.

**R. Thanks for the suggestion. Changed as suggested.**

L30: "We used the permutation feature importance of random forest regression models to assess the relative importance of each source on the signatures of each catchment." → "We used the permutation feature importance of random forest regression models to assess the relative importance of each source of uncertainty on the signatures."

**R. Thanks for the suggestion. Changed as suggested.**

**Introduction**

L53-56: Having limited knowledge of South American geography, these lines appear confusing to me. How exactly are the 'Andean glaciers', 'Patagonian Andes' and, 'Southern Andes' related? You mention that the Glaciers in the Patagonian Andes are the dominant source of ice loss in the Southern Andes, and that all Andean glaciers combined are one of the largest contributors to sea level rise. Am I right to assume that the Patagonian glaciers thus make up most of the total number of glaciers in the Andes?

**R. The terms "Andean glaciers," "Patagonian Andes," and "Southern Andes" refer to specific geographical features within the broader Andes mountain range in South America. Here's how they are related:**

**- Andean Glaciers: This term refers to the glaciers located throughout the Andes mountains. The Andes stretch along the western edge of South America, from Venezuela in the north to Chile and Argentina in the south.**

**- Patagonian Andes: This term refers to the southernmost portion of the Andes mountains, located primarily in southern Chile and Argentina.**

**- Southern Andes: The Southern Andes (RGI region 17) typically refer to the segment of the Andes that lies south of approximately 25°S latitude. This region includes the Patagonian Andes and extends towards the southern tip of South America.**

**In answer to your question, yes, you're right. In this region, glaciers represent 82% of the total glacierised area of the Andes at the time of the inventory (see Study Area section). In the sentences indicated by the comment, we have added the latitudinal ranges.**

L54: mass loss → ice mass loss

**R. Thanks for the suggestion. Changed as suggested.**

L60: in rivers with important glacier cover → in rivers in glacierized catchments. It is unclear what is meant by 'important glacier cover'. In addition, the introduction of the term 'catchments' in relation to rivers, at this stage, helps support your decision to study glacier evolution on catchment-scale rather than for each individual glacier.

**R. This has been addressed in the reformulation of the sentence (see next comment)**

L60-61: The subsentence 'some of which […]' downplays the rest of the sentence and seems contradicting to the first sentence of *Study Area* (L100). Consider rewriting to e.g., 'However, recent studies have reported increased flows in rivers in glacierized catchments (Masiokas et al., 2019; Vries et al., 2023), with a growing number of rivers showing significant trends ($p < 0.01$) in the last decade (e.g., Santa Cruz; Pasquini et al., 2021).

**R. Thanks for the suggestion. We have rewritten the sentences following your suggestion: "Nevertheless, recent studies have reported increased river flows in catchments with important glacierized area (Masiokas et al., 2019; Vries et al., 2023), with a growing number of rivers showing significant trends ($p < 0.01$) in the last decade (e.g., Santa Cruz; Pasquini et al., 2021)".**

L82: have examined the influence of → have simultaneously examined the influence of OR have compared the influence of OR have examined the respective influence.

**R. Thanks for the suggestion. Changed as suggested ("have compared the influence").**

L88: Unclear what is meant by 'the climate model chain components', this could be more explicit by stating e.g., '[…] the main source of uncertainty was associated to climate forcing data […]'

**R. Thanks for the suggestion. We have rewritten the sentence to directly mention the main sources of uncertainty (GCMs and SSPs).**

L92-97: The scenarios were tested […] → The resulting 1920 scenarios were simulated […]

**R. Changed as suggested**

**Study Area**

L106-107:  I suggest to mention here why you aggregate the catchments into nine hydrological zones. I.e., 'For this or that purpose/analysis, the catchments were aggregated into nine hydrological zones.' This could help prevent the confusion that was brought up during the initial revision. Otherwise, the reasoning needs to be more explicitly mentioned in the method section.

**R. Following your suggestion, we have clarified the purpose of aggregating glaciers into the catchments and then catchments into hydrological zones: "To better analyse the spatial variability in hydrological dynamics and to provide a framework for aggregating projected glacio-hydrological changes, the glaciers in the study area were grouped into catchments, which were then aggregated into nine hydrological zones (Fig. 1). This catchment-scale aggregation is consistent with ongoing efforts to integrate global glacier simulations into hydrological models (Hanus et al., 2024; Pesci et al., 2023; Wiersma et al., 2022), which often operate at the catchment or river scale" This adjustment aims to prevent confusion noted in the initial revision and ensures the reasoning is explicitly stated.**

L108: '[…], that showed a strong capacity to reproduce recent glacier changes.' Technically, 'that' refers to spatial patterns in precipitation and temperature, which cannot explain glacier changes. Either rephrase to '[…], that showed a strong capacity to reproduce recent spatial variability in glacier changes.' OR ''[…]. Precipitation and temperature showed a strong capacity to reproduce recent glacier changes.' or similar.

**R. Thank you for your suggestions. Following the argument of aggregation to hydrological zones, we have clarified the spatial aggregation: "The zones were selected based on the spatial patterns of precipitation and temperature, which have previously shown a strong capacity to explain recent spatial variability in glacier change (Caro et al., 2021)".**

Figure 1: It is unclear to my why the names of the main catchments are included in all the figures. In my opinion it makes the figures more cluttered and less obvious. If you are not explicitly referring to these names, I suggest to remove them.

**R. Thank you for your feedback. We have followed your suggestion and removed the names of the main catchments from the figures (except in Fig. 1) to reduce clutter and enhance clarity.**

**Methodology**

L128: In fact, you are using the same version at the model used by Marzeion et al. (2012), since you set Pf to 1 (i.e., ignore it) and include a Tspinup (which is similar to their Tbias), correct?

**R. No, we use an adapted version of the mass balance model presented in Marzeion et al. (2012). In particular, we do not use a bias correction β\* in our equation and the calibration strategy is quite different. Part of this new calibration strategy is a dynamic spinup, where we search for an initial past glacier state in 1980 from which the glacier evolves in a way that matches the RGI area. To find this initial state, we use Tspinup as described in point (iv) of L171. Therefore, Tspinup is only used to find an initial glacier state in 1980 and nowhere else. Finally, the model of Marzeion et al. was glacier-wide (with solid precipitation computed at the glacier top and melt at the terminus elevation) while the OGGM standard model is computed for all elevation bands.**

L136: The term 'positive degree-months' has not been introduced with respect to Eq. 1.

**R. Thanks for pointing that out. We have added the definition of the term in parenthesis.**

L154: It is unclear what 'this' refers to. Consider rephrasing to 'Not considering frontal ablation is an acknowledged shortcoming of our study […]'

**R. Thanks for the suggestion. Changed as suggested**

L163: Although these lines are rephrased with respect to the initial submission, it is still unclear to me what is meant by the residual term. Based on Maussion et al. (2019), it appears that this is a time-based correction of the observed mass balance. Is that correct? I suggest to mention explicitly what you have done.

**R. Thanks for pointing this out. The residual term is not time-based (either in our manuscript or in Maussion et al. 2019). In Maussion et al. (2019), the mass balance calibration relies on the concept of finding t\*, a year in the past when the current static glacier geometry would be in equilibrium with the climatic mass balance of that particular year. However, we now use a completely different strategy to find the apparent mass balance (we simply add a constant offset to the mass balance**

**profile to find a mass balance in equilibrium with the static glacier geometry). We have tried to make this clearer in the manuscript.**

L173: The Tspinup appears very similar to Tbias which is more often used in mass balance equations like Eq. 1. In fact, in Appendix A the term temperature bias is also used. For consistency and compatibility with other studies, consider adding Tbias to Eq. 1 and mention in step i that this parameter is initially set to 0 for reference period simulations.

**R. Thank you for highlighting this issue. We agree that the similarity between "Tspinup" and "Tbias" could lead to confusion. To address this, we will revise Appendix A to consistently use the term "Tspinup". This discrepancy arose from changes in variable naming between the initial manuscript and the current version, and we apologise for any confusion caused. This interestingly also led to name changes in the OGGM codebase to avoid confusion [link].**

L195: I suggest to explicitly introduce the distinction between historical and future uncertainty sources from the start and ideally use the same or similar headers as in 4.1.

**R. We have addressed this suggestion by explicitly introducing the distinction between historical and future sources of uncertainty from the beginning. In addition, we have ensured that the headings in the Methods and Results sections are similar to maintain consistency and clarity throughout the manuscript.**

L224: PMET outperformed ERA5 […] → In an earlier study, PMET outperformed ERA5 […]

**R. Thanks for the suggestion. Changed as suggested.**

L247: This description of the different bias correction methods is rather abstract. Although I understand that (explaining) these methods is not the focus of the study, I suggest to clarify. For instance (L246-248): '[…] method that combines quantile-based delta change and bias correction methods. Thus, it not only preserves the quantile changes predicted by climate projections, but also corrects the biases of modelled time series with respect to those of the reference time series.' Here, instead of explaining the method, the second sentence repeats the first sentence. In addition, 'but also corrects the biases of modelled time series with respect to those of the reference timeseries', is essentially the main aim of any bias correction methods, so it doesn't provide any real information on what distinguishes this method from the others.

**R. Thank you for this comment. We appreciate the opportunity to enhance the clarity of our descriptions. We have revised the text to better distinguish the methods and clarify their specific characteristics. Specifically, we now emphasize that:**

**- Quantile Delta Mapping (QDM) not only corrects biases but also preserves the projected changes across the entire distribution of the climate variable, distinguishing it from methods like Mean and Variance Scaling (MVA).**

**- Multivariate Bias Correction with N-dimensional PDF transformation (MBCn) goes a step further by correcting biases in multiple variables simultaneously while preserving the relationships between these variables (in our case, precipitation and temperature).**

L269: of → for

**R. Thanks for the suggestion. Changed as suggested.**

L282: I recommend to introduce the term 'commitment runs' here, as that term is used in the figures, but never explicitly coupled to these 16 runs.

**R. Thanks for the recommendation. We have introduced the term "commitment run" in the specific sentence.**

L284: You also corrected annual glacier area, volume and specific mass balance for the additional 16 runs.

**R. The specific part of the sentence has been removed to avoid confusion about the variables extracted during the commitment runs.**

Table 1: first DJF → (December – February) OR (December, January, and February)

**R. Thanks for the suggestion. Changed as suggested.**

L306: As indicated by one of the earlier reviews, the term RMSE is confusing because this usually refers to an error between predicted and actual (observed) values (hence 'error'), while you are comparing a number of predictions to another prediction using one of the selected options for each source of uncertainty. Is that right?

**R. We recognize that the use of RMSE might be confusing in this context. Typically, RMSE refers to the error between predicted and observed values, as you noted. However, in the permutation feature importance, the RMSE is used to measure the difference between predictions from the original model and those obtained after permuting individual features, rather than comparing predictions to actual observed values. The complete procedure is as follows:**

1. **Baseline Performance: First, the model's performance is measured using a chosen metric (such as RMSE) with all features intact. This provides a baseline performance score.**

2. **Feature Permutation: Next, the values of a single feature are shuffled, breaking the relationship between that feature and the target variable. The model is then run again using this modified dataset.**

3. **Performance Comparison: The performance of the model with the permuted feature is compared to the baseline performance. If the performance metric (i.e., RMSE) worsens significantly after shuffling the feature, it indicates that the feature is important to the model. Conversely, if the performance remains relatively unchanged, the feature is less important.**

4. **Repetition for All Features: This process is repeated for each feature in the dataset, allowing for the assessment of the relative importance of all features based on how much their permutation impacts model performance.**

**We have clarified this procedure in the text.**

In that case, does it matter which option (e.g., PMET as reference climate data set) is chosen as the initial selection? For example, imagine the projected glacier evolution to be very similar for ERA5, CR2MET and MSWEP, but very different for PMET: then the obtained total RMSE with respect to the PMET run would be larger than with respect to one of the others. I assume that the RMSE is always computed with respect to the previously selected option (instead of the initial one)? Maybe you can add a note on this in the text.

**R. Thanks for the question. It does not matter because this process is repeated for all features, and the choice of a reference among the different model options is solely for comparison purposes. The RMSE is consistently calculated relative to the selected reference, but since every feature undergoes the same permutation process, the results remain valid regardless of the initial choice. This ensures that the analysis accurately reflects the importance of each feature across all potential reference scenarios. This has been clarified in the procedure (see previous comment).**

**Results**

L326: 'While […] both years).' This sentence reads confusing while you could just say that 84.7% of glacier area in RGI6 was acquired in 2000, the majority (~65%) of the data in RGI7 was acquired in 2001. There is no need to hide that on average, there is only one year difference between the acquisition dates in the two datasets. You come back to this in the discussion where you state that the inclusion of local data and different processing (correction) techniques are more important than the acquisition dates.

**R. Thank you for your comment. Following your recommendation, we have adjusted the sentence to improve the clarity of the comparison.**

L336: The use of the term normalized thickness appears confusing to me because you haven't considered real ice thicknesses. Perhaps you can refer to it as normalized or scaled volumes, or 'ice volume per catchment area'.

**R. We have replaced the term normalised thickness with normalised volume, mentioning that the normalisation was performed using the catchment area (e.g. Fig. 4).**

L349: spatial climate diversity.

**R. Thanks for the suggestion. Changed as suggested.**

L352: When you refer to glacier area, is this based on RGI6, RIG7 or does it refer to the entire study domain?

**R. The calculation is based on RGI6. We have modified the sentence to clarify this point. This was only mentioned in the caption of Fig. 4**

L383: The term 'future climate uncertainty' is not properly introduced. At first, it was unclear to me that this refers to the combined product of four other sources of uncertainty. This section can easily be misunderstood as a preview of the discussion of the respective importance of the six sources of uncertainty on glacio-hydrological modeling. I suggest to introduce this term/analysis in section 3.3.

**R. Thanks for the comment. We have introduced the term in Section 3.3 as you suggested: "Finally, to assess the individual impact of each climate uncertainty source, we estimated the future climate**

**uncertainty, which we defined as the standard deviation across different reference climates (n = 4), GCMs (n = 4), SSPs (n = 4), and bias correction methods (n = 4), resulting in 480 possible combinations."**

L405: The results suggest that ice loss will vary according to the different sources of uncertainty → The results indicate variable ice loss depending on model choices.

**R. Thanks for the suggestion. Changed as suggested.**

L407: Mention the term 'commitment run' in brackets.

**R. Thanks. We have added this clarification.**

Header 4.3: Use same header as in 3, Hydrological importance of sources of uncertainty.

**R. Thanks for the suggestion. Changed as suggested.**

L448: accumulated → explained

**R. Thanks for the suggestion. Changed as suggested.**

L455: 'Consistently, […]' Wasn't this to be expected?

**R. We have replaced the "Consistently" with "As expected". This is explained by the fact that some signatures/metrics are calculated only taking into account the reference period and therefore the importance of future sources of uncertainty (GCM, SSP and bias correction method) is expected to be zero.**

**Discussion**

L475-477: This sentence is rather long and the last subsentence is unclear.

**R. Thanks for the comment. We have split the original sentence into two: "Despite the dependence of the specific mass balance (units: kg m$^{-2}$ yr$^{-1}$) on the emission scenarios (Fig. S1), the ice melt component of runoff (units: m$^3$ s$^{-1}$) did not show a clear dependence on the emission scenario (Fig. 9). This is because the runoff is not normalized by glacier area, which decreases throughout the century (Fig. S1)"**

L490: may have peaked (2021 +- 15) → may have peaked in 2021 (+-15 year).

**R. Thanks for the suggestion. Changed as suggested.**

L496: more important for → more important than the volume data source for

**R. Thanks for the suggestion. Changed as suggested.**

L519: what was the other methodology that Watanabe et al. (2019) used? Could any aspect of their methodology explain the different differences between introduced uncertainties?

**R. Thank you for your question. Although both studies applied the iterative calibration process (adjusting a precipitation factor, followed by a melt factor, and finally a temperature bias parameter) as outlined by Huss and Hock (2015), it's important to highlight several key differences that complicate a direct comparison between the two. These differences include:**

**- The studies utilized different modelling systems: GloGEMflow in one and HYOGA2 in the other.**

**- Compagno et al. (2021) used three historical climate data products (E-OBS, ERA-I, and ERA-5), whereas Watanabe et al. (2019) employed six historical climates, derived from combinations of two air temperature datasets and three precipitation datasets.**

**- The focus areas differ, with Compagno et al. (2021) studying Scandinavia and Iceland, while Watanabe et al. (2019) focused on High Mountain Asia. These regions may exhibit varying discrepancies between climate products. Additionally, Compagno et al. (2021) modelled 3,411 glaciers in Scandinavia, compared to the 28 randomly selected glaciers analysed by Watanabe et al. (2019).**

**For the sake of brevity, we have decided to leave the text as it is.**

L533: when comparing your projections to those of Rounce et al. (2023), do you consider only the runs with ERA5 forcing or a combination of all historical climate data sets? I believe Rounce uses ERA5.

**R. We used all possible combinations (n = 480 per SSP). We have added a clarification in the caption of Fig. S5 with the climate products used in both studies.**

End of 5.4: In 5.2 you concluded that historical sources of uncertainty (ref climate followed by glacier attributes) are more important than future sources of uncertainty, while most other studies focus on only the future sources of uncertainty. This could be emphasized more.

**R. Thank you for your comment. This aspect was more strongly emphasised in the original submission, but to reduce redundancy we decided not to repeat this idea in several sections of the discussion. However, we have adjusted the last point of the Conclusions (third point) to emphasise this idea more clearly.**

**Conclusion**

L581: 'six sources of data uncertainty' → 'six sources of uncertainty associated with model choices'.

**R. Thanks for the suggestion. Changed as suggested.**

**Appendix A**

L614: Here you refer to Tspinup as the temperature bias. See earlier comment.

**R. Thanks for pointing this out. We have corrected the parameter name (Tspinup)**

**Supplementary Material**

Table S2: Longitude – Latitude → Resolution (lat – lon)

**R. Thanks for the suggestion. Changed as suggested.**

**Anonymous referee #1**

Review revised manuscript "Unravelling the sources of uncertainty in glacier runoff projections in the Patagonian Andes" by Aguayo et al.

Compared to the previous submission, the manuscript has improved a lot and I think that the discussion and figures have become much clearer. Well done!

I still have a few comments that I think should be addressed before the manuscript can be published. I formulate them here below in order of the manuscript (thus not necessarily in order of importance).

**R. Thank you for your constructive feedback and for acknowledging the improvements made to the manuscript. We appreciate your recognition of the clearer discussion and figures. We have addressed all the comments you provided, and we believe these revisions have further strengthened the manuscript. Below, we detail how we have responded to each of your points**

1. In the previous round I commented on the use of "catchments" in the manuscript. In the revised version, this has not been removed, but less results are mentioned in terms % of catchment area. While I do see the added value of the results aggregated to catchments in the maps, I still think that the catchment division is not really needed here and, apart from the maps, is also not really used. Why not removing the aggregation to catchment area and rather focus on the hydrological zones and the individual glaciers? If there is a good reason to stay with the catchment aggregation, I would 1) explain what the "catchments" are – a catchment does not have a clear definition without explanation, i.e. the catchments could also have been smaller. How were these derived? What level do they represent? 2) Incorporate throughout the results a description of what came out of this catchment aggregation comparison, to make sure that the reader also understands the "catchment level" results.

**R. Thank you for your comments and for raising important points about the use of catchment-scale aggregation in our manuscript. We appreciate the opportunity to clarify our rationale and address your concerns. Here's why we believe it is important to maintain catchment-scale aggregation in our study:**

**The level of aggregation is a critical aspect of glacio-hydrological studies, and we recognise the challenges associated with selecting the optimal scale. Most previous studies have explored different spatial extents, ranging from gauged mountain catchments (e.g., Huss et al., 2014; Mackay et al., 2019) to major river basins (e.g., Huss and Hock, 2018; Ultee et al., 2022; Wimberly et al., 2024) and even global scales (e.g., GlacierMIP2). Our study differs by focusing on the regional scale and comparing multiple sources of data uncertainty, which required a specific scale of aggregation for effective analysis.**

By aggregating results at the catchment scale, we maintain consistency with previous studies assessing future climate impacts on streamflow (see Table A1 in Van Tiel et al., 2020). This consistency is critical for comparison purposes and is consistent with ongoing efforts to integrate global glacier simulations into regional/global hydrological models (e.g., GloGEM in Wiersma et al., 2022; OGGM in Pesci et al., 2023; OGGM in Hanus et al., 2024; PyGEM in Long et al. 2024), which often produce results at the catchment/river scale.

In addition, aggregating data at the catchment scale significantly enhances future research by enabling comparisons with other regional hydrological studies. For example, we have made the complete results available at the catchment scale in the Zenodo repository (see Data availability; >60 downloads). This availability supports additional hydrological research and provides crucial insights for future modelers, such as:

- Identifying the primary sources of uncertainty in specific catchments of the Patagonian Andes.
- Determining which modelling decisions should be prioritized to accurately estimate, for example, peak water year.

As you pointed out, the catchment scale remains in the maps but is also used in the Feature Permutation Importance procedure because the random forest models are based on catchment-wide glacier runoff/melt data. This choice allowed us to assess the impact of factors such as the varying number of glaciers within each catchment due to the use of different glacier inventories. While it would have been possible to use hydrological zones as the aggregation scale, we found this approach to be too broad given the significant spatial variability within hydrological zones.

Finally, we acknowledge that the initial introduction of catchments in our manuscript was inadequate, and we appreciate your feedback highlighting this issue. We recognise that our explanation lacked sufficient detail, leaving room for ambiguity regarding the delineation. To address this, we have revised the study area section to better introduce the catchments.

Wiersma, P., Aerts, J., Zekollari, H., Hrachowitz, M., Drost, N., Huss, M., Sutanudjaja, E. H., and Hut, R.: Coupling a global glacier model to a global hydrological model prevents underestimation of glacier runoff, Hydrol. Earth Syst. Sci., 26, 5971–5986, https://doi.org/10.5194/hess-26-5971-2022, 2022.

Pesci, M. H., Schulte Overberg, P., Bosshard, T., & Förster, K. (2023). From global glacier modeling to catchment hydrology: bridging the gap with the WaSiM-OGGM coupling scheme. Frontiers in Water, 5, 1296344. https://doi.org/10.3389/frwa.2023.1296344

Hanus, S., Schuster, L., Burek, P., Maussion, F., Wada, Y., and Viviroli, D.: Coupling a large-scale glacier and hydrological model (OGGM v1.5.3 and CWatM V1.08) – towards an improved

representation of mountain water resources in global assessments, Geosci. Model Dev., 17, 5123–5144, https://doi.org/10.5194/gmd-17-5123-2024, 2024.

Long, J., Wang, L., Chen, D., Li, N., Zhou, J., Li, X., et al. (2024). Hydrological projections in the Third Pole using artificial intelligence and an observation-constrained cryosphere-hydrology model. Earth's Future, 12, e2023EF004222. https://doi.org/10.1029/2023EF004222

2. The sentences describing "xx% of the glacier area" has reached this or that, came across as a little confusing, and I wonder if they could be formulated as number of glaciers (abs or relative), and then in brackets how much of the glacier area that is?

**R. In this study, the way the results are communicated and summarised is inherently complex due to the large number of simulated glaciers (n = ~ 2,000), scenarios (n = 1,920), metrics (n = 10) and variables (glacier runoff and melt) involved. In an earlier version, we considered using the number of glaciers, but this approach was less informative as a few dozen glaciers represent almost the entire glacier area (mainly glaciers in the Patagonian Icefields), and the number of glaciers varies between inventories (RGI6 and 7), adding an additional layer of complexity. Instead, we have decided to present the results in terms of catchment area and glacier area (only glacier area in the revised versions). We think this method better captures the extent of sources of uncertainty across glaciated areas. By focusing on area rather than individual glacier, we aim to provide a more comprehensive understanding of the spatial influence of historical and future uncertainty sources on glacier hydrological dynamics.**

3. The sentence "Highlighting the choices we make in the calibration" in the abstract, could some hints be provided on what it has an effect (baseline + model parameters + future starting point(?), as is presented in the discussion)?

**R. Thanks for the suggestion. We have adjusted the sentence: "In contrast, the reference climate was the main source in 69 % ± 22 % of the glacier area, highlighting the impact of calibration choices on baseline conditions, model parameters, and the initial starting point for future projections"**

4. P2 L58 (revised manuscript) – should it be relative contribution?

**R. Thanks for the suggestion. Yes, it should be relative contribution. Changed as suggested.**

5. P2 L63 – I am not sure to understand how limitations in the understanding of glacier processes in the Patagonian Andes can be addressed by downscaling methods?

**R. Thanks for the question. We have clarified the first sentences of the paragraph: "Despite advances in glacier research, modelling efforts in the Patagonian Andes remain constrained by limited data for calibration and validation. For example, to circumvent the limited ground-based atmospheric data, many modelling studies have used dynamic and/or statistical downscaling methods based on global climate reanalyses (Table S1)."**

6. P2 L65 "have overestimated" – all of the studies?

**R. According to the detailed analysis of Sauter (2020), previous studies have overestimated the icefield-wide precipitation with values greater than 7-8 m w.e. yr$^{-1}$ (Table 1 in Sauter 2020). According to the same study, the icefield-wide precipitation averages (period 2010–2016) are likely to be within 5.38 ± 0.59 and 6.09 ± 0.64 m w.e. yr$^{-1}$ on the NPI and 5.06 ± 0.51 and 5.99 ± 0.59 m w.e. yr$^{-1}$ on the SPI**

7. P3 L70 – SPI and NPI are not yet defined

**R. Thanks for the suggestion. We have defined SPI and NPI in the first mention.**

8. L62-L75 – I think that this paragraph is hard to follow, what message does it want to convey? Could it be reformulated?

**R. Following your previous comment (5. P2 L63), we have adjusted the first sentence to clarify the message of the paragraph. The revised first sentence now better emphasises that despite progress in glacier research, significant challenges remain due to gaps in data for calibration and validation. This sets the stage for a discussion of the specific problems (e.g. climate discrepancies) and limitations of glacier modelling efforts in the Patagonian Andes.**

9. P3 L83 – Sentence starting with "Huss et al. (2014)" – suggest to connect with previous sentence and add something along the lines of "Such studies have shown that xxx"

**R. Thanks for the comment. We have modified the sentence to improve the connection with the following examples (Huss et al. 2014; Mackay et al. 2019), highlighting the spatial scale of the previous studies. "However, few studies have compared the influence of multiple components of the modelling chain on projected glacio-hydrological changes, and those that have been conducted are typically local (basin-specific), limiting the broader applicability of their conclusions. For instance, Huss et al. (2014)…"**

10. P4 L108 "that showed a strong capacity to reproduce recent glacier changes" – what is meant here?

**R. Thanks for the question. Caro et al. (2021; already cited in the main text) used machine learning models (LASSO) to explore variables that explain the spatial variance of glacier surface area in the Andes. They found that the spatial variability of climatic variables had a higher explanatory power than morphometric variables. This means that the spatial patterns of precipitation and temperature within these zones were found to be accurate indicators or predictors of how glaciers have changed over time. A clarification has been added to the specific sentence.**

11. P5 L124 "to model the evolution of all the glaciers" – does that contradict with the catchment requirement of having a glacier area larger than 0.1%?

**R. Thank you for pointing out this inconsistency. We recognise that the statement "to model the evolution of all glaciers" contradicts the requirement that only glaciers with an area greater than 1 km² are considered, as well as the catchment requirement of having a glacier area greater than 0.1% (although all glaciers meeting this requirement are in catchments with a glacier area greater than 0.1%). To resolve this, we have removed the specific part of the sentence to accurately reflect the scope of our study ("all glaciers ").**

12. P6 L134 "obtained from the nearest grid point"- possibly explain grid point, as it could also have been a station (the use of gridded climate data is not yet clear?)

**R. Thanks for the comment. We have clarified this sentence: "The climate variables are obtained from the nearest grid point of the climate gridded product (see Section 3.2.2)."**

13. P6 L 135 "a value commonly used in the study area" – not only there, but everywhere, since it is the global average tropospheric lapse rate?

**R. Thanks for the comment. We have clarified the sentence: "…a value commonly used (see local examples in Table S1)."**

14. P7 L 162 "mass balance" – as it is written, it could be read as being zero? Should it be mass balance profile?

**R. Thanks for pointing this out. It should read that the apparent mass balance integrated over the whole glacier must be zero. See adapted paragraph.**

15. P7 L173 "using the reference climate" – would "perturbed reference climate" work here to avoid confusion

**R. Thank you for your suggestion, we adapted this as suggested for clarity.**

16. P12 L 305 – I commented on this before, but unfortunately it has not yet become clear; the change in model performance is calculated as the change in RMSE. But what is taken here is "Sim" and what as "obs"? In that sense, there is no model performance, as there are no measurements? Or what is referred to here?

**R. Thanks for the question, this point was also raised by the other reviewer. We recognize that the use of RMSE might be misleading considering that RMSE commonly refers to the error between predicted and observed values. In our study, the "Permutation Feature Importance" uses RMSE to measure the difference between predictions from the original model and those obtained after permuting individual features, rather than comparing predictions to actual observed values. The complete procedure is as follows:**

1. **Baseline Performance: First, the model's performance is measured using the chosen metric (i.e., RMSE) with all features intact. This provides a baseline performance score.**

2. **Feature Permutation: Next, the values of a single feature are shuffled, breaking the relationship between that feature and the target variable. The model is then run again using this modified dataset.**

3. **Performance Comparison: The performance of the model with the permuted feature is compared to the baseline performance. If the performance metric worsens significantly after**

**shuffling the feature, it indicates that the feature is important to the model. Conversely, if the performance remains relatively unchanged, the feature is less important.**

4. **Repetition for All Features: This process is repeated for each feature in the dataset, allowing for the assessment of the relative importance of all features based on how much their permutation impacts model performance.**

**We have clarified this procedure in the text.**

17. P19 L 409 – suggest to first mention precipitation projections (physical reason) and then the low ice volume (statistical reason)

**R. Thank you for your suggestion. We have rearranged the sentence to first address the precipitation projections, followed by the mention of low ice volume.**

18. P19 L412 – I was a bit confused as the sentence starts with "At the hydrological zone scale"… but this was also discussed in the sentences before?

**R. Thank you for pointing that out. To improve clarity and avoid redundancy, we have revised the sentence by removing the phrase "At the hydrological zone scale".**

19. Figure 10 – what is the aggregation scale used here? Glaciers, catchments, zones?

**R. Thanks for the question. The aggregation scale corresponds to catchments. From Section 3.5: "For this analysis, we selected 329 catchments with at least one glacier (area > 1 km2) in both inventories". We have clarified this in Figure 10 ("Each boxplot aggregates the results obtained from Permutation Feature Importance using the 329 catchments").**

20. P23 L 476-477 – Here I am not sure to completely follow, so it may be good to write the reasoning more explicitly. If the ice melt volume does not show a dependance on emission scenario, then that suggests that regardless of emission scenario, we get approximately the same amount of melt volume? It means that the increased melt is offset by the decreased glacier area? If yes, that would be good to describe as in terms of hydrology, the absolute volume is of importance.

**R. Thanks for the question. Although the specific mass balance shows a clear dependence on the emission scenarios (Fig. S1), the ice melt component of the runoff does not show a similar dependence (Fig. 9). This is because although higher emission scenarios lead to increased melt rates, the glacier area shrinks significantly throughout the century (Fig. S1). As a result, the total ice melt volume remains relatively constant across different emission scenarios. In other words, the increased melt is largely offset by the reduced glacier area. This balance explains why the ice melt volume does not vary significantly with emission scenarios, highlighting the importance of considering both melt rates and glacier area when assessing hydrological impacts. This clarification has been included in the revised version.**

21. P24 L 500-503 – does it therefore also affect the "starting conditions" of the future runs?

**R. Yes, the reference climate does affect the "starting conditions" of future runs. Since it establishes the baseline conditions against which future changes are assessed, it directly influences the initial setup and parameters of the models, impacting how future climate scenarios are simulated and interpreted.**

22. P24 L504 "low sensitivity" – should this be "high sensitivity" to fit with the following sentence?

**R. Thanks for the question. No, "low sensitivity" is correct in this context. The sentence emphasises that despite the overall variability in conditions, only a few regions showed low sensitivity to the reference climate. The following sentence explains that this low sensitivity is unusual because most climate products have significant differences in solid precipitation, which generally leads to high sensitivity in glacier runoff and melt evolution.**

23. P24 section 5.3 Here it would be good to have an actual discussion on how the model parameters differ with different reference climates. This would allow some discussion on how certain parameters + reference climate combinations lead to certain change in glacier runoff + glacier melt

**R. Thank you for your comment. In response, we have added a new figure to the supplementary material to illustrate how model parameters differ with various reference climates. Additionally, we have expanded the discussion in Section 5.2: "The reference climate influences temperature and precipitation patterns, which directly shape the seasonal response of glaciers by affecting both melt and accumulation processes. Moreover, the choice of reference climate plays a critical role in parameter calibration, subsequently impacting the model's sensitivity to climate change. For example, Fig. S5 demonstrates that the temperature sensitivity parameter varies significantly with**

**the reference climate, more so than other factors like glacier geometry and thickness. This highlights how specific combinations of model parameters and reference climates can lead to different outcomes in terms of glacier runoff and melt responses".**

[Figure]

**Temperature sensitivity for each historical scenario (n = 16). The historical conditions involved in the calibration process considered the geometry obtained from the glacier inventories (RGI6 and 7), the volume obtained from ice thickness datasets (F19 and M22), and the reference climate dataset (PMET, CR2MET, MSWEP and ERA5). More details on the historical conditions can be found in Section 3.2.1. Each boxplot aggregates all simulated glaciers (glacier area > 1 km2), corresponding to 2,034 and 1,837 glaciers for RGI6 and RGI7, respectively.**

24. P27 L607 "Downstream hydrology" – please specify which hydrological processes you mean

**R. Thanks for the comment. We have specified some hydrological processes in parentheses.**